# Role of ambient ammonia in particulate ammonium formation at a rural site in the North China Plain

Zhaoyang Meng[1], Xiaobin Xu[1], Weili Lin[2], Baozhu Ge[3], Yulin Xie[4,5], Bo Song[4], Shihui Jia[1,6], Rui Zhang[7,8], Wei Peng[1], Ying Wang[1], Hongbin Cheng[1], Wen Yang[7], Huarong Zhao[1]

[1] State Key Laboratory of Severe Weather & Key Laboratory for Atmospheric Chemistry of CMA, Chinese Academy of Meteorological Sciences, Beijing 100081, China

[2] CMA Meteorological Observation Centre, Beijing 100081, China

[3] State Key Laboratory of Atmospheric Boundary Layer Physics and Atmospheric Chemistry, Institute of Atmospheric Physics, Chinese Academy of Sciences, Beijing 100029, China

[4] University of Science and Technology Beijing, Beijing 100083, China

[5] Baotou Steel Group Mining Research Institute, Baotou 014010, China

[6] South China University of Technology, Guangzhou 510641, China

[7] Chinese Research Academy of Environmental Sciences, Beijing 100012, China

[8] Beijing Municipal Research Institute of Environmental Protection, Beijing 100037, China

*Correspondence to*: Zhaoyang Meng (mengzy@camscma.cn)

**Abstract.** The real-time measurements of $NH_3$ and trace gases were conducted, in conjunction with semi-continuous measurements of water-soluble ions in $PM_{2.5}$ at a rural site in the North China Plain (NCP) from May to September 2013 in order to better understand of chemical characteristics for ammonia, and of the impact on formation of secondary ammonium aerosols in the NCP. Extremely high $NH_3$ and $NH_4^+$ concentrations were observed after a precipitation event within 7-10 days following urea application. Elevated $NH_3$ levels coincided with elevated $NH_4^+$, indicating that $NH_3$ likely influenced particulate ammonium mass. For the sampling period, the average oxidation/conversion ratios for $SO_4^{2-}$ (SOR), $NO_3^-$ (NOR) and $NH_4^+$ (NHR) were estimated to be 0.64, 0.24 and 0.30, respectively. The increased $NH_3$ concentrations mainly from agricultural activities and regional transport, coincided with the prevailing meteorological conditions. The high $NH_3$ level with NHR about 0.30 indicate that the emission of $NH_3$ in the NCP is much higher than needed for aerosol acids neutralization and $NH_3$ plays an important role in the formation of secondary aerosols as a key neutralizer. The hourly data obtained were used to investigate gas-aerosol partitioning characteristics using the thermodynamic equilibrium model ISORROPIA II. Modelled $NO_3^-$, $SO_4^{2-}$, and $NH_3$ values agree well with the measurements, while the modelled $NH_4^+$ largely underestimate the measurements.

Our observation and modelling results indicate that strong acids in aerosol are completely neutralized. Additional $NH_4^+$ exists in aerosol, probably a result of presence of substantial amount of oxalic and other diacids.

**Keywords:** Ambient ammonia; ammonium in $PM_{2.5}$; the conversion ratio of $NH_4^+$; thermodynamic equilibrium; agricultural activity; North China Plain.

**1 Introduction**

Ammonia ($NH_3$) is a very important alkaline constituent in the atmosphere, plays an important role in
atmospheric chemistry and is closely related to ecosystems. $NH_3$ has both direct and indirect impacts on critical environmental issues, including regional fine particles, acid rain, and eutrophication (Roelle and Aneja, 2002; Krupa, 2003; Reche et al., 2015). In addition, $NH_3$ is a key species for neutralising $H_2SO_4$ and $HNO_3$ in the atmosphere and forming $(NH_4)_2SO_4$, $NH_4HSO_4$, and $NH_4NO_3$ (Erisman and Schaap, 2004; Walker et al., 2004), which are major inorganic components of fine particulate matters
and contribute to regional haze (Ye et al., 2011; Meng et al., 2014; Wei et al., 2015). Global ammonia emission has more than doubled since pre-industrial times, mainly because of agricultural intensification (Galloway et al., 2003). The total ammonia emission in China in 2006 was estimated to be 16.07 million tons (Mt) (Dong et al., 2010). Such high emission makes $NH_3$ one of the key species related to atmospheric environmental problems. Some studies have indicated that reducing $NH_3$
concentrations could be an effective method for alleviating secondary inorganic $PM_{2.5}$ pollution in China (Cao et al., 2009; Park et al., 2014; Wang et al., 2015; Xu et al., 2017).

As global food production requirements increase, agriculture plays an increasingly important role in local, regional, and global air quality (Walker et al., 2006). The North China Plain (NCP) is a highly populated region with intensive agricultural production as well as heavy industry. The region has been
affected by severe haze and photochemical pollution in recent years (Guo et al., 2010; Wang et al., 2010; Luo et al., 2013). Covering only 3.3% of the national area, the NCP region provides 40% and 25% of China's wheat and corn production. To sustain such high agricultural productivity, chemical fertilisers have been intensively applied. Less than 30% efficiency in N application causes approximately 40% N loss through various routes including the leaching of $NO_3^-$ and emission of $NH_3$,

N$_2$O, and N$_2$ (Zhang et al., 2010). So far, only a few limited studies have paid attentions to impacts of NH$_3$ on air pollution in the NCP region. According to some studies (Dong et al., 2010; Ianniello et al., 2010; Meng et al., 2011; Shen et al., 2011; Meng et al., 2015), the high NH$_3$ emission intensities observed in the NCP have been caused by high fertiliser application rates and numerous intensive

livestock farms. However there were few simultaneous high time resolution measurements of NH$_3$ and NH$_4^+$ in PM$_{2.5}$, and investigating the role in fine particulate formation in China. These studies are necessary to improve our understanding of ammonia pollution on regional air quality, and of the impact on formation of secondary ammonium aerosols in the NCP.

During May-September 2013, the intensive field measurements of NH$_3$ and other trace gases,

water-soluble ions in PM$_{2.5}$, and meteorological parameters took place at a rural site in the NCP. In this article, we report the results on NH$_3$, trace gases and major water-soluble ions in PM$_{2.5}$. We discuss temporal variations and diurnal patterns of NH$_3$ and NH$_4^+$. We also show results from thermodynamic equilibrium simulations and compared them with observations.

**2 Description of Experiment**

**2.1 Measurement site**

The measurements were performed from May to September 2013 at Gucheng (39 °08′N, 115 °40′E, 15.2 m a.s.l.), a rural site in the NCP, which is an Integrated Ecological Meteorological Observation and Experiment Station of the Chinese Academy of Meteorological Sciences. In Fig. 1, the location of the site is shown on the NCP map with the NH$_3$ emission distribution for the year 2012 from the

multi-resolution emission inventory of China (http://meicmodel.org/index.html). The measurement site chosen is situated for monitoring regional background concentrations of air pollutants in the North China Plain, has good regional representativeness (Lin et al., 2009). The site is approximately 110 km southwest of Beijing, 130 km west of Tianjin, and 160 km northeast of Shijiazhuang City in Hebei Province. The site is surrounded by farms, dense villages/towns, and the transportation network in the

NCP. The main crops in the area surrounding the site are wheat (winter and spring) and corn (summer and fall). The site is influenced by high NH$_3$ emissions from fertiliser use and animal husbandry in the surrounding area. Being in the warm temperate zone, the site has a typical temperate continental monsoon climate. Precipitation occurs mainly between May to August.

**2.2 Sampling and analysis**

Ambient NH$_3$ was measured using an ammonia analyser (DLT-100, Los Gatos Research, USA), which

utilize a unique laser absorption technology called Off-Axis Integrated Cavity Output Spectroscopy (OA-ICOS). The analyzer has a precision of 0.2 ppb at 100 sec average and a maximum drift of 0.2 ppb over 24 hrs. The response time of the analyzer is less than 2 s (with optional external N920 vacuum pump). During the campaign, $NH_3$ data were recorded as 100-s average. In principle, the $NH_3$ analyzer does not need external calibration, because the measured fractional absorption of light at an ammonia resonant wavelength is an absolute measurement of the ammonia density in the cell (Manual of Economical Ammonia Analyzer - Benchtop Model 908-0016, Los Gatos Research). However, we confirmed the good performance of the $NH_3$ analyzer using a reference gas mixture $NH_3/N_2$ (Scottgas, USA) traceable to US National Institute for Standards and Technology (NIST). The reference gas of $NH_3$ (25.92 ppm with an accuracy of $\pm2\%$) was diluted to different concentrations using zero air and supplied to the analyzer and a sequence with 5 points of different $NH_3$ concentrations (including zero) were repeated for several times to check the performance of the analyzer.

As shown in Fig. S1, the analyzer followed rapidly to changes of the $NH_3$ concentration, produced stable response under stabilized $NH_3$ concentrations, and repeated accurately (within the uncertainty) the supplied $NH_3$ concentrations. The $NH_3$ analyzer contains an internal inlet aerosol filter, which was cleaned before our campaign. Nevertheless, some very fine particles can deposit on the mirrors of the ICOS cell, leading to gradual decline in reflectivity. However, slight mirror contamination does not cause errors in $NH_3$ measurements because the mirror reflectivity is continually monitored and the measurement is compensated using the mirror ringdown time. Interferences to $NH_3$ measurements can be from the sample inlets, for example, due to water condensation or adsorption/desorption effects (e.g., Schwab, 2008; Norman et al., 2009). Such interferences were not quantified but reduced as possibly as we could. PTFE tubing (4.8 mm ID), which is one of the well suited materials for $NH_3$ measurement (Norman et al., 2009), was used to induced ambient air. The length of the tubing was kept as short as possible (about 5 m) to limit the residue time to less than 3 s. The aerosol filter at the inlet was changed every two weeks. Water condensation was avoided. Nevertheless, we cannot exclude the influence from the adsorption and desorption, which can also occur on dry surfaces. However, this influence should be small at our site, where the $NH_3$ concentration is very high, and cause mainly a lag in the recorded $NH_3$ concentration.

A set of commercial instruments from Thermo Environmental Instruments, Inc. were used to measure $O_3$ (TE 49C), $NO/NO_2/NO_x$ (TE 42CTL), CO (TE 48C), and $SO_2$ (TE 43CTL). All

instruments were housed in an air-conditioned room in the observation building at the site. Two parallel inlet tubes (Teflon, 4.8 mm ID×8 m length) were shared by the analyzers. The height of the inlets was 1.8 m above the roof of the building and about 8 m above the ground. The inlet residence time was estimated to be less than 5 s (Lin et al., 2009). Zero and span checks were performed weekly on the analyzers of these trace gases to identify possible analyser malfunctions and zero drifts. Multipoint calibrations of $SO_2$, $NO_x$, CO and $O_3$ analysers were performed on the instruments at approximately 1-month intervals. Measurement records were saved as 1-min averages. After the correction of data on the basis of the multipoint calibrations, hourly average data were calculated and used for the analysis.

An Ambient Ion Monitor (AIM) (URG 9000D Series, USA) was deployed at the site to measure hourly concentrations of water-soluble inorganic components in $PM_{2.5}$ during 15 June-11 August, 2013. A detailed description of performance evaluation of AIM-IC system is reported by Han et al. (2016). Briefly, ambient air was introduced in to the AIM with a 2 meter Teflon coated aluminum pipe and particles larger than 2.5 μm were removed by a cyclone at a flow rate of 3 L min$^{-1}$. A liquid diffusion denuder was used to remove the interfering acidic and basic gases, in combination with a Steam-Jet Aerosol Collector followed by an Aerosol Sample Collector, until the particles can be injected into the ion chromatograph (Hu et al., 2014). The detection limit of $NH_4^+$, $SO_4^{2-}$ and $NO_3^-$ were 0.05 μg m$^{-3}$, 0.04 μg m$^{-3}$ and 0.05 μg m$^{-3}$, respectively. For the AIM, multipoint calibrations were performed weekly by using calibration standard solutions. Acceptable linearity of ions was obtained with an $R^2$ of ≥0.999. The flow rate of the AIM was checked weekly at the sample inlet with a certified flow meter. The flow rate of the AIM was kept at 3 L min$^{-1}$ with standard derivation of <1%. Hourly data were obtained for the concentrations of water-soluble inorganic ions in summer 2013.

Meteorological parameters were measured at the site. Air temperature and relative humidity were monitored using a humidity and temperature probe (HMP155, Vaisala, Finland); wind speed and direction were measured using an anemometer (ZQZ-TFD12, Jiangsu Radio Scientific Institute Co., Ltd, China); rainfall was measured using a tilting rain gauge (SL2-1, Tianjin Meteorological Instrument Factory, China). Global radiation observation was made at the site but showed a drift by the end of July 2013. Instead we use the photolysis rate $j$NO$_2$ observed using a 2-pi-actinic-flux spectrograph (CCD type, Meteorologie Consult GmbH, Germany) to indicate radiation condition for photochemistry. Hourly meteorological data were calculated from the in-situ measurements and used in this paper. Planetary boundary layer height (PBLH) values at 14:00 were derived from the ERA-Interim data using

the Bulk Ricardson number method (Guo et al., 2016; Miao et al., 2017).

**2.3 Data analysis**

**2.3.1 Chemical conversions of species**

Sulfate and nitrate oxidation ratios (SOR and NOR) are defined as the molar ratio of $SO_4^{2-}$ and $NO_3^-$ in

PM$_{2.5}$ to the total oxidized S and N, respectively (Zhang et al., 2011).

$$SOR = \frac{SO_4^{2-}}{SO_4^{2-} + SO_2} \quad (1)$$

$$NOR = \frac{NO_3^-}{NO_3^- + NO_x} \quad (2)$$

Similarly, the conversion ratio of ammonium (NHR) is expressed in terms of the ratio of ammonium to total ammonia (NH$_x$), which could be a measure of the extent of transformation from NH$_3$ to NH$_4^+$ in areas with major local NH$_3$ sources (Hu et al., 2014).

$$NHR = \frac{NH_4^+}{NH_4^+ + NH_3} \quad (3)$$

**2.3.2 Thermodynamic equilibrium**

Thermodynamic gas-aerosol equilibrium characteristics during summer 2013 were examined using ISORROPIA II model (Fountoukis and Nenes, 2007; Fountoukis, etal., 2009). ISORROPIA II is a thermodynamic equilibrium model for inorganic gases and aerosols in the atmosphere (available at http://isorropia.eas.gatech.edu/index.php?title=Main_Page). The concentrations of the measured NH$_3$ and water-soluble ions in PM$_{2.5}$ were input into the model as total (gas + aerosol) concentrations, along with simultaneously measured relative humidity and temperature data. ISORROPIA II was run in the forward mode with metastable aerosol state salts precipitate once the aqueous phase becomes saturated with respect to salts, which often showed better performance than the stable state solution (solid + liquid) and was commonly applied in previous pH predictions (Guo et al., 2015; Bougiatioti et al., 2016; Liu et al., 2017). In this study, the aerosol properties as acidity and the water content of the aerosol are needed to investigate the aerosol acidity characteristics and role of heterogeneous chemistry in nitrate formation. The pH of aerosol water was calculated using the following equation:

$$pH = -\log_{10} \frac{1000 H^+{}_{air}}{AWC} \quad (4)$$

where $H^+{}_{air}$ (µg m$^{-3}$) is the ion concentration of equilibrium particle hydronium and AWC (µg m$^{-3}$) is

the aerosol water content from the ISORROPIA-II simulation. The evaluation of AWC prediction showed a good performance compared with observed particle water (Bian et al., 2014; Guo et al., 2015).

### 2.3.3 Trajectory calculation

The 72-h backward trajectories were calculated using the HYSPLIT 4.9 model (http://www.arl.noaa.gov/ready/hysplit4.html). The trajectories terminated at the height of 100 m above the ground. The trajectory calculations were done for four times (00:00, 06:00, 12:00, and 18:00 UTC) per day in summer 2013. Individual back trajectories were grouped into five clusters. The number of clusters is identified according to the changes of total spatial variance (TSV). Five is chosen as the final

number of clusters considering optimum separation of trajectories (larger number of clusters) and simplicity of display (lower number of cluster). The corresponding concentrations of trace gases and water soluble ions were averaged over the period of 3 h ahead and after the arrival time for each backward trajectory for further analysis.

### 3 Results and discussion

**3.1 Overview of concentration levels of measured species**

During 15 May-25 September 2013, the average concentrations (ranges) of $NH_3$, $SO_2$ and $NO_x$ were 36.2 (0.1-862.9), 5.0 (0-86.8) and 15.4 (2.7-67.7) ppb, respectively. As listed in Table 1, the concentration of $NH_3$ at the NCP rural site was lower than those reported in Asian and Africa urban sites such as Lahore (Pakistan) (Biswas et al., 2008), Colonelganj (India) (Behera et al., 2010) and

Cairo (Egypt) (Hassan et al., 2013), but higher than those from other areas in China, Europe and North American (Plessow et al., 2005; Yao et al., 2006; Lin et al., 2006; Walker et al., 2006; Hu et al., 2008; Meng et al., 2011; Shen et al., 2011; Schaap et al., 2011; Makkonen et al., 2012; Behera et al., 2013; Gong et al., 2013; Meng et al., 2014; Li et al., 2014). For example, the $NH_3$ at the NCP rural site was higher than those found at Shangdianzi regional background station in the NCP (Meng et al., 2011),

Lin'an regional background station in the Yangtze River Delta (YRD) in Eastern China (Meng et al., 2014) and the rural site in Beijing (Shen et al., 2011). The relatively high concentrations of $NH_3$ observed in this study were attributed to agricultural activities involving fertiliser use, vegetation, and livestock, as well as human excrement and waste disposal in the surrounding region.

According to an inventory study (Zhang et al., 2010), the total agricultural $NH_3$–N emission in 2004

in the NCP was 3071 kt $yr^{-1}$, accounting for 27% of the total emissions in China with the 1620 kt $yr^{-1}$

of $NH_3$–N emissions caused by fertiliser applications, which is the largest emission source accounting for more than half of the total agricultural emissions. In recent years, there were a few publications about China's national and regional emission inventories of $NH_3$ (e.g., Zhou et al., 2015; Xu et al., 2015, 2016; Kang et al., 2016). However, these inventories are based on bottom-up studies, subject to substantial uncertainties in spatial and temporal variations of $NH_3$ emissions. Ground based observations of $NH_3$ have been sparse. Our measurements, together with others, can be used for validating and constraining models that use bottom-up inventories, and hence help to reveal potential bias in $NH_3$ emission inventory.

The observed concentration of $SO_2$ at the NCP rural site was markedly lower than those reported for the same period in 2006-2007 (Lin et al., 2009). Because of a series of emission reduction measures implemented in recent years, $SO_2$ levels have decreased markedly in the NCP (Lin et al., 2011). The average concentration of $NO_x$ was higher than those at Shangdianzi (Meng et al., 2011) and Lin'an (Meng et al., 2014) regional background stations in the NCP and YRD region of China, which might be due to emission form agricultural activities and motor vehicle sources (Lei and Wuebbles, 2013; Liu et al., 2013) in the NCP, but was lower than those at urban sites in India (Behera et al., 2010) and Egypt (Hassan et al., 2013).

The average concentrations (ranges) of $NH_4^+$, $SO_4^{2-}$, and $NO_3^-$ in $PM_{2.5}$ were 19.8 (1.07-340.6), 20.5 (3.30-116.9) and 11.3 (1.09-109.3) $\mu g\ m^{-3}$, respectively, at the NCP rural site during 15 June-11 August 2013. The average concentration of $NH_4^+$ in $PM_{2.5}$ was higher than those observed at the rural or urban sites in the NCP (Meng et al., 2011), YRD (Meng et al., 2014), Beijing (Shen et al., 2011), Guangzhou (Hu et al., 2008), and Hong Kong (Yao et al., 2006) in China, and comparable to that at urban site in India (Behera et al., 2010). The average concentration of $SO_4^{2-}$ in $PM_{2.5}$ was higher than those at rural sites in the NCP (Meng et al., 2011) and YRD (Meng et al., 2014) in China, but was lower than that observed at rural sites in Guangzhou (Hu et al., 2008) in China, as well as urban sites in India (Behera et al., 2010) and Egypt (Hassan et al., 2013). The average concentration of $NO_3^-$ in $PM_{2.5}$ was higher than those observed at the rural sites in the YRD and Guangzhou (Hu et al., 2008) in China, and lower than those at urban sites in India (Behera et al., 2010) and Pakistan (Biswas et al., 2008). The elevated $NH_3$ and $NH_4^+$ in $PM_{2.5}$ concentrations at the NCP rural site demonstrate severe ammonia and fine particulate ammonium pollution in this area.

**3.2 Ambient ammonia**

### 3.2.1 Temporal variation of NH$_3$

The time series of hourly averages of NH$_3$ and other trace gases together with meteorological parameters during 15 May-25 September 2013 at Gucheng are shown in Fig. 2. NH$_3$ concentrations varied considerably during the observation period, ranging from 0.1 to 862.9 ppb. The Gucheng station has a farmland of 8.67 hectares. The observation period was in the time of the wheat harvest and corn seeding and growing. Corn was sown and fertilized with about 600 kg of fertilizer per hectare in late June. On 20 July corn was additionally fertilized with 225 to 300 kg of urea per hectare. After this fertilization, there was a raining period. The NH$_3$ concentration increased rapidly on the seventh day after the urea application on 20 July, peaking during the 27-30 July period (Fig. 2b). The highest hourly value of NH$_3$ (862.9 ppb) was observed at 04:00 local time on 29 July 2013, with the second highest concentration observed at 06:00 on the same day. The extremely high NH$_3$ concentrations were probably caused by intensified soil emissions after rainfall on 26 July, which enhanced the soil moisture. Precipitation and the resulting soil water dynamics are known to strongly affect urea hydrolysis and subsequent NH$_3$ emissions (Reynold and Wolf, 1987; Aranibar et al., 2004). The general increase in NH$_3$ emissions was observed when soils with high moisture content began to dry because of increased diffusion (Burch et al., 1989). In addition, high temperatures in summer promote NH$_3$ volatilisation from urea and ammonium dibasic phosphate applied to crops.

The monthly concentration of NH$_3$ depends on its source and meteorological conditions. The monthly average values of NH$_3$ were 28.4, 73.9, 26.4, and 13.5 ppb in June, July, August, and September 2013, respectively. In summer, high temperature promotes the emission of NH$_3$ from natural and fertilised soils, as well as vegetation. The concentration of NH$_3$ in July was approximately five times higher than that in September, which was influenced by higher temperature and increased emission rates of local agricultural NH$_3$ sources in July.

SO$_2$ and NO$_x$ are the main precursors of sulfate and nitrate aerosols, and O$_3$ play an important role in atmospheric chemistry because they act as sources of OH radicals through photolysis. The maximum hourly average concentrations of SO$_2$ and NO$_x$ were 86.8 ppb at 00:00 on 21 May and 67.7 ppb at 10:00 on 17 September, respectively. O$_3$ monthly levels were high in June (44.3 ppb) and July (43.7 ppb), with a maximum hourly average value of 149.9 ppb at 15:00 on 25 July 2013.

In contrast to NH$_3$, the highest monthly levels of SO$_2$ (7.0 ppb) and CO (885 ppb) were observed in June, which could be due to the open burning of agricultural waste (straw, cornstalk, and other crops)

after harvest in the surrounding area. Previous studies have shown that the burning of crop residues is a crucial source of trace gases such as $NO_x$ and CO in the NCP during summer (Meng et al., 2009; Lin et al., 2011). The obvious impact of biomass burning was observed during 16-19 June 2013 period. As CO is mainly emitted from anthropogenic sources such as the burning of biomass, the elevated CO concentrations (2529 ppb at 22:00 on 16 and 2488 ppb at 22:00 on 17 June) were observed. During this pollution episode, the average concentrations of $NH_3$, $SO_2$, $NO_x$, $O_3$ and CO were 42.6, 7.69, 18.8, 44.0 and 1092 ppb, respectively, which were about 1.2-1.5 times than the average values for the whole study period. The monthly concentrations of $SO_2$, $NO_x$, and CO in July and August decreased compared to those in June. In addition to less influences from biomass burning, meteorological conditions were also in favor of lowering the concentrations of these gases. Figure S2 shows the monthly average diurnal variations of $jNO_2$ and the time-series of hourly rainfall during June-August 2013. As can be seen, the average $jNO_2$ increased from June to August, indicating better conditions for photochemical reductionin July and August. There was also a slight increase in rainfall from June to August, which may promote removal of the pollutants. For the secondary pollutant $O_3$, the highest concentration was observed in June. This is consistent with previous results from Gucheng (Lin et al., 2009) and should be related with the annual maximum of background $O_3$ in the NCP, which occurs in June (Ding et al., 2008; Lin et al., 2008).

**3.2.2 Diurnal variations of $NH_3$**

The average diurnal variations of $NH_3$ during June to September 2013 are shown in Fig. 3. As indicated in Fig. 3a, $NH_3$ concentration maxima and minima were observed during 08:00-13:00 and 19:00-23:00, respectively. As for July, $NH_3$ concentrations showed a considerably more pronounced diurnal pattern with a maximum of 59.5 ppb at 08:00. The concentration of $NH_3$ gradually increased during 00:00-03:00, remained relatively constant during 04:00-06:00, and then rapidly increased from 06:00 (beginning just after sunrise). After peaking at approximately 08:00, a decrease was observed until it reached the minimum of 29.8 ppb at 19:00.

The morning peak of $NH_3$ was also observed elsewhere and could be resulted from emissions from fertilised soils and plant stomata, evaporation of dew, and human sources, as well as mixing down of ammonia from the residual layer (Trebs et al., 2004; Norman et al., 2009; Bash et al., 2010; Ellis et al., 2011). Figure 3b reveals that the relative humidity (90%-89%) and temperature (21.5-22.1 ℃) remained relatively constant before 06:00, but increased later in the morning. The increasing temperature can

heat the earth's surface and vegetation leaves and reduce the RH, potentially leading to evaporation of $NH_3$ from soil and plants and volatilization of ammonium aerosol (Trebs et al., 2004; Norman et al., 2009; Ellis et al., 2011), which may increase $NH_3$ concentrations in the morning. When the emission was occurring into a shallow boundary layer, $NH_3$ increase would be more prominent. In addition, the morning rise might also be due to the breakup of the nocturnal boundary layer. During the sampling period, the majority of peaks of ammonia over 50 ppb occurred at night, which were attribute to local emissions, such as from agricultural activity, into a shallow nocturnal boundary layer. It was supposed by Ellis et al. (2011) that the downward mixing of air containing higher $NH_3$ from the residual layer could lead to an increase of surface $NH_3$ after the breakup of the nocturnal boundary layer.

From Fig. 3a, it can be seen that in July the $NH_3$ level was the highest and peaked earliest. One reason for this might be the increased emissions of local agricultural $NH_3$ sources in July compared with those in June, August, and September. On the average, the level $NH_3$ in July had a maximum nighttime increase (20.0 ppb from 20:00 to 06:00), which is much large than those in June (5.2 ppb), August (9.9 ppb) and September (1.8 ppb). The early morning increase of $NH_3$ in July started from a much higher level than in other months, resulting a earliest $NH_3$ peak in July.

The Gucheng site is an experiment station for agrometeorological studies. Corn is the main crop in the station area and nearly all the agricultural areas in the surrounding. According the climate in the NCP, corn is planted around the middle of June and grows rapidly in July. Therefore, July is the key period for the application of nitrogen fertilisers like urea. As mentioned above, the urea application in the station on 20 July 2013 and a precipitation process afterwards caused huge $NH_3$ spikes during the end of July (Fig. 2b). In addition, the highest nighttime temperature in July (Fig. 3b) could promote the soil emission of $NH_3$, and the relatively lower wind speed (Fig. 3b) and lower PBLH (Fig. S3) in July was in favor of the accumulation of $NH_3$ in surface air.

In summary, ambient $NH_3$ at Gucheng showed interesting diurnal cycles, which look significantly different in different summer months. We believe the interplay of some processes, such as emissions from agricultural sources, meteorological conditions (temperature, relativity humidity, wind speed, and PBLH, etc.) as well as chemical conversion, are important in the determination of levels and diurnal patterns of $NH_3$ at the site. Whether or not these processes are all important in the morning variation of $NH_3$? How important are they? And what makes the difference in the peaking time and concentration of $NH_3$ in different months? These are questions to be answered in the future.

### 3.3 Ambient ammonium aerosol

Secondary inorganic aerosols form from gas-phase precursors, which are mostly from anthropogenic activities such as industrial, agricultural, and motor vehicle emissions. Therefore, the major precursors ($NH_3$, $SO_2$ and $NO_x$) are responsible for the formation of particulate ammonium, sulphate, and nitrate.

The hourly $NH_4^+$ concentrations during 15 June-11 August 2013 ranged from 1.07 to 340.6 $\mu g\ m^{-3}$, with an average concentration of 19.8 $\mu g\ m^{-3}$. The highest monthly level of $NH_4^+$ appeared in July and lowest level appeared in June 2013. Similar to $NH_3$, the concentration of $NH_4^+$ also increased sharply after urea fertilisation, with the highest value (340.6 $\mu g\ m^{-3}$) observed at 09:00 on 28 July 2013. The temporal variations of $NH_4^+$ basically coincided with $SO_4^{2-}$ and $NO_3^-$ (discussed in Sect. 3.6), reflecting that $NH_4^+$ largely originated from the neutralization between $NH_3$ and acidic species.

The highest hourly $SO_4^{2-}$ concentration (116.9 $\mu g\ m^{-3}$) was observed at 10:00 on 9 July and the second highest value was 111.4 $\mu g\ m^{-3}$ at 18:00 on 6 August, 2013, with an average concentration of 20.5±13.6 $\mu g\ m^{-3}$. Despite the lower concentrations of $SO_2$, higher $SO_4^{2-}$ concentrations in summer were attributed to the higher temperature, $O_3$ concentration and solar radiation, which increase the photochemical activities, the atmospheric oxidation and markedly faster conversion of $SO_2$ to $SO_4^{2-}$. The average concentration of $NO_3^-$ in $PM_{2.5}$ was 11.3 ±9.1 $\mu g\ m^{-3}$. The highest value of 109.3 $\mu g\ m^{-3}$ was observed at 14:00 on 22 June 2013 at the highest RH (93%) and AWC (910 $\mu g\ m^{-3}$). The high RH conditions in summer might dissolve a significant fraction of $HNO_3$ and $NH_3$ in humid particles, therefore increasing the concentrations of $NO_3^-$ and $NH_4^+$ in the atmosphere (Krupa et al., 2003; Trebs et al., 2004; Ianniello et al., 2010). The high $NO_3^-$ concentrations were also mostly associated with large aerosol water content, which indicates the importance of heterogeneous hydrolysis in the production of nitrate (Pathak et al., 2009). On the other hand, $NH_3$ was more efficient in summer to react with $SO_2$ to form $(NH_4)_2SO_4$.

Figure 4 shows correlations of observed $NH_4^+$ with the sum of observed with observed $SO_4^{2-}$, $SO_4^{2-}+NO_3^-$ and $SO_4^{2-}+NO_3^-+Cl^-$. Although all the correlations are significant, the slopes of the regression lines are far from unit. This cannot be due bias in measurements. The major ion balance shows ratio of 1.05:1.0 for cation:anion. The slope is 0.56 when all three strong acid are considered, suggesting that the neutralization of the strong acids explain 56% of the observed $NH_4^+$. In other words, nearly 44% of the observed $NH_4^+$ was due to the presence of other acids in aerosol particles.

### 3.4 Results from thermodynamic equilibrium simulation

We have used the thermodynamic equilibrium model ISORROPIA II to investigate gas-aerosol partitioning characteristics. The model outputs include equilibrium $NO_3^-$, $SO_4^{2-}$, $NH_4^+$, $H^+_{air}$, $HNO_3$, $NH_3$, AWC, etc. As shown in Fig. 5, the modelled $NO_3^-$, $SO_4^{2-}$, $NH_3$ show excellent correlations with the corresponding measurements, but modelled $NH_4^+$ is much worse correlated with the measured one.

Modelled $NO_3^-$, $SO_4^{2-}$, and $NH_3$ values agree well with the measurements, while the modelled $NH_4^+$ largely underestimate the measurements. Considering the unbalance between observed $NH_4^+$ and the sum of observed $SO_4^{2-}+NO_3^-+Cl^-$, we can confirm that other acids in aerosol particles are important in the conversion of $NH_3$ to $NH_4^+$. These other acids may be oxalic acid and other dicarboxylic acids. Although we did not measure organic acids in aerosol, the presence of oxalic acid and other low

molecular weight dicarboxylic acids in aerosols is often reported (e.g., Hsieh et al., 2007; Kawamura et al., 2010, 2013; Sauerwein and Chan, 2017). There is no doubt about the presence of significant amount of dicarboxylic acids over the North China Plain particularly during summer (Kawamura et al., 2013). Therefore, it is highly possible that neutralizing dicarboxylic acids in aerosol particles contributed significantly to the conversion of ammonia to ammonium.

The average concentration of simulated $HNO_3$ was 0.7 μg m$^{-3}$, showing a maximum value of 7.41 μg m$^{-3}$ at 11:00 on 19 June 2013. The average diurnal variations of $HNO_3$ and $H^+_{air}$ are shown in Fig. Fig. 6a. A typical high $HNO_3$ concentrations during daytime and low values at nighttime during the observation period is predicted by the model, which is consistent with other studies (Makkonen et al., 2012; Sudheer et al., 2015). The diurnal cycle of $H^+_{air}$ is predicted with the highest level around 17:00.

The concentrations of $NH_3$ were closely associated with $H^+_{air}$, and higher $NH_3$ always corresponded to lower $H^+_{air}$ (Liu et al., 2017). The pH values of aerosol water, estimated based on the simulated results using equation (4), are mostly in the range of 2.5-4.5. The fine particles were moderately acidic in summer, with an average pH values of 3.5. On average, pH is over 3.5 during nighttime and below 3.5 during daytime (Fig. 6b). Under the medium acidic conditions and high $NH_3$ concentrations, organic

acid like diacids are able to reaction with ammonia to form ammonium. Because we used ISORROPIA-II for inorganic aerosol composition and no organic acids measurements are available, we cannot analyze in detail the role of organic acids though the model performed quite well (Fig. S4).

**3.5 Relationship between ammonia and ammonium aerosol**

The gas-to-particle conversion between $NH_3$ and $NH_4^+$ has been reported to be strongly affected by

temperature, RH, radiation conditions, the concentration of primary acid gas, and other factors. In this

study, $NH_4^+$ concentrations correlated significantly and positively with $NH_3$ with a correlation coefficient of 0.78 and a slope of 1.48 (Fig.7a, n=915, P < 0.01), suggesting that $NH_3$ played a important precursor role in $NH_4^+$ in $PM_{2.5}$ formation.

The ratio of $NH_3$ to $NH_x$ ($NH_3+NH_4^+$) has been used to identify the source of $NH_x$ and the relative contribution of $NH_3$ to $NH_x$ deposition (Lefer et al., 1999; Walker et al., 2004). A value higher than 0.5 signifies that $NH_x$ is mainly from local $NH_3$ sources and that the dry deposition of $NH_3$ dominates the $NH_x$ deposition. Robarge et al. (2002) reported that more than 70% of $NH_x$ was in the form of $NH_3$ at an agricultural site in the South-eastern United States, and concluded that given a larger deposition velocity of ammonia compared with that of ammonium, a considerable fraction of $NH_x$ could be deposited locally rather than be transported out of the region. According to hourly average concentrations, the ratio of $NH_3/NH_x$ varied from 0.22 to 0.97, with a mean ratio of 0.69±0.14, suggesting that $NH_3$ remained predominantly in the gas phase rather than the aerosol phase in summer 2013 at Gucheng.

The diurnal changes in gaseous precursors and aerosol species are controlled by emission and deposition processes, horizontal and vertical transport and gas-particle partitioning. To investigate gas-particle conversion further, diurnal variation of $NH_x$ was showed in Fig. 7b. Between 08:00-18:00, a decrease in $NH_3$ would result in an increase in $NH_4^+$, which coincided with higher sulfate concentrations. The decrease in gas phase ammonia is likely the result of uptake onto aerosols to from $(NH_4)_2SO_4$. The diurnal variability of $NH_x$ may be controlled by transport and vertical exchange. Between the hours of 08:00-18:00, $NH_3$ decreased by 43% while $NH_x$ decreased by 49%, suggested that $NH_3$ remained predominantly in the gas phase. Between 19:00 and 07:00, $NH_3$ increased by 42% and $NH_x$ increased by 51%, indicating that gas-particle partitioning contributes significantly to the decrease in gas phase ammonia during this time.

### 3.5.1 Gas- to-particle conversion ratio of $NH_3$

Sulphate and nitrate oxidation ratios (SOR and NOR) are defined in literature to investigate $SO_4^{2-}$ and $NO_3^-$ formation and gas-particle transformation (Zhang et al., 2011). The average values of SOR and NOR were estimated to be 0.64 and 0.24 during the observation period at Gucheng, with SOR and NOR being higher than previous measurements (Zhou et al., 2009; Du et al., 2011; Zhang et al., 2011). Yao et al. (2002) pointed out an SOR lower than 0.10 under conditions of primary source emissions and higher than 0.10 when sulphate was mainly produced through the secondary transformation of $SO_2$

oxidation. The value of SOR reached to 0.70 in August 2013, which may due to the enhanced atmospheric oxidant levels, sufficient ammonia for neutralization, and higher RH in summer at Gucheng (Tang et al., 2016).

To gain further insights into the transformation of $NH_3$ to $NH_4^+$, the conversion ratio of ammonium (NHR) was investigated. NHR is a measure of the extent of transformation from $NH_3$ to $NH_4^+$ in areas with significant local $NH_3$ sources, although it encompasses both transport and local equilibrium, the latter dominating further downwind from the source. In this study, the average hourly values of NHR ranged from 0.03 to 0.77, with an average of 0.30 during summer 2013. The average NHR level in this study was higher than that observed at an urban site in Beijing (Meng et al., 2017), indicating that high $NH_3$ concentrations resulting from agricultural activities had a marked influence on the formation of ammonium.

### 3.5.2 Diurnal patterns of NHR, SOR and NOR

Fig. 8 presents the diurnal patterns of NHR, SOR, NOR, gaseous precursors, and major water soluble ions, and meteorological factors. As a key species contributing to the oxidisation capacity of the atmosphere, $O_3$ can promote $HNO_3$ formation, affecting the conversion ratio of $NH_3$. $O_3$ exhibited low levels in the morning and enhanced levels in the late afternoon. The lower morning concentrations may be due to the depositional loss of $O_3$ under stable atmospheric conditions in early morning hours, and the higher levels in the afternoon could be due to the photochemical production of $O_3$. The $NH_4^+$ concentration started to increase from morning, reaching the maximum value (16.1 μg m$^{-3}$) at 18:00 , with a diurnal difference of 3.7 μg m$^{-3}$. This diurnal pattern may be due to a combination of high $NH_3$ concentrations, the intense solar radiation at noon, and the high oxidisation capacity of the atmosphere in the afternoon. A clear diurnal cycle of NHR existed, with an amplitude of 0.10 and a peak of 0.35 at 18:00, which is consistent with the higher SOR and RH.

The $SO_2$ concentration showed a maximum at 09:00, with a secondary peak at 22:00. The concentration of $SO_4^{2-}$ showed small peak at 11:00, 14:00 and 18:00, respectively, but no strong diurnal variation. SOR displayed a diurnal cycle with the highest value of 0.74 observed at 05:00. It is noted that the SOR was lower during daytime when photochemical reaction is intense. Higher SOR during nighttime suggests importance of dark reactions. $SO_2$ is highly soluble and can easily absorbed by wet aerosol particles. The much RH during night may promoted this process.

As for the diurnal cycle of $NO_x$, a peak was observed at 06:00 when the mixing layer was stable,

and a broad valley was observed in the daytime, reflecting the influences of a higher mixing layer and stronger photochemical conversion. During the night, $NO_x$ concentrations increased again, resulting in the second maximum at 23:00. $NO_3^-$ concentrations did not show profound diurnal variations, but slightly higher values during the night time, probably because of the hydrolysis of dinitrogenpentoxide ($N_2O_5$) and the condensation of $HNO_3$ under the relatively low temperature. NOR displayed a diurnal pattern with a maximum of 0.28 at 10:00, which was likely related to photochemical reactions under the conditions of high $O_3$ concentrations and $j$$NO_2$ levels.

Nighttime formation, aerosol uptake and hydrolysis of $N_2O_5$ are highly uncertain as has been pointed out (e.g., Xue et al. 2014). The $NO_x$ concentration during nighttime was higher than during daytime, while the $NO_3^-$ level during nighttime was only slightly higher than that during daytime. By assuming high aerosol surface to mass ratio (33.7 $m^2$ $g^{-1}$, Okuda, 2013) and a high uptake coefficient (0.1, Seinfeld and Pandis, 2006), we estimate the nighttime $N_2O_5$ under the conditions over our site to be in the range of about 3-10 ppb, corresponding to a $HNO_3$ production rate of about 1-3 ppb $hr^{-1}$ (or 2.6-7.7 $\mu g$ $m^{-3}$). This rate of $HNO_3$ production would cause an obvious night production of $NH_4^+$. Indeed we can see increases in the $NH_4^+$ concentration and NHR during night (Fig. 8). However, a more or less accurate estimate of the relative contribution of the night $N_2O_5$ chemistry to $NH_3$ conversion needs to be made in the future.

**3.6 A case study of a pollution period**

On several days during the study period, very high $NH_3$ and inorganic $PM_{2.5}$ concentrations were observed. Here make a case study of a pollution period during 7-11 August 2013. Data of gases, major aerosol ions and some key meteorological parameters are presented in Fig. 9. Some calculated parameters during this period are given in Fig. S5. As shown in Figs. 9 and S5, there was a sharp increase of $NO_x$ during the night and early morning of 10 August, followed by that of $NH_3$ (peak value of 64 ppb) at 03:00. In the meantime, a large peak of AWC occurred and gaseous $HNO_3$ decreased to nearly zero (Fig. S5), suggesting rapid uptake of wet aerosol. This event caused the first largest peak of [$SO_4^{2-}$]+[$NO_3^-$]+[$NH_4^+$]. After this event $NH_3$ rose again and reached a even higher peak (76.3 ppb) shortly before noon of 10 August. This peak of $NH_3$ coincided with a valley of $NO_x$, but the $HNO_3$ level increased and pH value decreased was observed in parallel. A few hours later $SO_2$ showed a large peak and the second largest peak of [$SO_4^{2-}$]+[$NO_3^-$]+[$NH_4^+$] occurred. These data show that high $NH_3$ concentration was accompanied by the large increase in concentrations of $SO_4^{2-}$, $NO_3^-$ and $NH_4^+$,

confirming that $NH_3$ play an important role in PM mass formation and that gas-particle conversion occurred when $NH_3$ was available, though $SO_4^{2-}$ partitions to the aerosol phase regard less of $NH_3$ level (Gong et al., 2013).

The secondary ions concentrations had similar temporal distributions with slow accumulation and relatively rapid clearing under favourable meteorological conditions. There were good correlation between $NH_3$ with $NH_4^+$, $SO_4^{2-}$ and $NO_3^-$ (R=0.33, 0.27 and 0.49, respectively, with P < 0.01). However, there was also situation when high $NH_3$ did not associate with high $[SO_4^{2-}]+[NO_3^-]+[NH_4^+]$, as indicated by the data around noon of 8 August (Fig. 9). During this case, AWC was extremely low and RH was around 40%. These conditions do not favour heterogeneous reactions.

During 7-11 August 2013, the relationships of the observed $NH_4^+$ versus those of $SO_4^{2-}$, the sum of $SO_4^{2-}$ and $NO_3^-$ and the sum of $SO_4^{2-}$, $NO_3^-$ and $Cl^-$ are presented in Fig. 10. It is known that $(NH_4)_2SO_4$ is preferentially formed and the least volatile, $NH_4NO_3$ is relatively volatile, while $NH_4Cl$ is the most volatile. $NH_4^+$ is thought to be first associated with $SO_4^{2-}$, afterwards, the excess of $NH_4^+$ is with nitrate and chloride (Meng et al., 2015). It is noted that the correlation of $NH_4^+$ with the sum of $SO_4^{2-}$ and $NO_3^-$ (R=0.91, slope=1.23, with P < 0.01) was better than that of $NH_4^+$ with $SO_4^{2-}$ (R=0.80, slope=1.65, with P < 0.01), suggesting that both $SO_4^{2-}$ and $NO_3^-$ were associated with $NH_4^+$. As shown in Fig.10, sulfate and nitrate were almost completely neutralized with most of the data above the 1:1 line. A few scattered data below the 1:1 line may be caused by uncertainties in measurements. Little different was found between the regression slopes of $NH_4^+$ with the sum of $SO_4^{2-}$ and $NO_3^-$ and the sum of $SO_4^{2-}$, $NO_3^-$ and $Cl^-$ due to the very low amount of $NH_4Cl$. In this study, the level of $NH_3$ was high enough to neutralize both $SO_4^{2-}$ and $NO_3^-$, and likely to be form $(NH_4)_2SO_4$ and $NH_4NO_3$. In addition to these substances, it is likely that $NH_3$ also reacted with oxalic acid and other dicarboxylic acid to form ammonium oxalate and other organic ammonium aerosols, as discussed above.

### 3.7 Long rang transport and local ammonia and ammonium

The Gucheng site is located in the densely populated rural area in the NCP, it is influenced by local sources in the surrounding areas and by long range transport of pollutants from the residential and industrial centers around it. Dependence of the concentrations of $NH_3$ on wind direction at Gucheng is studied to get insight into the distribution of local emission sources around the monitoring site. As shown in Fig. 11, during the sampling period, the prevailing surface winds at Gucheng were northeasterly and southwesterly. High $NH_3$ originated from the southwest sector of the measurement

site, which may be due to a local unidentified agricultural or industrial source or transport from the Xushui township, which is approximately 15 km away from Gucheng. Lower $NH_3$ concentrations were observed under winds from other sectors. Since $NH_3$ is either readily converted to $NH_4^+$ or subjected to dry deposition, high concentrations are found only close to the surface and near the emission sources.

Previous studies have reported an inverse relationship between ground-level concentrations of trace gases, such as ammonia, and wind speed (Robarge et al., 2002; Lin et al., 2011; Meng et al., 2011). Thus, $NH_3$ concentrations might be generally lower at higher wind speeds because of turbulent diffusion.

To identify the impact of long-range air transport on the surface air pollutants levels and secondary

ions at Gucheng, the 72-h backward trajectories were calculated using the HYSPLIT 4.9 model.

As can be seen in Fig. 12, the Clusters 1, 2 and 3 represent relatively low and slow moving air parcels, with cluster 2 coming from northwest areas at the lowest transport height among the five clusters. The air mass of Cluster 1 and 3 originate from southeast of Gucheng. The Cluster 4 and 5 represent air parcels mainly from the far northwest.

The trajectories in Clusters 2 came from the local areas around Gucheng, and it was the most important cluster to the Gucheng site, contributing 56% to the air masses. Based on the statistics, the number of trajectories in Cluster 1, 2 and 3 accounts to 88% of the all trajectories. As more than 80% air masses originated from or passing over the North China Plain region can influence the surface measurements at Gucheng, the observation results at Gucheng can well represent the regional situation

of atmospheric components in the North China Plain region.

Since the emission sources of pollutants are unevenly distributed in the areas surrounding the Gucheng site, air masses from different directions containing different levels of pollutants. The corresponding mean concentrations of $NH_3$, $SO_2$, $NO_x$, $NH_4^+$, $SO_4^{2-}$ and $NO_3^-$ in $PM_{2.5}$ in different clusters of backward trajectories are also included in Table 2 in order to characterize the dependences

of the pollutants concentrations on air masses.

Large differences in the concentrations of $NH_3$, $SO_2$, $NO_x$, $NH_4^+$, $SO_4^{2-}$ and $NO_3^-$ in $PM_{2.5}$ existed among the different clusters, with cluster 2 corresponding to the highest $NH_3$ (48.9 ppb ) and second highest $NO_x$, $NH_4^+$ and $SO_4^{2-}$ (14.4 ppb, 17.5 μg m$^{-3}$ and 22.1 μg m$^{-3}$, respectively).

The cluster 1 corresponds to highest $SO_2$, $NH_4^+$, $SO_4^{2-}$ and $NO_3^-$ (7.9 ppb, 22.3 μg m$^{-3}$, 22.6 μg m$^{-3}$

and 17.7 μg m$^{-3}$, respectively), the second highest $NH_3$ level (32.8 ppb). The cluster 3 had the highest

NO$_x$ level (15.1 ppb), the second highest SO$_2$ and NO$_3^-$ (4.8 ppb and 11.8 μg m$^{-3}$, respectively), and had the third highest concentration of NH$_3$, NH$_4^+$ and SO$_4^{2-}$ levels (28.5 ppb, 14.6 μg m$^{-3}$, and 20.2 μg m$^{-3}$, respectively).

Based on table 2, the lowest NH$_3$, SO$_2$, NO$_x$, NH$_4^+$, SO$_4^{2-}$ and NO$_3^-$ levels were corresponding to clusters 5, which was expected to bring cleaner air masses into surface. As demonstrated by backward trajectory, more than half of the air masses during the sampling period from North China Plain region contributed to the atmospheric NH$_3$ variations, and both regional sources and long-distance transport from southeast played important roles in the observed ammonium aerosol at the rural site in the NCP.

**4 Conclusions**

Online measurements of NH$_3$, trace gases, and water-soluble ions in PM$_{2.5}$ were conducted during May-September 2013 at a rural site in the NCP, where a large amount of ammonia was emitted because of agricultural activities. The average concentrations (ranges) of NH$_3$ and NH$_4^+$ in PM$_{2.5}$ were 36.2 (0.1-862.9) ppb during 15 May-25 September, 2013, and 19.8 (1.07-340.6) μg m$^{-3}$ during 15 June-11 August, 2013, respectively; these are considerably higher than those reported at other sites in China, Europe and North American. Extremely high NH$_3$ and NH$_4^+$ concentrations were observed, which was attributed to high soil moisture level due to rainfall on these days following the urea application. Elevated NH$_3$ levels coincided with elevated NH$_4^+$, indicating the contribution of atmospheric NH$_3$ to secondary inorganic aerosols during periods of agricultural activity. NH$_3$ contributed 69% to the total NH$_3$+NH$_4^+$ in summer, suggesting that NH$_x$ remained predominantly in the gas phase rather than the aerosol phase in summer 2013 at Gucheng.

The average conversion/oxidation ratio for NH$_4^+$ (NHR), SO$_4^{2-}$ (SOR), and NO$_3^-$ (NOR) were estimated to be 0.30, 0.64, and 0.24 in summer 2013, respectively. Results reveal that the concentrations of NH$_3$, NH$_4^+$, and NHR had clear diurnal variations during the observation period. High NH$_3$ and NH$_4^+$ were observed during late night and early morning period. NHR also showed higher values during night, suggesting the importance of heterogeneous reactions driven by high nighttime RH. The hourly data obtained were used to investigate gas-aerosol partitioning characteristics using the thermodynamic equilibrium model ISORROPIA II. Modelled NO$_3^-$, SO$_4^{2-}$, and NH$_3$ values agree well with the measurements, while the modelled NH$_4^+$ largely underestimate the measurements. Our measurement and modelling results indicate that the strong acids in aerosol particles over the rural site were well neutralized by NH$_3$. Nearly a half of the ammonium was not associated with strong acids

but probably with oxalic acid and other diacids, which may present under the medium aerosol acidity (pH around 3.5).

The back trajectory analysis indicates that the transport from the North China Plain region contributed for 56% of air mass with high $NH_3$ levels, meanwhile the long-distance transport from southeast accounted for 32% of air mass with high $NH_4^+$, $SO_4^{2-}$ and $NO_3^-$ at the rural site in the NCP.

$NH_3$ is currently not included in China's emission control policies of air pollution precursors though people have been discussing the necessity for years. Our findings highlight the important role of $NH_3$ in the participation of secondary inorganic and organic aerosol formation. As the emission and concentration of $NH_3$ in the NCP are much higher than needed for aerosol acids neutralization, we speculate that a substantial amount of reduction in $NH_3$ emission is required to see its effect on the alleviation of $PM_{2.5}$ pollution in the NCP. Therefore, further strong reduction of the emissions of primary aerosol, $NH_3$, $SO_2$, $NO_x$, and VOCs is suggested to address the serious occurrence of $PM_{2.5}$ pollution on the North China Plain.

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

**Acknowledgements**

This work was supported by the National Natural Science Foundation of China (2137716, 41330422 and 21777191), the Environmental Protection Public Welfare Scientific Research Project, Ministry of Environmental Protection of the People's Republic of China (grant No. 201509001, 201509002-04 and 201309009), and the China Special Fund for Meteorological Research in the Public Interest (GYHY201206015). The authors would like to thank Professor Xiangde Xu and Yu Song for their helpful support and suggestions. The authors would like Professor Yaqiang Wang for providing $NH_3$ emission data and the plotting helps of Chao He.

**Table Captions**

Table 1. The comparisons of the concentration of trace gases (ppb) and water-soluble ions in $PM_{2.5}$ ($\mu g$ $m^{-3}$) at Gucheng with other researches.

Table 2. Occurrence frequency and mean values of $NH_3$, other trace gases and water-soluble ions in $PM_{2.5}$ for each type of air masses arriving at Gucheng during summer 2013.

**Figure Captions**

Figure 1. Sampling location in the North China Plain with emission distributions of $NH_3$ for the year 2012 from the multi-resolution emission inventory of China (http://meicmodel.org/index.html).

Figure 2. Time series of hourly data of $NH_3$, other trace gases and meteorological parameters measured during the sampling period (a) and a blow-up of the period with extremely high $NH_3$ values during 27-31 July 2013 (b).

Figure 3. Diurnal variation of $NH_3$ (a) and meteorological parameters (b) during the sampling period.

Figure 4. Correlation of observed $NH_4^+$ with observed $SO_4^{2-}$, $SO_4^{2-}+NO_3^-$ and $SO_4^{2-}+NO_3^-+Cl^-$.

Figure 5. Observed and modelled concentrations of $NH_3$, $NH_4^+$, $SO_4^{2-}$ and $NO_3^-$ in summer 2013.

Figure 6. Simulated diurnal variation of $HNO_3$ and $H^+_{air}$ (a) and calculated diurnal variation of pH value of aerosol water (b) in summer 2013.

Figure 7. Relationship between $NH_3$ and $NH_4^+$ (a) and diurnal variation of $NH_x$ (b) in summer 2013.

Figure 8. Diurnal variation of NHR, SOR, NOR, gaseous precursors, major water soluble ions, and meteorological factors in summer 2013.

Figure 9. Hourly concentrations of gaseous, ionic species and $jNO_2$ measured in the pollution episode during 7-11 August 2013.

Figure 10. Correlations between $[NH_4^+]$ and $[SO_4^{2-}]$ (left), $[NH_4^+]$ and $[SO_4^{2-}]+[NO_3^-]$ (middle) and $[NH_4^+]$ and $[SO_4^{2-}]+[NO_3^-]+[Cl^-]$ (right) during 7-11 August 2013.

Figure 11. The average $NH_3$, $NH_4^+$ and meteorological data roses in different wind sectors during summer 2013.

Figure 12. 72-h backward trajectories for 100 m above ground level at Gucheng during sampling period 2013.

Figure S1. Confirmation of the performance of $NH_3$ analyzer using diluted standard gas (mixture $NH_3/N_2$). Instrument response to changed $NH_3$ concentration and stability (a) and repeated multipoint calibrations(b).

Figure S2. Monthly diurnal variations of photolysis rate coefficient of $NO_2$ ($jNO_2$) (a) and hourly amount of precipitation (b) in summer 2013.

Figure S3. The monthly planetary boundary layer heights at 14:00 during 2013 at Gucheng.

Figure S4. Correlation of modelled $NH_4^+$ with modelled $SO_4^{2-}$, $SO_4^{2-}+NO_3^-$ and $SO_4^{2-}+NO_3^- +Cl^-$.

Figure S5. Time series of predicted fine particle pH, particle water mass, $HNO_3$, $H_{air}^+$, $NH_3$ and inorganic ions during 7-11 August 2013.

Table 1. The comparisons of the concentration of trace gases (ppb) and water-soluble ions in $PM_{2.5}$ (µg m$^{-3}$) at Gucheng with other researches.

| Location | Type | Period | NH$_3$ | SO$_2$ | NO$_x$ | NH$_4^+$ | SO$_4^{2-}$ | NO$_3^-$ | Reference |
|---|---|---|---|---|---|---|---|---|---|
| Gucheng, China | Rural | May.-Sept. 2013 | 36.2±56.4 | 5.0±6.5 | 15.4±9.3 | 19.8±33.2 | 20.5±13.6 | 11.3±9.1 | This study |
| Shangdianzi, China | Rural | Jun.2008-Dec.2009 | 10.4±8.1 | 5.9±4.6 | 12.0±6.8 | 7.03±7.76 | 15.0±15.7 | 11.6±11.4 | Meng et al., 2011 |
| Beijing, China | Rural | Aug.2006-Jul.2007 | 21.1±10.5 | – | 37.8±11.6 | 8.8±6.7 | 22.4±16.2 | 15.1±11.4 | Shen et al., 2011 |
| Lin'an, China | Rural | Sept.2009-Dec.2010 | 16.5±11.2 | 6.4±4.2 | 10.8±5.2 | 4.3±3.5 | 9.6±6.1 | 7.3±7.5 | Meng et al., 2014 |
| Guangzhou, China | Rural | Oct.-Nov. 2004 | 10.5 | 21.2 | – | 9.2 | 24.1 | 7.2 | Hu et al., 2008 |
| Hong Kong, China | Urban | Autumn 2000 | 3 | – | – | 2.4 | 9 | 1 | Yao et al., 2006 |
| Taichung, Taiwan | Urban | Jan.-Dec. 2002 | 12.2±4.31 | – | | 4.6±2.0 | 15±8.7 | 6.0±4.0 | Lin et al., 2006 |
| Lahore, Pakistan | Urban | Dec.2005-Feb.2006 | 72.1 | 7.4 | – | 16.1 | 19.2 | 18.9 | Biswas et al., 2008 |
| Colonelganj, India | Urban | Summer 2007 | 41.3±10.5 | 6.95±1.99 | 33.8±8.56 | 18.4±4.7 | 27.8±7.6 | 29.2±7.5 | Behera et al., 2010 |
| Singapore | Urban | Sep.-Nov. 2011 | 3.6 | 8.3 | – | 1.76 | 4.41 | 1.29 | Behera et al., 2013 |
| Oberbärenburg, Germany | Forest | Oct.2001-Apr.2003 | 0.69 | 2.24 | – | 1.55 | 3.07 | 2.22 | Plessow et al., 2005 |
| Netherlands | Rural | Aug. 2007 and 2008 | 12.9 | 0.5 | – | 2.4 | 3.1 | 5.9 | Schaap et al., 2011 |
| Helsinki, Finnish | Urban | Spring 2010 | 0.40±0.59 | 0.29±0.38 | – | 0.46±0.50 | 1.64±1.08 | 1.40±2.04 | Makkonen et al., 2012 |
| Cairo, Egypt | Suburban | Summer 2009 | 64.7 | 5.59 | 28.7 | 7.5 | 28 | 4.2 | Hassan et al., 2013 |
| Clinton, USA | Agricultural | Jan.1999-Dec.2000 | 8 | 1.5 | – | 1.76 | 4.22 | 2.05 | Walker et al., 2006 |
| Houston, USA | Urban | Aug. 2010 | 3.0±2.5 | – | – | 0.5±1.0 | 4.5±4.3 | 0.3±0.2 | Gong et al., 2013 |
| Wyoming, USA | Rural | Dec.2006-Dec.2011 | 0.24 | – | – | 0.26 | 0.48 | 0.32 | Li et al., 2014 |

Table 2. Occurrence frequency and mean values of $NH_3$, other trace gases (ppb) and ionic species in $PM_{2.5}$ ($\mu g\ m^{-3}$) for each type of air masses arriving at Gucheng in summer 2013.

| Air mass | Ratio(%) | $NH_3$ | $SO_2$ | $NO_x$ | $NH_4^+$ | $SO_4^{2-}$ | $NO_3^-$ |
|---|---|---|---|---|---|---|---|
| Clusters 1 | 15 | 32.8 | 7.9 | 14.0 | 22.3 | 22.6 | 17.7 |
| Clusters 2 | 56 | 48.9 | 3.7 | 14.4 | 17.5 | 22.1 | 10.3 |
| Clusters 3 | 17 | 28.5 | 4.8 | 15.1 | 14.6 | 20.2 | 11.8 |
| Clusters 4 | 10 | 23.4 | 2.4 | 12.8 | 12.9 | 15.3 | 7.2 |
| Clusters 5 | 3 | 16.3 | 0.6 | 9.4 | 7.5 | 8.1 | 5.0 |

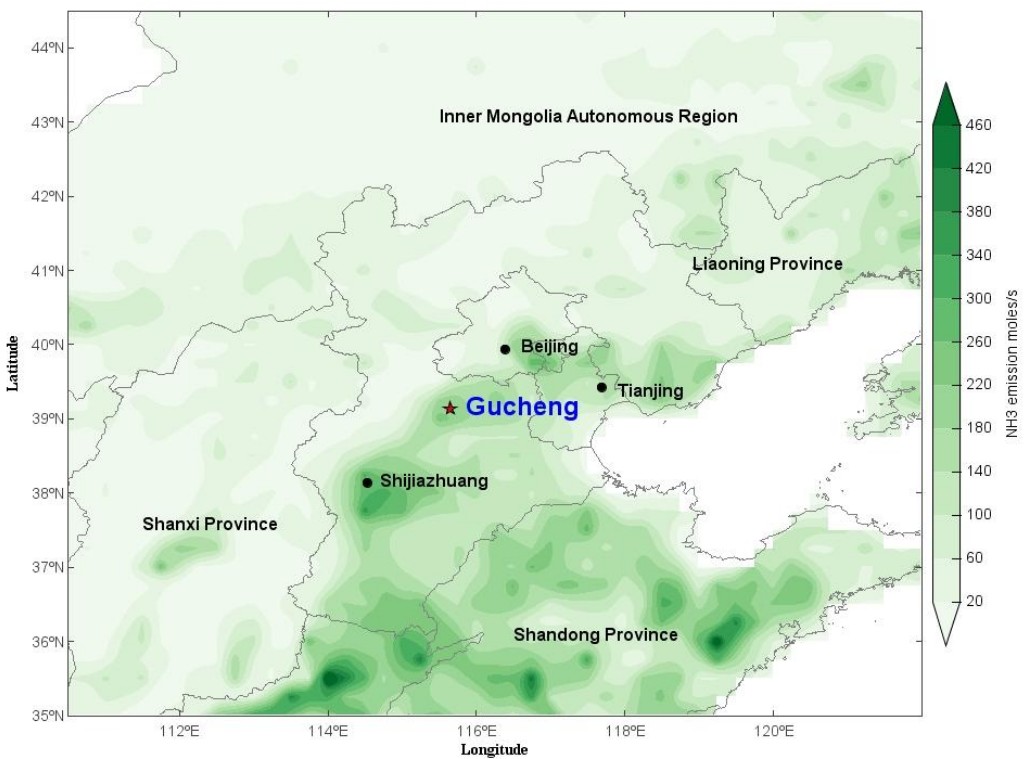

Figure 1. Sampling location in the North China Plain with emission distributions of NH$_3$ for the year 2012 from the multi-resolution emission inventory of China (http://meicmodel.org/index.html).

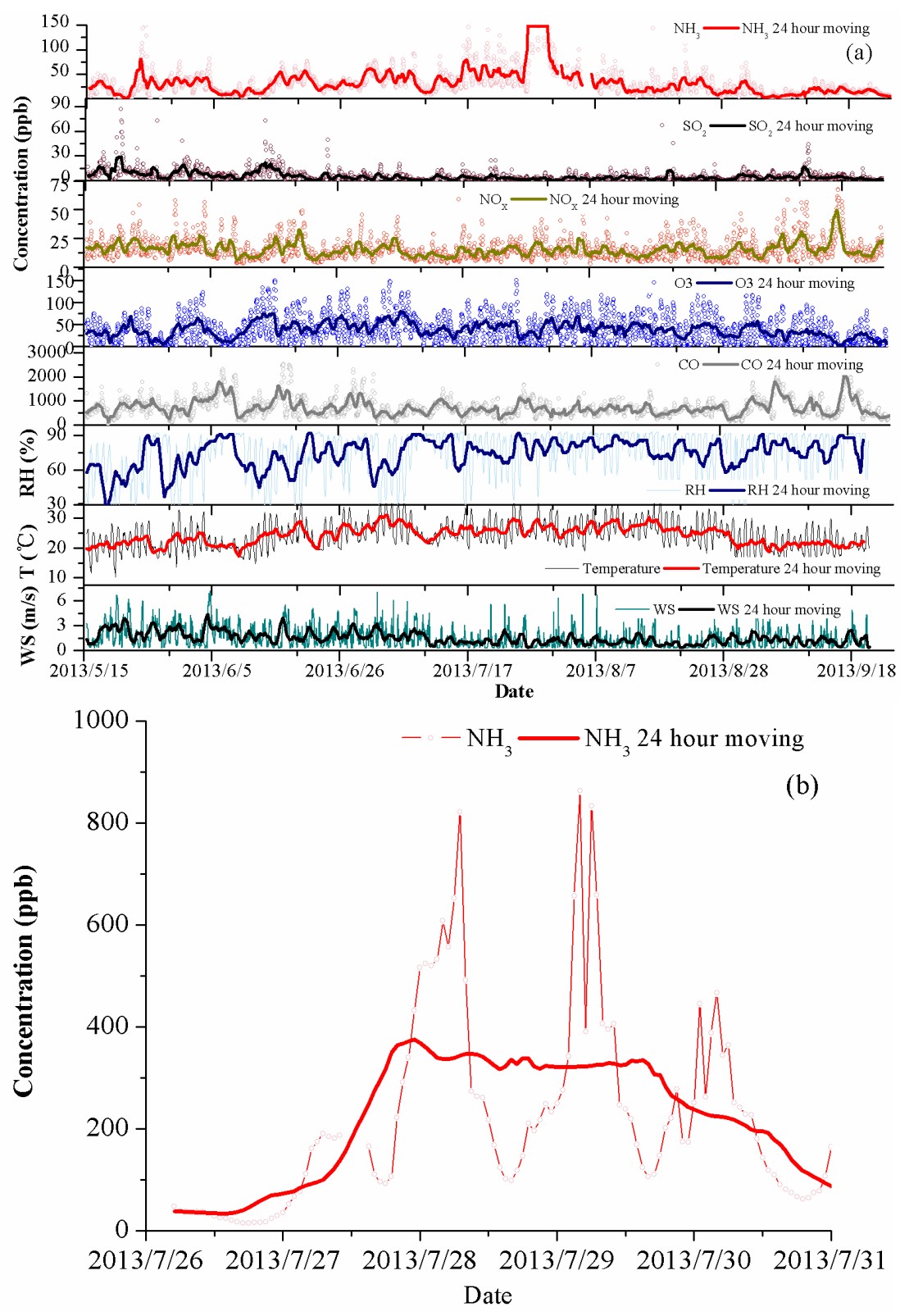

Figure 2. Time series of hourly data of $NH_3$, other trace gases and meteorological parameters measured during the sampling period (a) and a blow-up of the period with extremely high $NH_3$ values during 27-31 July 2013 (b).

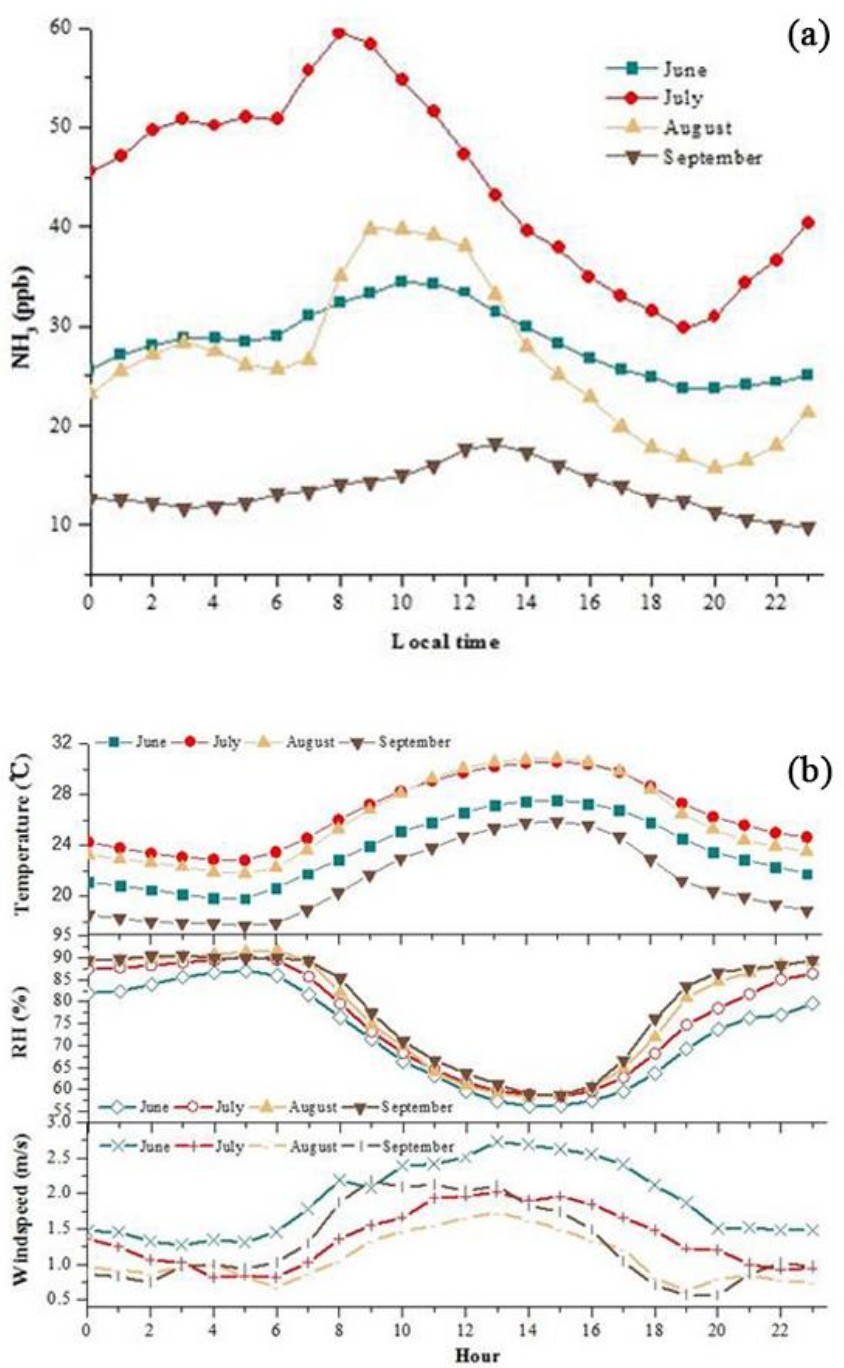

Figure 3. Diurnal variation of NH$_3$ (a) and meteorological parameters (b) during the sampling period.

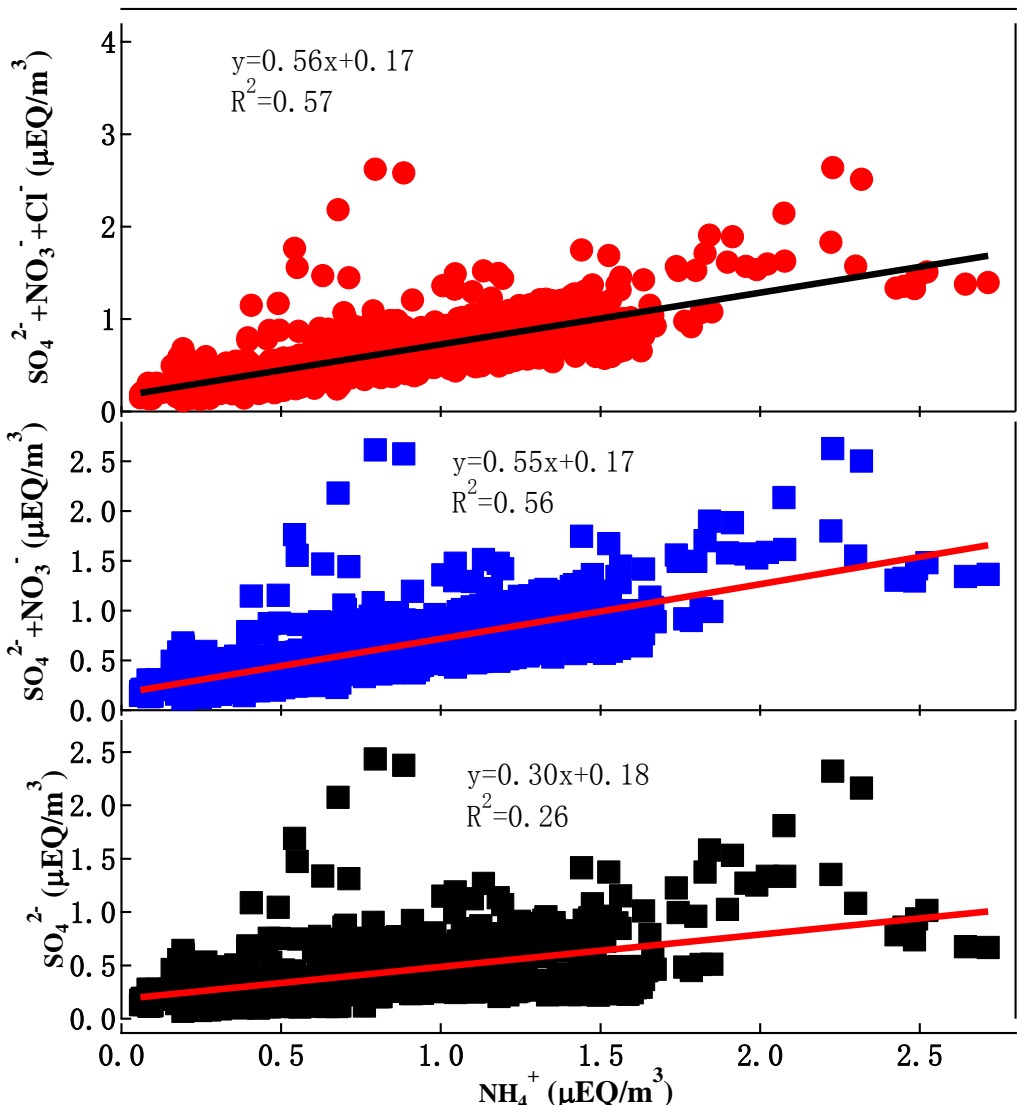

Figure 4. Correlation of observed $NH_4^+$ with observed $SO_4^{2-}$, $SO_4^{2-}+NO_3^-$ and $SO_4^{2-}+NO_3^-+Cl^-$.

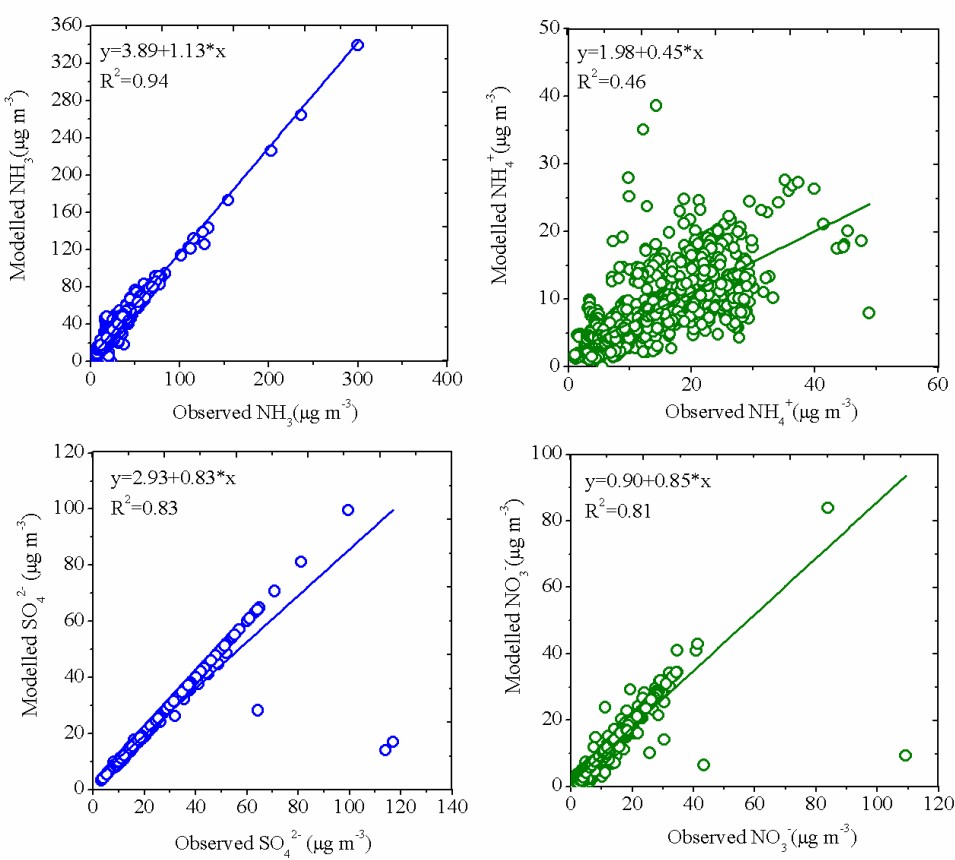

Figure 5. Observed and modelled concentrations of $NH_3$, $NH_4^+$, $SO_4^{2-}$ and $NO_3^-$ in summer 2013.

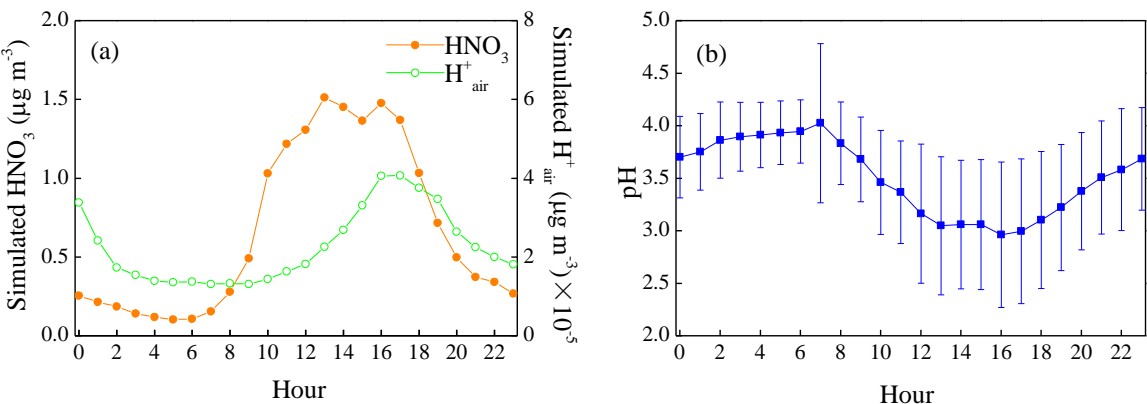

Figure 6. Simulated diurnal variation of HNO$_3$ and H$^+_{air}$ (a) and calculated diurnal variation of pH value
of aerosol water (b) in summer 2013.

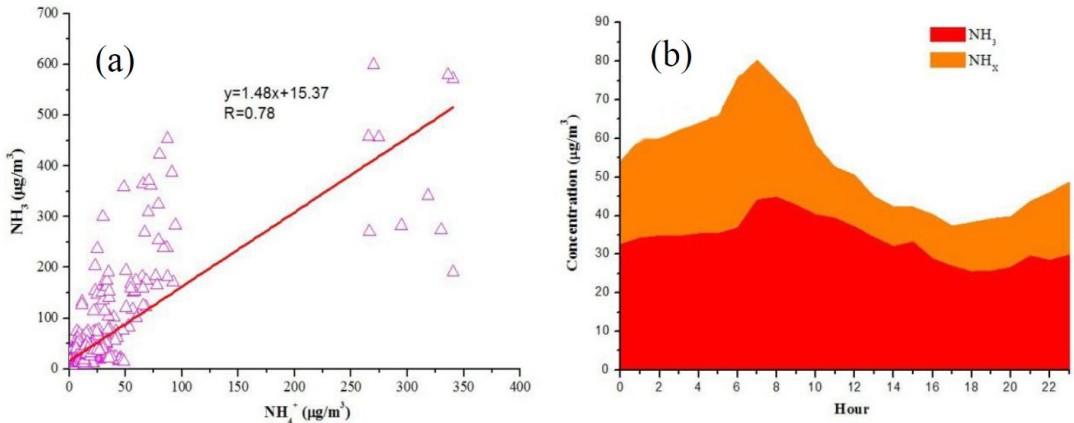

Figure 7. Relationship between NH$_3$ and NH$_4^+$ (a) and diurnal variation of NH$_x$ (b) in summer 2013.

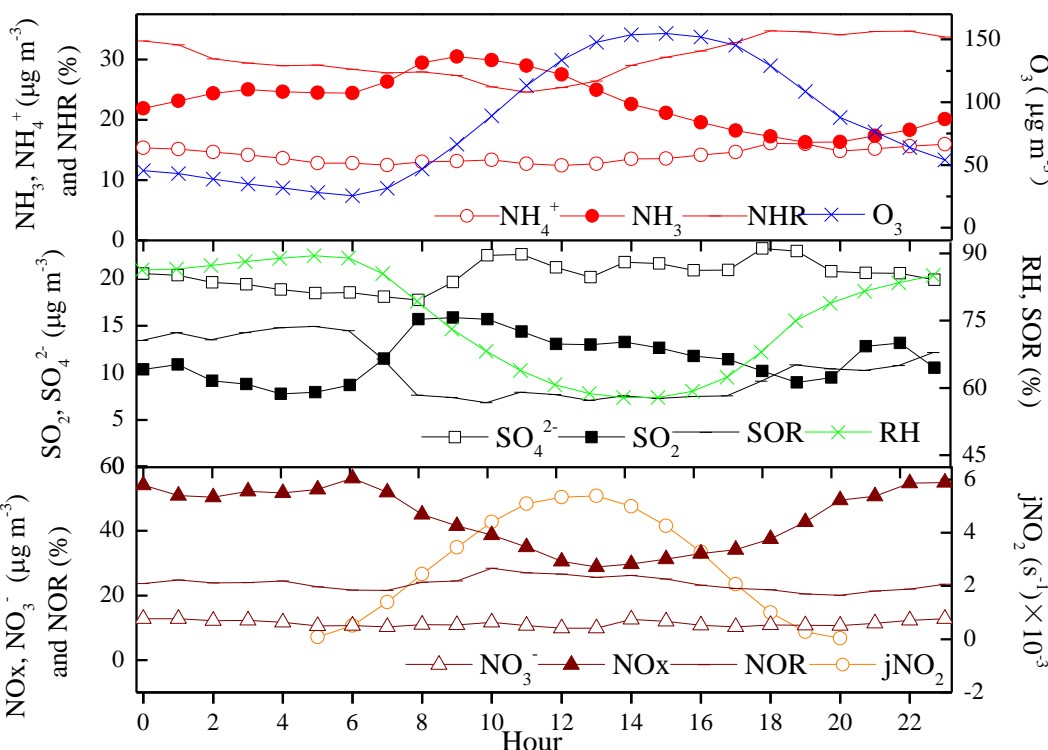

Figure 8. Diurnal variation of NHR, SOR, NOR, gaseous precursors, major water soluble ions, and meteorological factors in summer 2013.

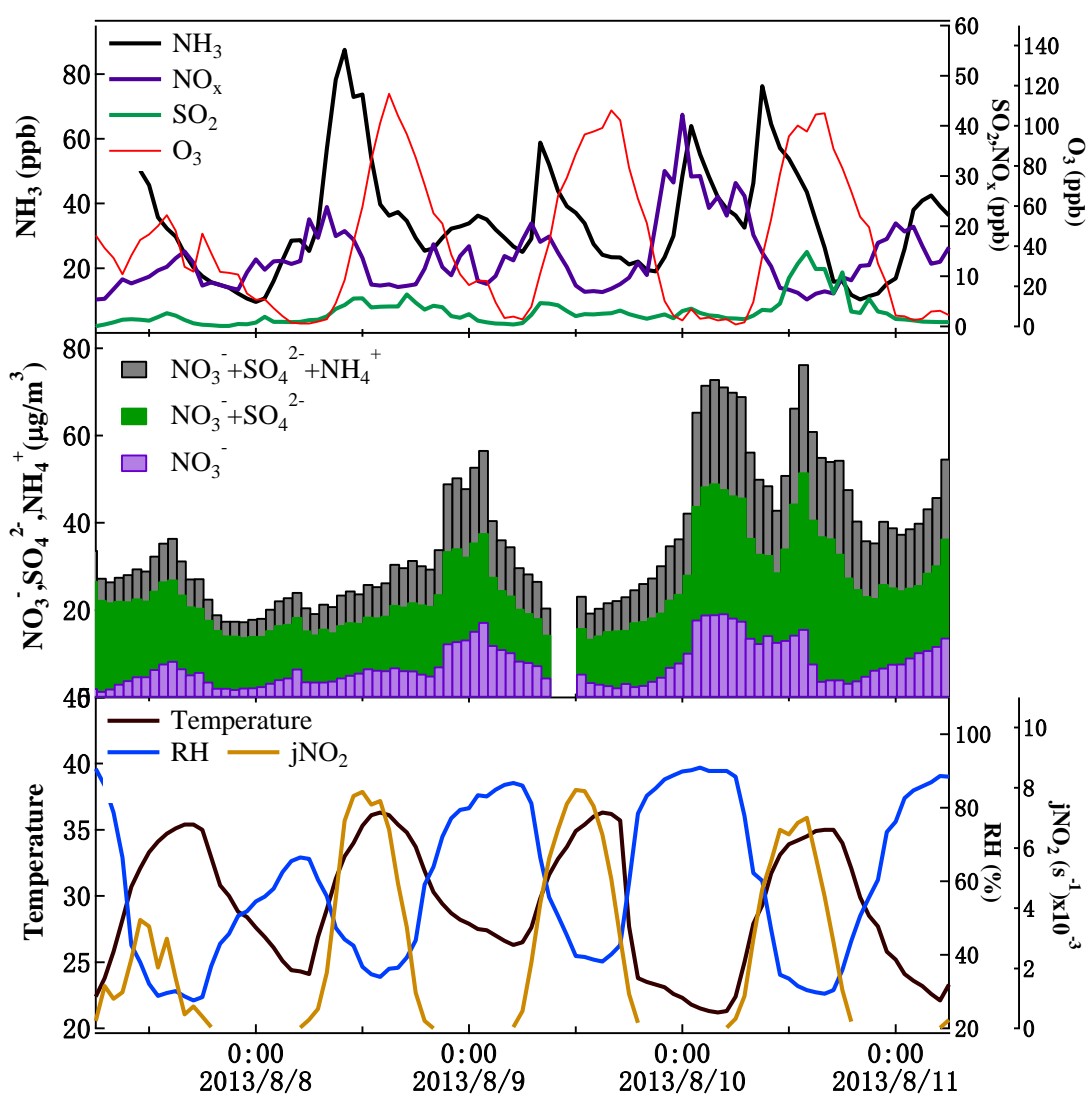

Figure 9. Hourly concentrations of gaseous, ionic species and $j$NO$_2$ measured in the pollution episode during 7-11 August 2013.

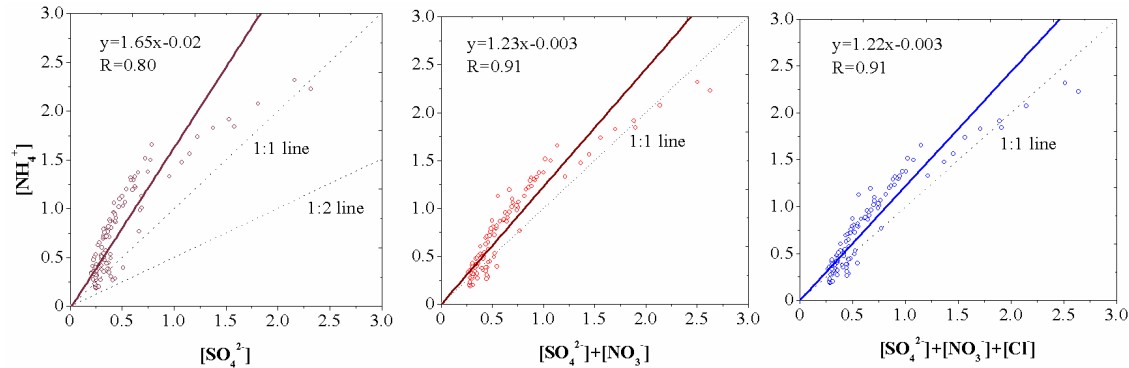

Figure 10. Correlations between [NH$_4^+$] and [SO$_4^{2-}$] (left), [NH$_4^+$] and [SO$_4^{2-}$]+[NO$_3^-$] (middle) and [NH$_4^+$] and [SO$_4^{2-}$]+[NO$_3^-$]+[Cl$^-$] (right) during 7-11 August 2013.

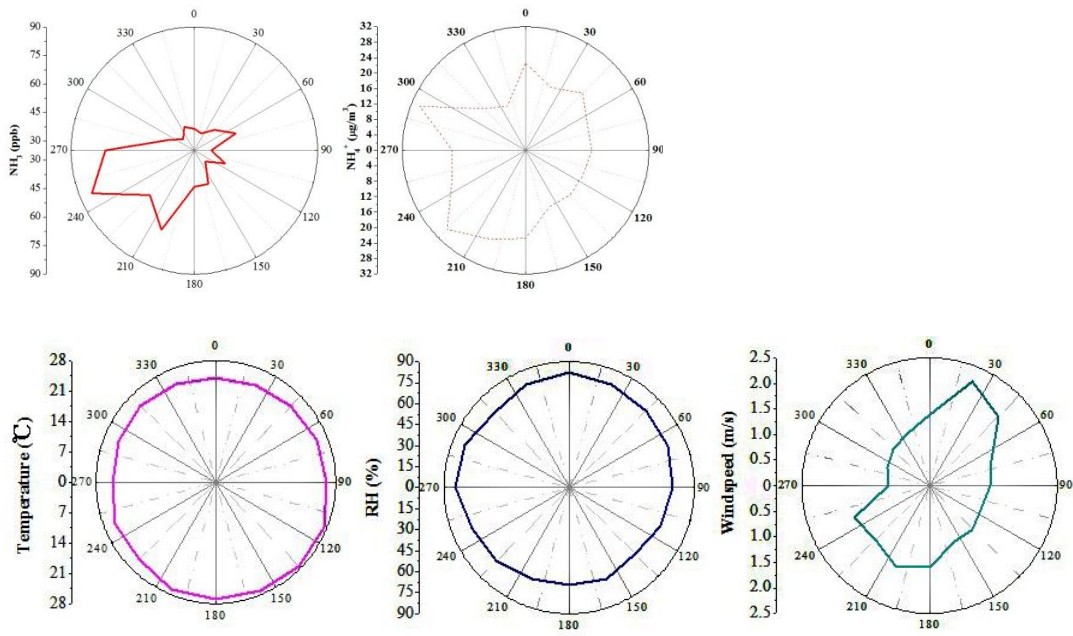

Figure 11. The average NH$_3$, NH$_4^+$ and meteorological data roses in different wind sectors during summer 2013.

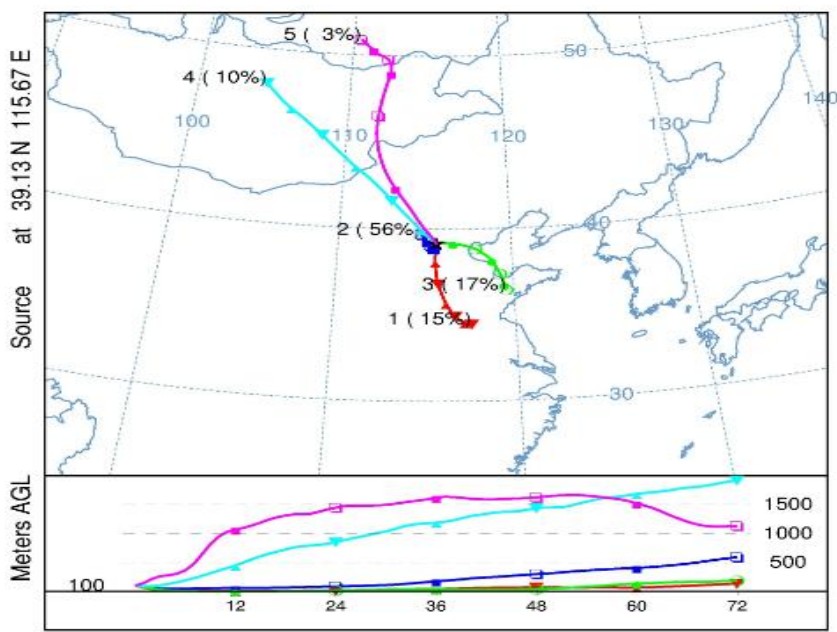

Figure 12. 72-h backward trajectories for 100 m above ground level at Gucheng during sampling period 2013.

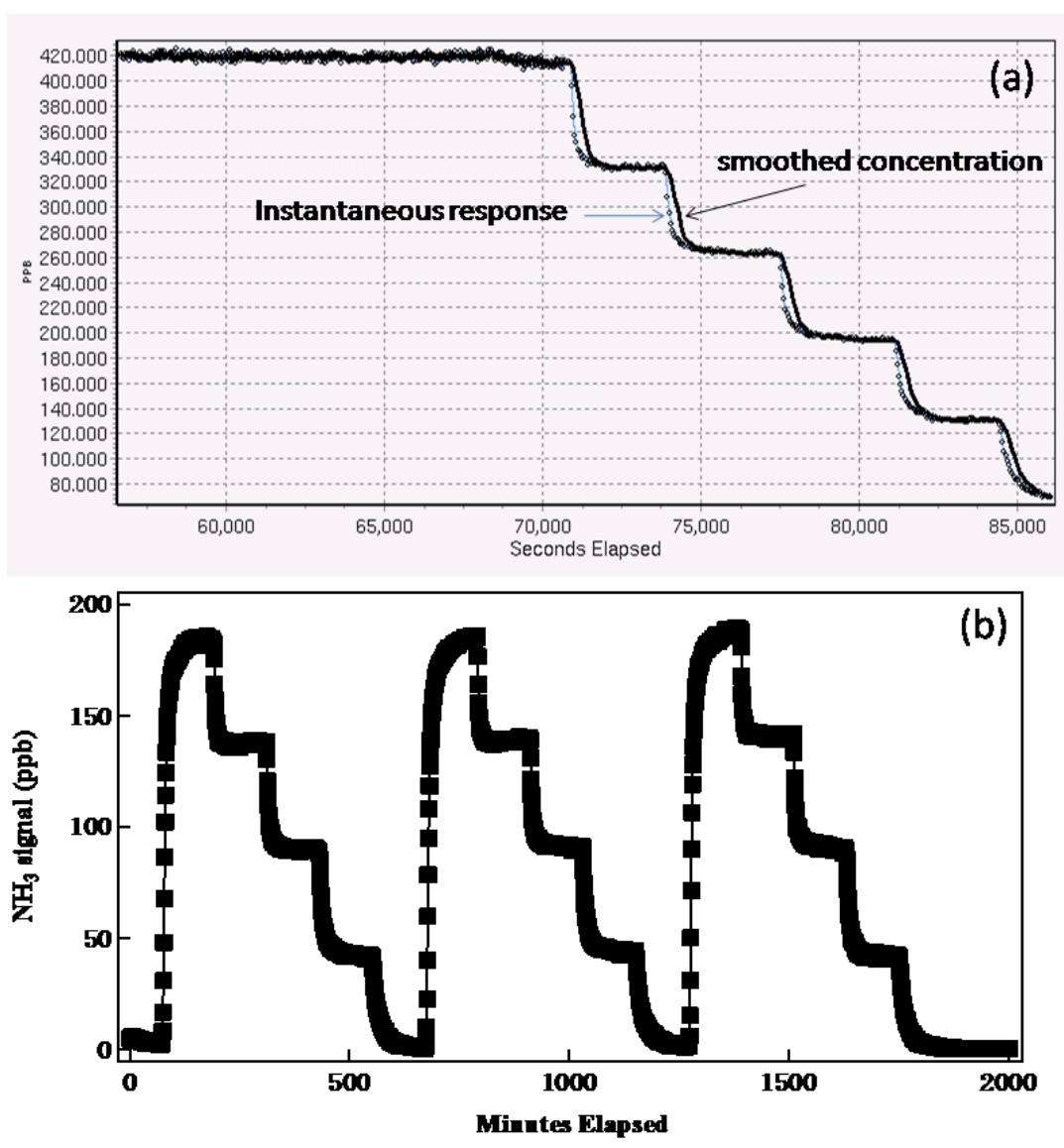

Figure S1. Confirmation of the performance of NH$_3$ analyzer using diluted standard gas (mixture NH$_3$/N$_2$). Instrument response to changed NH$_3$ concentration and stability (a) and repeated multipoint calibrations (b).

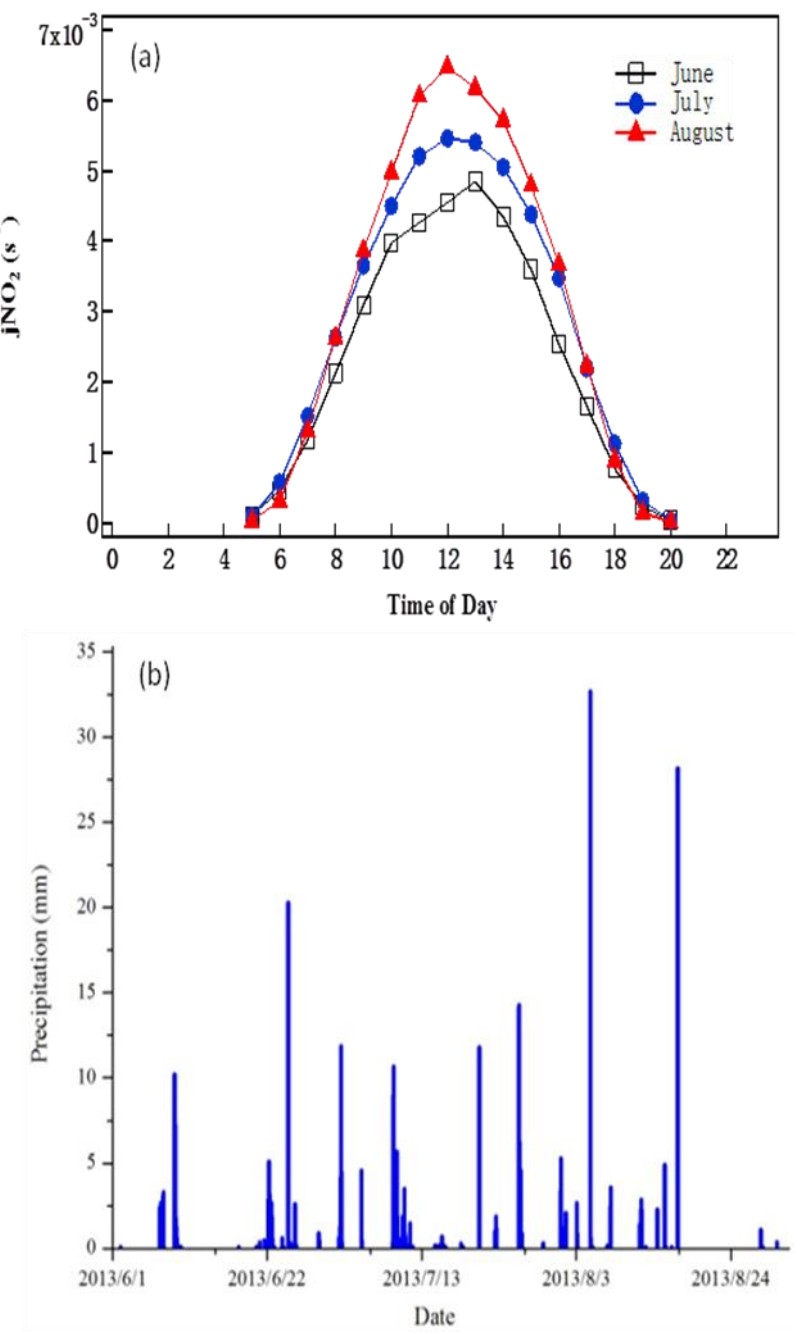

Figure S2. Monthly diurnal variations of photolysis rate coefficient of $NO_2$ ($jNO_2$) (a) and hourly amount of precipitation (b) in summer 2013.

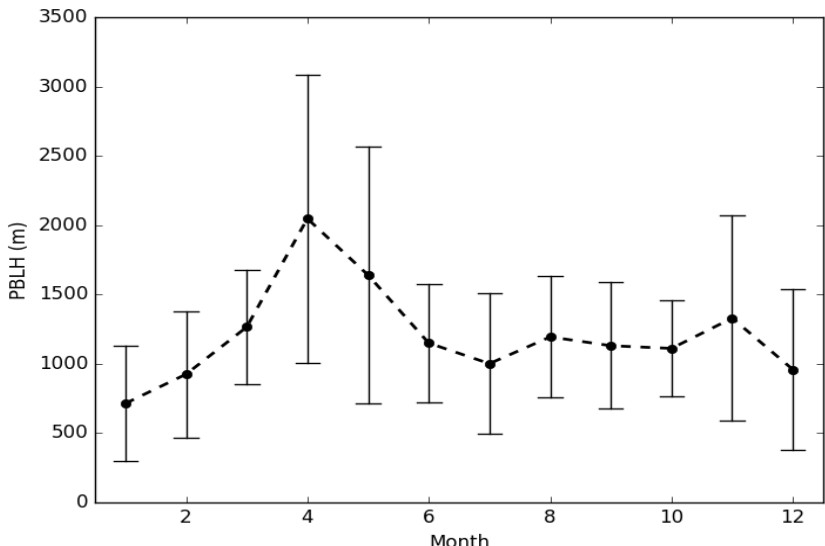

Figure S3. The monthly planetary boundary layer heights at 14:00 during 2013 at Gucheng.

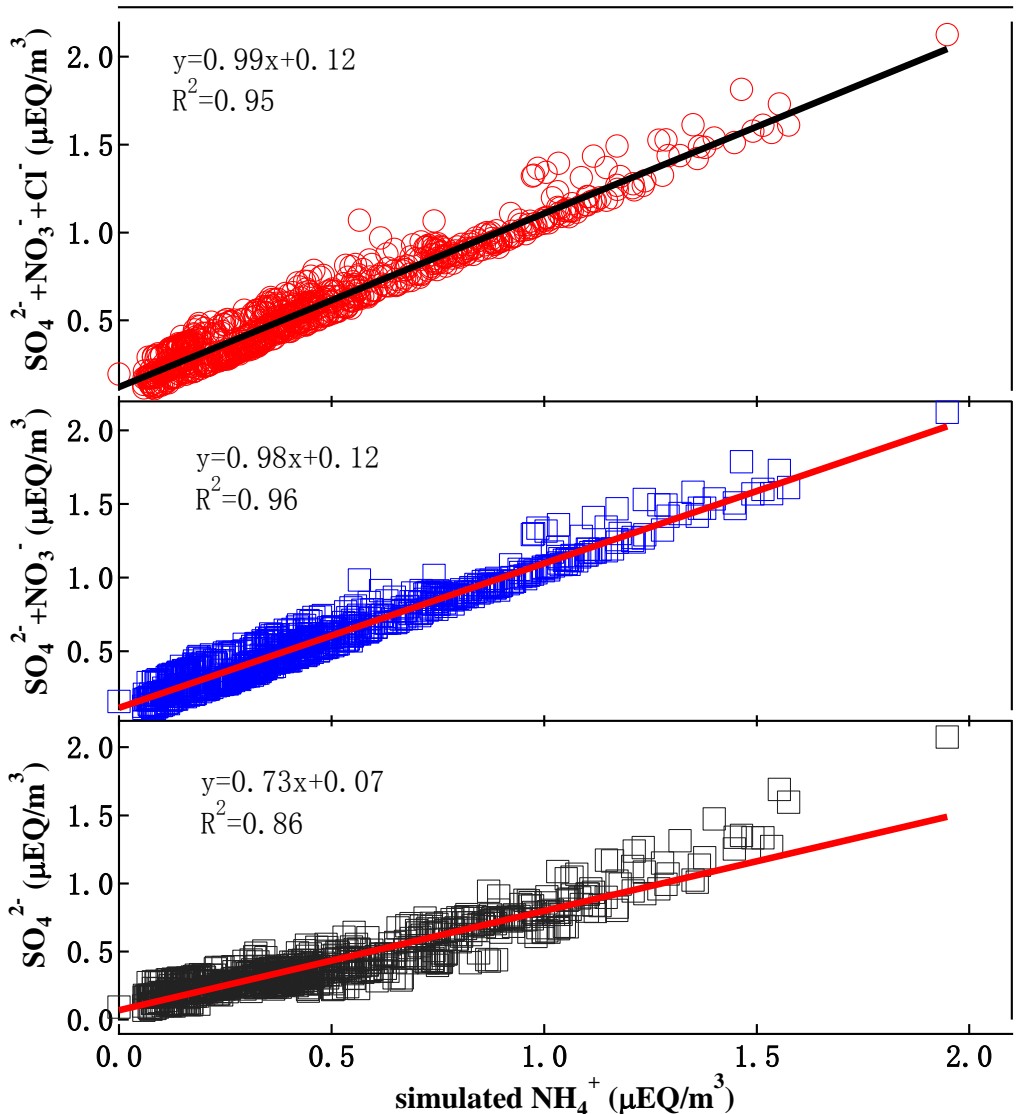

Figure S4. Correlation of modelled $NH_4^+$ with modelled $SO_4^{2-}$, $SO_4^{2-}+NO_3^-$ and $SO_4^{2-}+NO_3^-+Cl^-$.

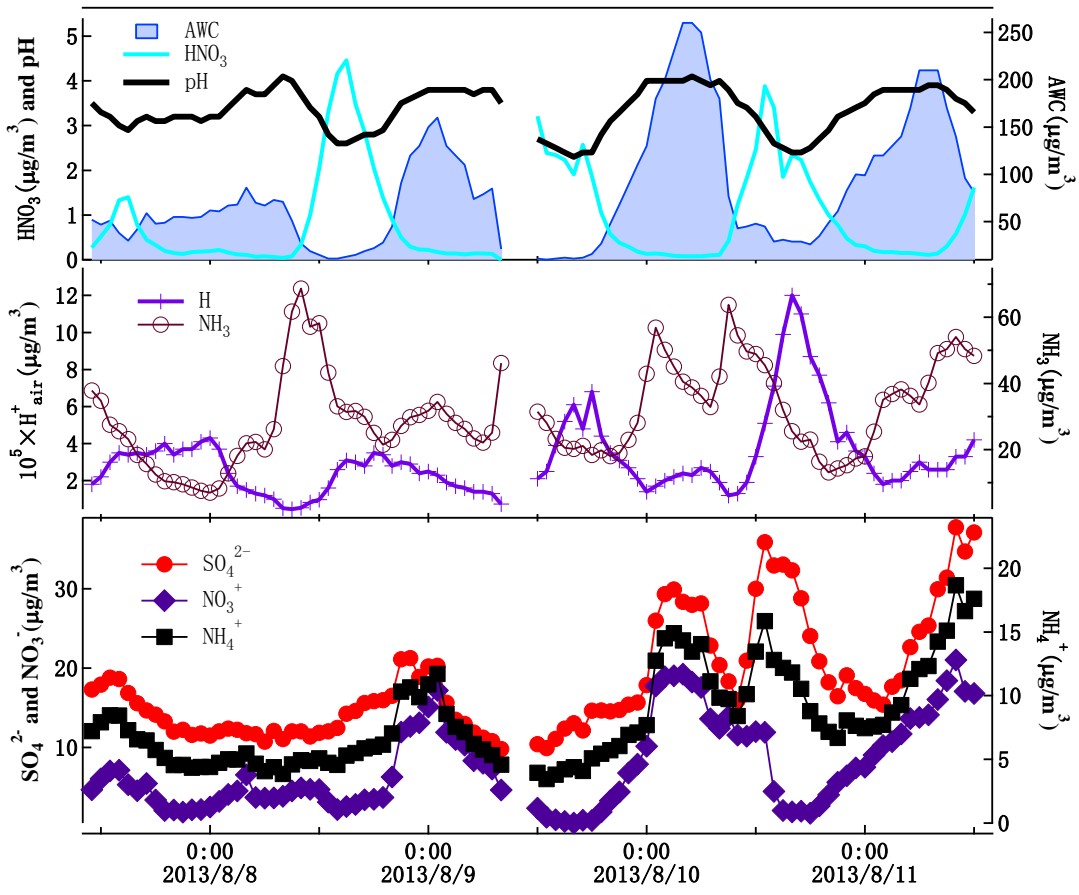

Figure S5. Time series of predicted fine particle pH, particle water mass, HNO₃, H$_{air}^+$, NH₃ and inorganic ions during 7-11 August 2013.