# Peer review of "Role of ambient ammonia in particulate ammonium formation at a rural site in the North China Plain"

_Atmospheric Chemistry and Physics, 2017_

## Referee Comment (RC1) · Anonymous Referee #2 · 31 May 2017

This manuscript presents a comprehensive ambient measurement dataset, including various trace gases and particulate species, for over four months at a rural site in the North China Plain (NCP). Ammonia (NH3) is the focus of this study for its role in the formation of secondary inorganic aerosols, which accounts for a major fraction of PM2.5 in NCP. The hourly resolution, higher than many of the previous ambient ammonia measurements, enables detailed studies on individual pollution events and the diurnal variations. However, I hope the authors can take better advantage of this dataset, and go deeper into Atmospheric Chemistry and Atmospheric Physics, as indicated by the journal name. For example, this work aims to understand the impact of ammonia on secondary ammonium aerosols (page 1, line 20), facilitate developing future ammonia emission control policies (page 1, line 32), and examine the sources of ammonia and ammonium and their chemical conversion mechanism (page 3, line 8). These are all

important issues, but I am not convinced that this article has advanced our current knowledge and understanding about these issues after reading it.

Major comments

1. It takes a significant part of this manuscript to explain the observed concentrations. However many of the explanations are qualitative and even speculative. Further quantitative evidences are needed. To name a few:

Page 8, lines 5-6, "the monthly concentration of SO2, NOx, and CO in July and August decreased because of raid photochemical reduction, additional removal by rainfall, and excellent vertical mixing." What are the evidences of photochemistry, wet scavenging, and vertical mixing?

Page 8, lines 8-9, the ozone was highest in June because of "photochemical production, intense burning of biomass, and transport of regional pollution". What are the evidences of more photochemistry, biomass burning, and regional transport in June? Shouldn't July have larger photochemical rates?

Page 8, lines 26-27, the downward mixing of the residual layer containing higher ammonia concentration could lead to an increase of ammonia in the morning. This would require a very large pool of ammonia in the residual layer. Why did it not happen in other months?

Page 9, line 5, the author explains the earlier ammonia morning peak in July by increased emissions. Further evidence?

2. A high observed concentration can always be explained by more emission, less mixing, or less removal. I think a publication in ACP should go beyond reporting the concentrations of these short-lived species, as the concentrations are highly variable. This study used the ratio between ammonium and NHx to infer the gas-particle conversion of ammonia. However the ammonium and ammonia may be from different sources, where ammonium is formed in the city with NOx and SO2, and ammonia is

emitted locally. In other words, what if NHx and NH3 are decoupled?

At page 10, line 13, it is summarized that "This observation emphasizes the important role of NH3 in the formation of secondary SO4, NO3 and NH4 aerosols, which should be further explored . . .". The title of this manuscript is about the role of ammonia on secondary inorganic aerosols, but what exactly is this role? It is not satisfying to only know it is important and needs further exploration.

Specific comments

Page 1, line 29: please define "transport of air mass from the North China Plain region", as the site itself is in the middle of NCP.

Page 3, line 22: how large is the "surrounding area" that impacts the measurements of this site?

Section 2.2: what's the response time of the Los Gatos instrument? What is the concentration and accuracy of the calibration gas?

Page 6, line 3: please define "human activity" as it seems a very broad concept.

Page 6, line 5: there is no "Zhng et al., 2010" in the reference list.

Page 6, line 8-9: please clarify how can these results "be used in improving NH3 emission inventory and making future emission control policies".

Figure 2: I understand ammonia is shown in log scale because its concentration spanned three orders of magnitude. However I suggest add a sub plot with linear scale so that the variability is comparable at different concentration levels and the individual spikes from pollution episodes are clearer.

Page 7, line 6: where the urea was applied and how large was the applied area?

Page 7, line 29: what are these "trace gases"?

Page 8, lines 14-15 and page 8, lines 30: these two sentences seem contradict each

other.

Page 10, line 1: higher NO3 level than what?

Page 10, line 20-25: I suggest add a figure showing the slope and correlations. The SO4 should be normalized with its number of charge. What is the evidence for the existence of NH4HSO4?

Page 11, line 17: again, it is better to have more evidences showing that NH3 dry deposition dominates NHx deposition.

Page 12, line 2: where is the reference Meng et al. 2017?
* * *

---

## Referee Comment (RC2) · Anonymous Referee #1 · 5 Jul 2017

Overview

This paper as the other reviewer has pointed out potentially has an amazing dataset which is really needed for greater understanding of air pollution and its impacts in agricultural regions. The paper could be hugely improved by moving away from the gas-particle ratio analysis to more detailed atmospheric chemistry and physics which would allow insight into the processes occurring and whether current understanding of emission, transformation and deposition can explain the observations. There is an overuse of "the data suggests..." and "this indicates..." without backup of information.

Major comments:

Introduction:

[Figure]

I think the ambition of the paper (as described in the last paragraph) needs to be more detailed and then the critical analysis done in the paper.

P3 More details of instrumentation is needed, in particular the calibration and response time of the NH3 instrument is required. Did the authors see an influence on the response time from PM deposition on the inlet and instrument filters (see Bobrutski et al 2009 and other papers for details of this issue). Some raw data and calibrations would be useful – ACP is not figure limited. Though rainfall is mentioned as a key meteorological driver, the method of measurement and the data are not shown at all.

P 11 section 3.4: Relationship between ammonium and ammonia: This discussion is very brief and limited. In particular after noting previously (and probably correctly) that local ammonia emissions dominate the chemical speciation observed, the authors then infer "NH3 dominates NHx deposition". With the dataset they have they could have performed calculations of deposition vs emission over the 4 month period would have given much more insight, i.e. the process is bidirectional therefore it is uncertain whether any net deposition would occur under the ambient conditions. This is a missed opportunity to explore the atmospheric chemistry and physics of the system

Given that there are several thermodynamic models freely available, it would have been useful to explore the dataset against what is predicted by models. What is the ion balance – are dicarboxylic acids needed to explain aerosol neutralisation? (is it neutral?). Is the atmospheric chemistry at the site dominated by thermodynamic equilibrium or is there kinetic limitations on the processes? The authors have missed an opportunity with the dataset to fully understand the chemistry and rely in the results and discussions to discuss the ratios between gas and aerosol partitioning to explain scientific concepts which are known and therefore not surprising.

In the places in the manuscript which use atmospheric chemistry to explain data (e.g. 3.4.2), there are no calculations to check if what is observed is what would be expected under the conditions. Is ozone being lost to the surface or is there a haze which allows

aqueous processing in the atmosphere, what might be the role of organics...), what is the surface area of PM (given composition and RH) and hence can N2O5 hydrolysis explain the observations completely? There are lots of questions which are not touched upon, though they are key to understanding the role of NH3.

Having read the paper I am still not sure what the authors want a reader to learn from the gas-particle ratios. I would suggest the authors revise to include pollution/wind rose diagrams to look at the pollution footprint (e.g. ones are available on Open air and other packages), use current thermodynamics and kinetics of the system to see if current models would accurately represent the observations, if not what may be missing?

Oxidised nitrogen chemistry and the gas-aerosol partitioning dynamic are mentioned in passing but are key to understanding whether NH3 is driving the PM formation or it is a reservoir gas which grows PM when the presence of the other pollutants is there. Biomass burning is mentioned and K and CO as the indicators. With the dataset they could estimate the fraction of PM due to biomass burning, how much of the PM is explained by biomass burning, and does the biomass burning "seed" larger PM events. Finally the discussion and conclusion would be enhanced if some discussion about the impacts and potential solutions to the impacts. What is the evidence that limiting the NH3 emission would improve the air quality - it may well achieve this, but to make the case, evidence or hypothesis is needed to back the statements up.

Minor corrections:

P1 Line 22 The observation that NH3 drives NH4 content of PM is not new, so I do not think the word " suggesting" is appropriate

P1 Line 24: This is a percentage not a ratio.

P1 Line 25: use previous NCP abbreviation

P2 line 15: Actually most atmospheric chemistry text books discuss this, would cite them rather than research papers.

P5, line 19: asymmetric errors would be more appropriate given that one cannot have negative concentrations.

P25 Figure 2: given that it rained during the 4 months, why does the RH never go above 90%?

P27: for the PM composition it would be useful to have them as stacked so that one can see the variation of composition through time References: There are not many references from 2015 and 2016 despite many papers being published on this subject area. I would suggest the authors review the recent literature.

---

## Author Comment (AC2) · 20 Sep 2017

Please see our response in the pdf file.

Please also note the supplement to this comment:
https://www.atmos-chem-phys-discuss.net/acp-2017-174/acp-2017-174-AC2-supplement.pdf
* * *
[Figure]
**Response to comments by editor and anonymous reviewers**

We appreciate the editor and anonymous reviewers' comments and helpful suggestions. We have revised the manuscript according to their comments and suggestions. We hope the revised manuscript can meet the quality requirements of *Atmospheric Chemistry and Physics*.

**Response to Reviewer #1's comments**

**Anonymous Reviewer #1**

**Overview**

**This paper as the other reviewer has pointed out potentially has an amazing dataset which is really needed for greater understanding of air pollution and its impacts in agricultural regions. The paper could be hugely improved by moving away from the gas-particle ratio analysis to more detailed atmospheric chemistry and physics which would allow insight into the processes occurring and whether current understanding of emission, transformation and deposition can explain the observations. There is an overuse of "the data suggests..." and "this indicates..." without backup of information.**

**Answer:** Thank you for your comments and suggestions, which are valuable in improving the quality of our manuscript. We have made additional data analysis and revised the manuscript according to the comments and suggestions by both reviewers. To gain more insight into the role of ammonia in the formation of secondary inorganic aerosol, simulations were made using the thermodynamic equilibrium model ISORROPIA II. The measurements were used as input of model to simulate the variations of the components in gas, liquid and solid phases, which are useful in the investigation of the gas-aerosol equilibrium characteristics.

**Major comments:**

**Introduction:**

**I think the ambition of the paper (as described in the last paragraph) needs to be more detailed and then the critical analysis done in the paper.**

**P3 More details of instrumentation is needed, in particular the calibration and response time of the NH3 instrument is required. Did the authors see an influence on the response time from PM deposition on the inlet and instrument filters (see Bobrutski et al 2009 and other papers for details of this issue). Some raw data and calibrations would be useful – ACP is not figure limited. Though rainfall is mentioned as a key meteorological driver, the method of measurement and the data are not shown at all.**

**Answer:** We have added some more details about the instrumentation and issues regarding quality control. We paid attention to the influences on $NH_3$ measurements from the inlet and PM deposition on it and tried to reduce such influences. Although we do not exclude some unperceivable influences from adsorption and desorption, these influences should be much smaller than the high $NH_3$ values we observed and cause mainly slightly slower response or a lag in the recorded $NH_3$ concentration, which may not impact our analysis based on hourly average data. Some figures are added in the revised

**Fig. 1.**

**Supplement:**

**Response to comments by editor and anonymous reviewers**

We appreciate the editor and anonymous reviewers' comments and helpful suggestions. We have revised the manuscript according to their comments and suggestions. We hope the revised manuscript can meet the quality requirements of *Atmospheric Chemistry and Physics*.

Response to Reviewer #1's comments Anonymous Reviewer #1 Received and published: 5 July 2017 Overview

This paper as the other reviewer has pointed out potentially has an amazing dataset which is really needed for greater understanding of air pollution and its impacts in agricultural regions. The paper could be hugely improved by moving away from the gas-particle ratio analysis to more detailed atmospheric chemistry and physics which would allow insight into the processes occurring and whether current understanding of emission, transformation and deposition can explain the observations. There is an overuse of "the data suggests..." and "this indicates..." without backup of information.

**Answer:** Thank you for your comments and suggestions, which are valuable in improving the quality of our manuscript. We have made additional data analysis and revised the manuscript according to the comments and suggestions by both reviewers. To gain more insight into the role of ammonia in the formation of secondary inorganic aerosol, simulations were made using the thermodynamic equilibrium model ISORROPIA II. The measurements were used as input of model to simulate the variations of the components in gas, liquid and solid phases, which are useful in the investigation of the gas-aerosol equilibrium characteristics.

**Major comments:**

**Introduction:**

I think the ambition of the paper (as described in the last paragraph) needs to be more detailed and then the critical analysis done in the paper.

P3 More details of instrumentation is needed, in particular the calibration and response time of the NH3 instrument is required. Did the authors see an influence on the response time from PM deposition on the inlet and instrument filters (see Bobrutski et al 2009 and other papers for details of this issue). Some raw data and calibrations would be useful – ACP is not figure limited. Though rainfall is mentioned as a key meteorological driver, the method of measurement and the data are not shown at all.

**Answer:** We have added some more details about the instrumentation and issues regarding quality control. We paid attention to the influences on NH3 measurements from the inlet and PM deposition on it and tried to reduce such influences. Although we do not exclude some unperceivable influences from adsorption and desorption, these influences should be much smaller than the high NH3 values we observed and cause mainly slightly slower response or a lag in the recorded NH3 concentration, which may not impact our analysis based on hourly average data. Some figures are added in the revised

manuscript and supplementary materials to provide information about calibration, meteorological condition, etc. Section 2.2 "Sampling and analysis" has been rewritten as follows:

"Ambient NH3 was measured using an ammonia analyser (DLT-100, Los Gatos Research, USA), which utilize a unique laser absorption technology called Off-Axis Integrated Cavity Output Spectroscopy (OA-ICOS). The analyzer has a precision of 0.2 ppb at 100 sec average and a maximum drift of 0.2 ppb over 24 hrs. The response time of the analyzer is less than 2 s (with optional external N920 vacuum pump). During the campaign, NH3 data were recorded as 100-s average. In principle, the NH3 analyzer does not need external calibration, because the measured fractional absorption of light at an ammonia resonant wavelength is an absolute measurement of the ammonia density in the cell (Manual of Economical Ammonia Analyzer - Benchtop Model 908-0016, Los Gatos Research). However, we confirmed the good performance of the NH3 analyzer using a reference gas mixture  $NH_3/N_2$  (Scottgas, USA) traceable to US National Institute for Standards and Technology (NIST). The reference gas of  $NH_3$  (25.92 ppm with an accuracy of  $\pm 2\%$ ) was diluted to different concentrations using zero air and supplied to the analyzer and a sequence with 5 points of different  $NH_3$ concentrations (including zero) were repeated for several times to check the performance of the analyzer. As shown in Fig. S1, the analyzer followed rapidly to changes of the  $NH_3$  concentration, produced stable response under stabilized NH3 concentrations, and repeated accurately (within the uncertainty) the supplied NH3 concentrations. The NH3 analyzer contains an internal inlet aerosol filter, which was cleaned before our campaign. Nevertheless, some very fine particles can deposit on the mirrors of the ICOS cell, leading to gradual decline in reflectivity. However, slight mirror contamination does not cause errors in NH3 measurements because the mirror reflectivity is continually monitored and the measurement is compensated using the mirror ringdown time. Interferences to NH3 measurements can be from the sample inlets, for example, due to water condensation or adsorption/desorption effects (e.g., Schwab, 2008; Norman et al., 2009). Such interferences were not quantified but reduced as possibly as we could. PTFE tubing (4.8 mm ID), which is one of the well suited materials for NH3 measurement (Norman et al., 2009), was used to induced ambient air. The length of the tubing was kept as short as possible (about 5 m) to limit the residue time to less than 3 s. The aerosol filter at the inlet was changed every two weeks. Water condensation was avoided. Nevertheless, we cannot exclude the influence from the adsorption and desorption, which can also occur on dry surfaces. However, this influence should be small at our site, where the NH3 concentration is very high, and cause mainly a lag in the recorded NH3 concentration.

A set of commercial instruments from Thermo Environmental Instruments, Inc. were used to measure  $O_3$  (TE 49C), NO/NO2/NOx (TE 42CTL), CO (TE 48C), and SO2 (TE 43CTL). All instruments were housed in an air-conditioned room in the observation building at the site. Two parallel inlet tubes (Teflon, 4.8 mm ID×8 m length) were shared by the analyzers. The height of the inlets was 1.8 m above the roof of the building and about 8m above the ground. The inlet residence time was estimated to be less than 5 s (Lin et al., 2009). Zero and span checks were performed weekly on the analyzers of these trace gases to identify possible analyzer malfunctions and zero drifts. Multipoint calibrations of SO2, NOx, CO and O3 analyzers were performed on the instruments at approximately 1-month intervals. Measurement records were saved as 1-min averages After the correction of data on the basis of the multipoint calibrations, hourly average data were calculated and used for the analysis.

An Ambient Ion Monitor (AIM) (URG 9000D Series, USA) was deployed at the site to measure hourly concentrations of water-soluble inorganic components in PM2.5 during 15 June–11 August, 2013. A detailed description of performance evaluation of AIM-IC system is reported by Han et al., (2016). Briefly, ambient air was introduced in to the AIM with a 2 meter Teflon coated aluminum pipe and particles larger than 2.5  $\mu$ m were removed by a cyclone at a flow rate of 3 L/min. A liquid diffusion denuder was used to remove the interfering acidic and basic gases, in combination with a Steam-Jet Aerosol Collector followed by an Aerosol Sample Collector, until the particles can be injected into the ion chromatograph (Hu et al., 2014). The detection limit of NH4+, SO42- and NO3- were 0.05  $\mu$ g m-3, 0.04  $\mu$ g m-3 and 0.05  $\mu$ g m-3, respectively. For the AIM, multipoint calibrations were performed weekly by using calibration standard solutions. Acceptable linearity of ions was obtained with an R2 of ≥0.999. The flow rate of the AIM was checked weekly at the sample inlet with a certified flow meter. The flow rate of the AIM was kept at 3 L/min with standard derivation of

Figure S1 Confirmation of the performance of  $NH_3$  analyzer using diluted standard gas (mixture  $NH_3/N_2$ ). (a) Instrument response to changed  $NH_3$  concentration and stability; (b) repeated multipoint calibrations.

Figure S2. Monthly average diurnal variations of the NO2 photolysis frequency  $(jNO_2)$  (a) and hourly rainfall (b) observed at Gucheng during June-August 2013.

- Han, B., Zhang, R., Yang, W., Bai, Z., Ma, Z., and Zhang, W.: Heavy haze episodes in Beijing during January 2013: inorganic ion chemistry and source analysis using highly time-resolved measurements from an urban site, Sci. Total Environ., 544, 319-329, 2016.
- Lin, W., Xu, X., and Zhang, X.: Characteristics of gaseous pollutants at Gucheng, a rural site southwest of Beijing, J. Geophys. Res., 114, 10339, 2009.
- Norman, M., Spirig, C., Wolff, V., Trebs, I., Flechard, C., Wisthaler, A., Schnitzhofer, R., Hansel, A., and Neftel, A.: Intercomparison of ammonia measurement techniques at an intensively managed grassland site (Oensingen, Switzerland), Atmos. Chem. Phys., 9, 2635–2645, 2009.
- Schwab, J.J.: Ambient Gaseous Ammonia: Evaluation of Continuous Measurement Methods Suitable for Routine Deployment, Final Report Prepared for the New York State Energy Research and Development Authority (NYSERDA), Final Report 08-15, New York, October 2008.
- Wu, W. S. and Wang, T.: On the performance of a semi-continuous PM2.5 sulphate and nitrate instrument under high loadings of particulate and sulphur dioxide. Atmos. Environ. 41, 5442-5451, 2007.

P 11 section 3.4: Relationship between ammonium and ammonia: This discussion is very brief and limited. In particular after noting previously (and probably correctly) that local ammonia emissions dominate the chemical speciation observed, the authors then infer "NH3 dominates NHx deposition". With the dataset they have they could have performed calculations of deposition vs emission over the 4 month period would have given much more insight, i.e. the process is bidirectional therefore it is uncertain whether any net deposition would occur under the ambient conditions. This is a missed opportunity to explore the atmospheric chemistry and physics of the system.

**Answer:** Thank you for your valuable comments. Indeed it would be great if we could use our measurements to systematically explore the atmospheric chemistry and physics over the our site and to gain more insight, including the emission and deposition of  $NH_3$  and  $NH_4^+$ . However, we did not observe the emission and deposition of  $NH_3$  and  $NH_4^+$ . The air/surface exchange of NH3 is very complex, bidirectional and highly variable, and is influenced by many factors (e.g., Schrader et al., 2016). Given the parameters we observed, it is hardly possible to obtain robust and quantitative results about the emission vs deposition purely based on available measurements. To well understand the processes of emission, conversion and deposition of NHx species in the polluted NCP region, it is highly needed to design observational and 3D modeling studies. Our in-situ measurements present in this paper can be used as supporting data in future modeling studies.

Schrader, F., Brümmer, C., Flechard, C.R., Wichink Kruit, R.J., van Zanten, M.C., Zöll, U., Hensen, A., and Erisman, J.W.: Non-stomatal exchange in ammonia dry deposition models: comparison of two state-of-the-art approaches, Atmos. Chem. Phys., 16, 13417–13430, 2016.

Given that there are several thermodynamic models freely available, it would have been useful to explore the dataset against what is predicted by models. What is the ion balance – are

dicarboxylic acids needed to explain aerosol neutralisation? (is it neutral?). Is the atmospheric chemistry at the site dominated by thermodynamic equilibrium or is there kinetic limitations on the processes? The authors have missed an opportunity with the dataset to fully understand the chemistry and rely in the results and discussions to discuss the ratios between gas and aerosol partitioning to explain scientific concepts which are known and therefore not surprising.

**Answer:** Thank you for your suggestions. We have used the thermodynamic equilibrium model (ISORROPIA II) to investigate gas-aerosol partitioning characteristics and compared the modeling results with our measurements. The pH values of aerosol, estimated based on the simulated results, are mostly in the range of 2.5-4.5, with an average of 3.5. While simulated sulfate and nitrate are well comparable with the measurements, simulated ammonium substantially underestimates the observed one, indicating the importance of organic acids in the formation of ammonium. Although we did not measure organic acids in aerosol, the presence of oxalic acid and other low molecular weight dicarboxylic acids in aerosols is often reported (e.g., Hsieh et al., 2007; Kawamura et al., 2010, 2013; Sauerwein and Chan, 2017). There is no doubt about the presence of significant amount of dicarboxylic acids over the North China Plain particularly during summer (Kawamura et al., 2013). Therefore, it is highly possible that neutralizing dicarboxylic acids in aerosol particles contributed significantly to the conversion of ammonia to ammonium. Our results also suggest that the gas-aerosol partitioning at the Gucheng site is dominated by thermodynamic equilibrium. We have added a section about the ISORROPIA II model in the revised manuscript and more text about the simulation results and related discussions as follows:

**"2.3.2 ISORROPIA II model**

Thermodynamic gas-aerosol equilibrium characteristics during summer 2013 were examined using ISORROPIA II model (Fountoukis and Nenes, 2007). ISORROPIA II is a thermodynamic equilibrium model for inorganic gases and aerosols in the atmosphere (available at http://isorropia. eas.gatech.edu). To obtain the best available predictions of aerosol pH, ISORROPIA II, was run in the forward mode with metastable aerosol state salts precipitate once the aqueous phase becomes saturated with respect to salts, which often showed better performance than the stable state solution (solid + liquid) and was commonly applied in previous pH predictions (Liu et al., 2017). The concentrations of the measured NH3 and water-soluble ions in PM2.5 were input into the model as total (gas + aerosol) concentrations, along with simultaneously measured relative humidity and temperature data. The bulk particle pH was calculated using the following equation:

$$pH = -\log_{10} \frac{1000H^+_{air}}{AWC} \qquad (4)$$

where  $H^+_{air}$  (µg m-3) and AWC (µg m-3) are the ISORROPIA-II predicted equilibrium particle hydronium ion concentration per volume air and aerosol water content, respectively. Evaluation of the

AWC prediction have been reported in previous studies and shows a good performance compared with particle water measurements (Bian et al., 2014; Guo et al., 2015). "

We have added a new section (Section 3.3 Results from thermodynamic equilibrium simulation) to present our main modeling results and discussed them also other section.

**"3.4 Results from thermodynamic equilibrium simulation**

We have used the thermodynamic equilibrium model ISORROPIA II to investigate gas-aerosol partitioning characteristics.  $NO_3^-$ ,  $SO_4^{2-}$  and  $NH_4^+$ . The model outputs include equilibrium  $NO_3^-$ ,  $SO_4^{2-}$ ,  $NH_4^+$ .  $H^+_{alr}$ ,  $H^+_{alr}$ ,  $HNO_3$ ,  $NH_3$ , AWC, etc. As shown in Fig. 5, the modelled  $NO_3^-$ ,  $SO_4^{2-}$ ,  $NH_3$  show excellent correlations with the corresponding measurements, but modelled  $NH_4^+$  is much worse correlated with the measured one. Modelled  $NO_3^-$ ,  $SO_4^{2-}$ , and  $NH_3$  values agree well with the measurements, while the modelled  $NH_4^+$  largely underestimate the measurements. Considering the unbalance between observed NH4+ and the sum of observed  $SO_4^{2-}+NO_3^-+CI^-$ , we can confirm that other acids in aerosol particles are important in the conversion of  $NH_3$  to  $NH_4^+$ . These other acids may be oxalic acid and other low molecular weight dicarboxylic acids in aerosols is often reported (e.g., Hsieh et al., 2007; Kawamura et al., 2010, 2013; Sauerwein and Chan, 2017). There is no doubt about the presence of significant amount of dicarboxylic acids over the North China Plain particularly during summer (Kawamura et al., 2013). Therefore, it is highly possible that neutralizing dicarboxylic acids in aerosol particles contributed significantly to the conversion of ammonia to ammonium.

The simulated HNO3 concentrations was  $0.9 \pm 1.1 \ \mu g \ m^{-3}$ , showing a maximum value of 7.41  $\mu g \ m^{-3}$  at 11:00 on 19 June 2013. The average diurnal variations of HNO3 and H+air are shown in Fig. S4. The fine particles were moderately acidic in summer, with an average pH values of 3.5. The pH values of aerosol water, estimated based on the simulated results using equation (4), are mostly in the range of 2.5-4.5, with an average of 3.5. On average, pH is over 3.5 during nighttime and below 3.5 during daytime (Fig. 6). Under the medium acidic conditions and high NH3 concentrations, organic acid like diacids are able to reaction with ammonia to for ammonium. Because we used ISORROPIA-II for inorganic aerosol composition and no organic acids measurements are available, we cannot analyze in detail the role of organic acids though the model performed quite well (Fig. S5). "

---

## Author Comment (AC3) · 20 Sep 2017

**Response to comments by editor and anonymous reviewers**

We appreciate the editor and anonymous reviewers' comments and helpful suggestions. We have revised the manuscript according to their comments and suggestions. We hope the revised manuscript can meet the quality requirements of *Atmospheric Chemistry and Physics*.

**Response to Reviewer #2's comments**

**Anonymous Reviewer #2**

**Received and published: 31 May 2017**

This manuscript presents a comprehensive ambient measurement dataset, including various trace gases and particulate species, for over four months at a rural site in the North China Plain (NCP). Ammonia (NH3) is the focus of this study for its role in the formation of secondary inorganic aerosols, which accounts for a major fraction of PM2.5 in NCP. The hourly resolution, higher than many of the previous ambient ammonia measurements, enables detailed studies on individual pollution events and the diurnal variations. However, I hope the authors can take better advantage of this dataset, and go deeper into Atmospheric Chemistry and Atmospheric Physics, as indicated by the journal name. For example, this work aims to understand the impact of ammonia on secondary ammonium aerosols (page 1, line 20), facilitate developing future ammonia emission control policies (page 1, line 32), and examine the sources of ammonia and ammonium and their chemical conversion mechanism (page 3, line 8). These are all important issues, but I am not convinced that this article has advanced our current knowledge and understanding about these issues after reading it.

**Answer:** We thank your comments and suggestions. We have made additional data analysis and revised the manuscript according to the comments and suggestions by both reviewers. To gain more insight into the role of ammonia in the formation of secondary inorganic aerosol, simulations were made using the thermodynamic equilibrium model ISORROPIA II. The measurements were used as input of model to simulate the variations of the components in gas, liquid and solid phases, which are useful in the investigation of the gas-aerosol equilibrium characteristics.

**Major comments**

1. It takes a significant part of this manuscript to explain the observed concentrations. However many of the explanations are qualitative and even speculative. Further quantitative evidences are needed. To name a few:

Page 8, lines 5-6, "the monthly concentration of SO2, NOx, and CO in July and August decreased because of raid photochemical reduction, additional removal by rainfall, and excellent vertical mixing." What are the evidences of photochemistry, wet scavenging, and vertical mixing?

**Answer:** We have added the analysis of some related meteorological conditions and show figures with measurements of rainfall and the  $NO_2$  photolysis frequency (*j* $NO_2$ ) during June-August 2013 as supplementary materials. Changes are made to text as follows:

"The monthly concentrations of  $SO_2$ ,  $NO_x$ , and CO in July and August decreased compared to those in June. In addition to less influences from biomass burning, meteorological conditions were also in favor

of lowering the concentrations of these gases. Figure S2 shows the monthly average diurnal variations of jNO2 and the time-series of hourly rainfall during June-August 2013. As can be seen, the average jNO2 increased from June to August, indicating better conditions for photochemical reduction in July and August. There was also a slight increase in rainfall from June to August, which may promote removal of the pollutants."

Figure S2. Monthly average diurnal variations of the NO2 photolysis frequency (jNO2) (a) and hourly rainfall (b) observed at Gucheng during June-August 2013.

Page 8, lines 8-9, the ozone was highest in June because of "photochemical production, intense burning of biomass, and transport of regional pollution". What are the evidences of more photochemistry, biomass burning, and regional transport in June? Shouldn't July have larger photochemical rates?

**Answer:** After careful consideration we think the statement should be revised. The sentence is changed to "For the secondary pollutant  $O_3$ , the highest concentration was observed in June. This is consistent with previous results from Gucheng (Lin et al., 2009) and should be related with the annual maximum of background  $O_3$  in the NCP, which occurs in June (Lin et al., 2008; Ding et al., 2008). "

- Ding, A. J., Wang, T., Thouret, V., Cammas, J.-P., and Nédélec, P.: Tropospheric ozone climatology over Beijing: analysis of aircraft data from the MOZAIC program, Atmos. Chem. Phys., 8, 1-13, 2008.
- Lin, W., Xu, X., Zhang, X., and Tang, J.: Contributions of pollutants from North China Plain to surface ozone at the Shangdianzi GAW Station, Atmos. Chem. Phys., 8, 5889–5898, 2008.
- Lin, W., Xu, X., and Zhang, X.: Characteristics of gaseous pollutants at Gucheng, a rural site southwest of Beijing, J. Geophys. Res., 114, 10339, 2009.

Page 8, lines 26-27, the downward mixing of the residual layer containing higher ammonia concentration could lead to an increase of ammonia in the morning. This would require a very large pool of ammonia in the residual layer. Why did it not happen in other months?

**Answer:** Indeed, the interpretation about the morning peak is mainly based on the opinions in cited references, which are mostly speculations. We think more investigations are necessary to be able to

clearly and quantitatively explain the morning peak phenomenon. We have revised the 2nd and 3rd paragraphs in section 3.2.2 as follows:

"The morning peak of NH3 was also observed elsewhere and could be resulted from emissions from fertilized soils and plant stomata, evaporation of dew, and human sources, as well as mixing down of ammonia from the residual layer (Trebs et al., 2004; Norman et al., 2009; Bash et al., 2010; Ellis et al., 2011). Figure 3b reveals that the relative humidity (90%-89%) and temperature (21.5-22.1 °C) remained relatively constant before 06:00, but increased later in the morning. The increasing temperature can heat the earth's surface and vegetation leaves and reduce the RH, potentially leading to evaporation of NH3 from soil and plants and volatilization of ammonium aerosol (Trebs et al., 2004; Norman et al., 2009; Ellis et al., 2011), which may increase NH3 concentrations in the morning. When the emission was occurring into a shallow boundary layer, NH3 increase would be more prominent. In addition, the morning rise might also be due to the break-up of the nocturnal boundary layer. During the sampling period, the majority of peaks of ammonia over 50 ppb occurred at night, which were attribute to local emissions, such as from agricultural activity, into a shallow nocturnal boundary layer. It was supposed by Ellis et al. (2011) that the downward mixing of air containing higher NH3 from the residual layer could lead to an increase of surface NH3 after the breakup of the nocturnal boundary layer. "

Figure 3. Diurnal variation (a) NH3 and (b) meteorological parameters during the sampling period.

- Bash, J. O., Walker, J. T., Katul, G. G., Jones, M. R., Nemitz, E., and Robarg, W. P.: Estimation of In-Canopy Ammonia Sources and Sinks in a Fertilized Zea mays Field, Environ. Sci. Tech., 44, 1683-1689, 2010.
- Ellis, R. A., Murphy, J. G., Markovic, M. Z., VandenBoer, T. C., Makar, P. A., Brook, J., and Mihele, C.: The influence of gas-particle partitioning and surface-atmosphere exchange on ammonia during BAQS-Met, Atmos. Chem. Phys., 11, 133-145, 2011.

Norman, M., Spirig, C., Wolff, V., Trebs, I., Flechard, C., Wisthaler, A., Schnitzhofer, R., Hansel, A.,

and Neftel, A.: Intercomparison of ammonia measurement techniques at an intensively managed grassland site (Oensingen, Switzerland), Atmos. Chem. Phys., 9, 2635–2645, 2009.

- Schwab, J.J.: Ambient Gaseous Ammonia: Evaluation of Continuous Measurement Methods Suitable for Routine Deployment, Final Report Prepared for the New York State Energy Research and Development Authority (NYSERDA), Final Report 08-15, New York, October 2008.
- Trebs, I., Meixner, F. X., Slanina J., Oties, R. P., and Andreae, M. O.: Real-time measurements of ammonia, acidic trace gases and water-soluble inorganic aerosol species at a rural site in the Amazon Basin, Atmos. Chem. Phys., 4, 967–987, 2004.

**Page 9, line 5, the author explains the earlier ammonia morning peak in July by increased emissions. Further evidence?**

**Answer:** We have no direct evidence of emissions. However, the Gucheng site is an experiment station for agrometeorological studies. Corn is the main crop in the station area and nearly all the agricultural areas in the surrounding. According the climate in the NCP, corn is planted around the middle of June and grows rapidly in July. Therefore, July is the key period for the application of nitrogen fertilizers like urea. For example, during last ten days of July 2013, 225-300 kg of urea were applied per hectare of station area (Meng et al., 2015), causing huge NH3 spikes during the end of July (Fig. 2). In addition, the highest nighttime temperature in July (Fig. 3b) could promote the soil emission of NH3 and the relatively lower wind speed (Fig. 3b) and lowest PBLH (Fig. S3) in July was in favor of the accumulation of NH3 in surface air. We have revised the 4th and 5th paragraphs in section 3.2.2 as follows:

"-From Fig. 3a, it can be seen that in July the NH3 level was the highest and peaked earliest. One reason for this might be the increased emissions of local agricultural NH3 sources in July compared with those in June, August, and September. On the average, the level NH3 in July had a maximum nighttime increase (20.0 ppb from 20:00 to 06:00), which is much large than those in June (5.2 ppb), August (9.9 ppb) and September (1.8 ppb). The early morning increase of NH3 in July started from a much higher level than in other months, resulting a earliest NH3 peak in July.

There is no direct evidence of increased agricultural NH3 emission in July. However, the Gucheng site is an experiment station for agrometeorological studies. Corn is the main crop in the station area and nearly all the agricultural areas in the surrounding. According the climate in the NCP, corn is planted around the middle of June and grows rapidly in July. Therefore, July is the key period for the application of nitrogen fertilizers like urea. For exampleAs mentioned above, during last ten days of the urea application in the station on 20 July 2013 and a precipitation process afterwards, 225 300 kg of urea were applied per hectare of station area (Meng et al., 2015), causinged huge NH3 spikes during the end of July (Fig. 2b). In addition, the highest nighttime temperature in July (Fig. 3b) could promote the soil emission of NH3, and the relatively lower wind speed (Fig. 3b) and lowest PBLH (Fig. S3) in July was in favor of the accumulation of NH3 in surface air.

In summary, ambient  $NH_3$  at Gucheng showed interesting diurnal cycles, which look significantly different in different summer months. We believe the interplay of some processes, such as emissions from agricultural sources, meteorological conditions (temperature, relativity humidity, wind speed, and

BHL height, etc.) as well as chemical conversion are important in the determination of levels and diurnal patterns of  $NH_3$  at the site. Whether or not these processes are all important in the morning variation of  $NH_3$ ? How important are they? And what makes the difference in the peaking time and concentration of  $NH_3$  in different months? These are questions to be answered in the future."

Figure S3. The monthly planetary boundary layer heights at 14:00 during 2013 at Gucheng.

- Guo, J., Miao, Y., Zhang, Y., Liu, H., Li, Z., Zhang, W., He, J., Lou, M., Yan, Y., Bian, L., and Zhai, P.: The climatology of planetary boundary layer height in China derived from radiosonde and reanalysis data, Atmos. Chem. Phys., 16, 13309-13319, 2016.
- Miao, Y., Guo, J., Liu, S., Liu, H., Li, Z., Zhang, W., and Zhai, P.: Classification of summertime synoptic patterns in Beijing and their associations with boundary layer structure affecting aerosol pollution, Atmos. Chem. Phys., 17, 3097-3110, 2017.

2. A high observed concentration can always be explained by more emission, less mixing, or less removal. I think a publication in ACP should go beyond reporting the concentrations of these short-lived species, as the concentrations are highly variable. This study used the ratio between ammonium and NHx to infer the gas-particle conversion of ammonia. However the ammonium and ammonia may be from different sources, where ammonium is formed in the city with NOx and SO2, and ammonia is emitted locally. In other words, what if NHx and NH3 are decoupled? Answer: We have made simulations using the ISORROPIA II model and analyzed the model results together measurements. We think we have gained more insight than before but also acknowledge that there are limitations in our observation, modeling and data analysis. Some open issues remain to be addressed in future studies.

The ratios NHR ( $NH_4^+/NHx$ ), SOR ( $SO_4^{2^+}/(SO_4^{2^+}+SO_2)$ ) and NOR ( $NO_3^-/(NO_3^-+NOx)$ ) were calculated as measures of chemical conversion of  $NH_3$ ,  $SO_2$  and NOx. You are right that NHx and NH3 may be decoupled. Sources of  $NH_3$ ,  $SO_2$  and NOx may be dislocated. The lifetimes of these gases are different and hence the dispersion areas. FIn-fine aerosol particles may travel much longer than the precursor gases. In the real situation we always observe gases and aerosols originating both from cities and from rural areas, emitted by different sources, and chemically produced. Wherever we measure, we measure is a mixture impacted by different sources, from locations and processes. In this sense, we

The ratio of  $NH_3$  to  $NH_4$ - $(NH_3+NH_4^+)$  has been used to identify the source of  $NH_4$  and the relative contribution of  $NH_3$  to  $NH_4$ -deposition (Lefer et al., 1999; Walker et al., 2004). The parameter NHR defined here could be a measure of the extent of transformation from  $NH_3$  to  $NH_4^+$ -in areas with significant local  $NH_3$  sources, although it encompasses both transformation and local equilibrium, the latter dominating further downwind from the source (Hu et al., 2014).

We have drawn the Figure 8 and added following interpretation in response to reviewer's comment, as follow:

In this study,  $NH_4^+$  concentrations correlated significantly and positively with  $NH_4$  with a correlation coefficient of 0.78 and a slope of 1.48 (Fig.8a, n=915, P<0.01), suggesting that  $NH_4$  played a important precursor role in  $NH_4^+$  in  $PM_{2,5}$  formation.

According to hourly average concentrations, the ratio of  $NH_{3}/NH_{*}$  varied from 0.22 to 0.97, with a mean ratio of 0.69±0.14, suggesting that  $NH_{3}$  remained predominantly in the gas phase rather than the acrosol phase in summer 2013 at Gucheng.

The -change and aerosol species are controlled by emission <del>conversion further, diurnal variation of NH4 was showed in Fig. 8b. Between 08:00–18:00,</del> in an increase in NH4+, which coincided with higher sulfate 4 ammonia is likely result of uptal om  $(NH_4)_2SO_4$ . The diurnal variability of  $NH_*$  may be controlled by transport and vertical of 08:00 18:00 NH, decreased by 43% while NH 10% suggested that NH2 remained predominantly in the gas phase. Between 19:00 and 07:00, NH2 increased and NH\* increased by 51%, indicating that gas-particle partitioning contribute <del>by</del> significantly to the decrease in gas phase ammonia during this time

---

## Author Comment (AC5) · 20 Sep 2017

**Response to comments by editor and anonymous reviewers**

We appreciate the editor and anonymous reviewers' comments and helpful suggestions. We have revised the manuscript according to their comments and suggestions. We hope the revised manuscript can meet the quality requirements of *Atmospheric Chemistry and Physics*.

**Response to Reviewer #1's comments**

**Anonymous Reviewer #1**

**Overview**

**This paper as the other reviewer has pointed out potentially has an amazing dataset which is really needed for greater understanding of air pollution and its impacts in agricultural regions. The paper could be hugely improved by moving away from the gas-particle ratio analysis to more detailed atmospheric chemistry and physics which would allow insight into the processes occurring and whether current understanding of emission, transformation and deposition can explain the observations. There is an overuse of "the data suggests..." and "this indicates..." without backup of information.**

**Answer:** Thank you for your comments and suggestions, which are valuable in improving the quality of our manuscript. We have made additional data analysis and revised the manuscript according to the comments and suggestions by both reviewers. To gain more insight into the role of ammonia in the formation of secondary inorganic aerosol, simulations were made using the thermodynamic equilibrium model ISORROPIA II. The measurements were used as input of model to simulate the variations of the components in gas, liquid and solid phases, which are useful in the investigation of the gas-aerosol equilibrium characteristics.

**Major comments:**

**Introduction:**

**I think the ambition of the paper (as described in the last paragraph) needs to be more detailed and then the critical analysis done in the paper.**

**P3 More details of instrumentation is needed, in particular the calibration and response time of the NH3 instrument is required. Did the authors see an influence on the response time from PM deposition on the inlet and instrument filters (see Bobrutski et al 2009 and other papers for details of this issue). Some raw data and calibrations would be useful – ACP is not figure limited. Though rainfall is mentioned as a key meteorological driver, the method of measurement and the data are not shown at all.**

**Answer:** We have added some more details about the instrumentation and issues regarding quality control. We paid attention to the influences on $NH_3$ measurements from the inlet and PM deposition on it and tried to reduce such influences. Although we do not exclude some unperceivable influences from adsorption and desorption, these influences should be much smaller than the high $NH_3$ values we observed and cause mainly slightly slower response or a lag in the recorded $NH_3$ concentration, which may not impact our analysis based on hourly average data. Some figures are added in the revised

manuscript and supplementary materials to provide information about calibration, meteorological condition, etc. Section 2.2 "Sampling and analysis" has been rewritten as follows:

"Ambient $NH_3$ was measured using an ammonia analyser (DLT-100, Los Gatos Research, USA), which utilize a unique laser absorption technology called Off-Axis Integrated Cavity Output Spectroscopy (OA-ICOS). The analyzer has a precision of 0.2 ppb at 100 sec average and a maximum drift of 0.2 ppb over 24 hrs. The response time of the analyzer is less than 2 s (with optional external N920 vacuum pump). During the campaign, $NH_3$ data were recorded as 100-s average. In principle, the $NH_3$ analyzer does not need external calibration, because the measured fractional absorption of light at an ammonia resonant wavelength is an absolute measurement of the ammonia density in the cell (Manual of Economical Ammonia Analyzer - Benchtop Model 908-0016, Los Gatos Research). However, we confirmed the good performance of the $NH_3$ analyzer using a reference gas mixture $NH_3/N_2$ (Scottgas, USA) traceable to US National Institute for Standards and Technology (NIST). The reference gas of $NH_3$ (25.92 ppm with an accuracy of $\pm2\%$) was diluted to different concentrations using zero air and supplied to the analyzer and a sequence with 5 points of different $NH_3$ concentrations (including zero) were repeated for several times to check the performance of the analyzer. As shown in Fig. S1, the analyzer followed rapidly to changes of the $NH_3$ concentration, produced stable response under stabilized $NH_3$ concentrations, and repeated accurately (within the uncertainty) the supplied $NH_3$ concentrations. The $NH_3$ analyzer contains an internal inlet aerosol filter, which was cleaned before our campaign. Nevertheless, some very fine particles can deposit on the mirrors of the ICOS cell, leading to gradual decline in reflectivity. However, slight mirror contamination does not cause errors in $NH_3$ measurements because the mirror reflectivity is continually monitored and the measurement is compensated using the mirror ringdown time. Interferences to $NH_3$ measurements can be from the sample inlets, for example, due to water condensation or adsorption/desorption effects (e.g., Schwab, 2008; Norman et al., 2009). Such interferences were not quantified but reduced as possibly as we could. PTFE tubing (4.8 mm ID), which is one of the well suited materials for $NH_3$ measurement (Norman et al., 2009), was used to induced ambient air. The length of the tubing was kept as short as possible (about 5 m) to limit the residue time to less than 3 s. The aerosol filter at the inlet was changed every two weeks. Water condensation was avoided. Nevertheless, we cannot exclude the influence from the adsorption and desorption, which can also occur on dry surfaces. However, this influence should be small at our site, where the $NH_3$ concentration

is very high, and cause mainly a lag in the recorded $NH_3$ concentration.

A set of commercial instruments from Thermo Environmental Instruments, Inc. were used to measure $O_3$ (TE 49C), $NO/NO_2/NO_x$ (TE 42CTL), CO (TE 48C), and $SO_2$ (TE 43CTL). All instruments were housed in an air-conditioned room in the observation building at the site. Two parallel inlet tubes (Teflon, 4.8 mm ID$\times$8 m length) were shared by the analyzers. The height of the inlets was 1.8 m above the roof of the building and about 8m above the ground. The inlet residence time was estimated to be less than 5 s (Lin et al., 2009). Zero and span checks were performed weekly on the analyzers of these trace gases to identify possible analyzer malfunctions and zero drifts. Multipoint calibrations of $SO_2$, $NO_x$, CO and $O_3$ analyzers were performed on the instruments at approximately 1-month intervals. Measurement records were saved as 1-min averages After the correction of data on the basis of the multipoint calibrations, hourly average data were calculated and used for the analysis.

An Ambient Ion Monitor (AIM) (URG 9000D Series, USA) was deployed at the site to measure hourly concentrations of water-soluble inorganic components in $PM_{2.5}$ during 15 June–11 August, 2013. A detailed description of performance evaluation of AIM-IC system is reported by Han et al., (2016). Briefly, ambient air was introduced in to the AIM with a 2 meter Teflon coated aluminum pipe and particles larger than 2.5 μm were removed by a cyclone at a flow rate of 3 L/min. A liquid diffusion denuder was used to remove the interfering acidic and basic gases, in combination with a Steam-Jet Aerosol Collector followed by an Aerosol Sample Collector, until the particles can be injected into the ion chromatograph (Hu et al., 2014). The detection limit of $NH_4^+$, $SO_4^{2-}$ and $NO_3^-$ were 0.05 μg m$^{-3}$, 0.04 μg m$^{-3}$ and 0.05 μg m$^{-3}$, respectively. For the AIM, multipoint calibrations were performed weekly by using calibration standard solutions. Acceptable linearity of ions was obtained with an $R^2$ of ≥0.999. The flow rate of the AIM was checked weekly at the sample inlet with a certified flow meter. The flow rate of the AIM was kept at 3 L/min with standard derivation of <1%. Hourly data were obtained for the concentrations of water-soluble inorganic ions in summer 2013.

Meteorological parameters were measured at the site. Air temperature and relative humidity were monitored using a humidity and temperature probe (HMP155, Vaisala, Finland); wind speed and direction were measured using an anemometer (ZQZ-TFD12, Jiangsu Radio Scientific Institute Co., Ltd, China); rainfall was measured using a tilting rain gauge (SL2-1, Tianjin Meteorological Instrument Factory, China). Global radiation observation was made at the site but showed a drift by the end of July,

2013. Instead we use the photolysis rate jNO$_2$ observed using a 2-pi-actinic-flux spectrograph (CCD type, Meteorologie Consult GmbH, Germany) to indicate radiation condition for photochemistry. Hourly meteorological data were calculated from the in-situ measurements and used in this paper. Planetary boundary layer height values at 14:00 were derived from the ERA-Interim data using the Bulk Ricardson number method (Guo et al., 2016; Miao et al., 2017)."

[Figure]

Figure S1 Confirmation of the performance of NH$_3$ analyzer using diluted standard gas (mixture NH3/N2). (a) Instrument response to changed NH$_3$ concentration and stability; (b) repeated multipoint calibrations.

[Figure]

Figure S2. Monthly average diurnal variations of the NO$_2$ photolysis frequency (jNO$_2$) (a) and hourly rainfall (b) observed at Gucheng during June-August 2013.

Han, B., Zhang, R., Yang, W., Bai, Z., Ma, Z., and Zhang, W.: Heavy haze episodes in Beijing during January 2013: inorganic ion chemistry and source analysis using highly time-resolved measurements from an urban site, Sci. Total Environ., *544*, 319-329, 2016.

Lin, W., Xu, X., and Zhang, X.: Characteristics of gaseous pollutants at Gucheng, a rural site southwest of Beijing, J. Geophys. Res., 114, 10339, 2009.

Norman, M., Spirig, C.,Wolff, V., Trebs, I., Flechard, C., Wisthaler, A., Schnitzhofer, R., Hansel, A., and Neftel, A.: Intercomparison of ammonia measurement techniques at an intensively managed grassland site (Oensingen, Switzerland), Atmos. Chem. Phys., 9, 2635–2645, 2009.

Schwab, J.J.: Ambient Gaseous Ammonia: Evaluation of Continuous Measurement Methods Suitable for Routine Deployment, Final Report Prepared for the New York State Energy Research and Development Authority (NYSERDA),Final Report 08-15, New York, October 2008.

Wu, W. S. and Wang, T.: On the performance of a semi-continuous $PM_{2.5}$ sulphate and nitrate instrument under high loadings of particulate and sulphur dioxide. Atmos. Environ. 41, 5442-5451, 2007.

**P 11 section 3.4: Relationship between ammonium and ammonia: This discussion is very brief and limited. In particular after noting previously (and probably correctly) that local ammonia emissions dominate the chemical speciation observed, the authors then infer "NH3 dominates NHx deposition". With the dataset they have they could have performed calculations of deposition vs emission over the 4 month period would have given much more insight, i.e. the process is bidirectional therefore it is uncertain whether any net deposition would occur under the ambient conditions. This is a missed opportunity to explore the atmospheric chemistry and physics of the system.**

**Answer:** Thank you for your valuable comments. Indeed it would be great if we could use our measurements to systematically explore the atmospheric chemistry and physics over the our site and to gain more insight, including the emission and deposition of $NH_3$ and $NH_4^+$. However, we did not observe the emission and deposition of $NH_3$ and $NH_4^+$. The air/surface exchange of NH3 is very complex, bidirectional and highly variable, and is influenced by many factors (e.g., Schrader et al., 2016). Given the parameters we observed, it is hardly possible to obtain robust and quantitative results about the emission vs deposition purely based on available measurements. To well understand the processes of emission, conversion and deposition of NHx species in the polluted NCP region, it is highly needed to design observational and 3D modeling studies. Our in-situ measurements present in this paper can be used as supporting data in future modeling studies.

Schrader, F., Brümmer, C., Flechard, C.R., Wichink Kruit, R.J., van Zanten, M.C., Zöll, U., Hensen, A., and Erisman, J.W.: Non-stomatal exchange in ammonia dry deposition models: comparison of two state-of-the-art approaches, Atmos. Chem. Phys., 16, 13417–13430, 2016.

**Given that there are several thermodynamic models freely available, it would have been useful to explore the dataset against what is predicted by models. What is the ion balance – are**

**dicarboxylic acids needed to explain aerosol neutralisation? (is it neutral?). Is the atmospheric chemistry at the site dominated by thermodynamic equilibrium or is there kinetic limitations on the processes? The authors have missed an opportunity with the dataset to fully understand the chemistry and rely in the results and discussions to discuss the ratios between gas and aerosol partitioning to explain scientific concepts which are known and therefore not surprising.**

**Answer:** Thank you for your suggestions. We have used the thermodynamic equilibrium model (ISORROPIA II) to investigate gas-aerosol partitioning characteristics and compared the modeling results with our measurements. The pH values of aerosol, estimated based on the simulated results, are mostly in the range of 2.5-4.5, with an average of 3.5. While simulated sulfate and nitrate are well comparable with the measurements, simulated ammonium substantially underestimates the observed one, indicating the importance of organic acids in the formation of ammonium. Although we did not measure organic acids in aerosol, the presence of oxalic acid and other low molecular weight dicarboxylic acids in aerosols is often reported (e.g., Hsieh et al., 2007; Kawamura et al., 2010, 2013; Sauerwein and Chan, 2017). There is no doubt about the presence of significant amount of dicarboxylic acids over the North China Plain particularly during summer (Kawamura et al., 2013). Therefore, it is highly possible that neutralizing dicarboxylic acids in aerosol particles contributed significantly to the conversion of ammonia to ammonium. Our results also suggest that the gas-aerosol partitioning at the Gucheng site is dominated by thermodynamic equilibrium. We have added a section about the ISORROPIA II model in the revised manuscript and more text about the simulation results and related discussions as follows:

**"2.3.2 ISORROPIA II model**

Thermodynamic gas-aerosol equilibrium characteristics during summer 2013 were examined using ISORROPIA II model (Fountoukis and Nenes, 2007). ISORROPIA II is a thermodynamic equilibrium model for inorganic gases and aerosols in the atmosphere (available at http://isorropia. eas.gatech.edu). To obtain the best available predictions of aerosol pH, ISORROPIA II, was run in the forward mode with metastable aerosol state salts precipitate once the aqueous phase becomes saturated with respect to salts, which often showed better performance than the stable state solution (solid + liquid) and was commonly applied in previous pH predictions (Liu et al., 2017). The concentrations of the measured $NH_3$ and water-soluble ions in $PM_{2.5}$ were input into the model as total (gas + aerosol) concentrations, along with simultaneously measured relative humidity and temperature data. The bulk particle pH was calculated using the following equation:

$$pH = -\log_{10}\frac{1000H^{+}_{air}}{AWC} \quad (4)$$

where $H^{+}_{air}$ ($\mu g\ m^{-3}$) and AWC ($\mu g\ m^{-3}$) are the ISORROPIA-II predicted equilibrium particle hydronium ion concentration per volume air and aerosol water content, respectively. Evaluation of the

AWC prediction have been reported in previous studies and shows a good performance compared with particle water measurements (Bian et al., 2014; Guo et al., 2015). ”

We have added a new section (Section 3.3 Results from thermodynamic equilibrium simulation) to present our main modeling results and discussed them also other section.

"**3.4 Results from thermodynamic equilibrium simulation**

We have used the thermodynamic equilibrium model ISORROPIA II to investigate gas-aerosol partitioning characteristics. $NO_3^-$, $SO_4^{2-}$ and $NH_4^+$. The model outputs include equilibrium $NO_3^-$, $SO_4^{2-}$, $NH_4^+$. $H_{air}^+$, $HNO_3$, $NH_3$, AWC, etc. As shown in Fig. 5, the modelled $NO_3^-$, $SO_4^{2-}$, $NH_3$ show excellent correlations with the corresponding measurements, but modelled $NH_4^+$ is much worse correlated with the measured one. Modelled $NO_3^-$, $SO_4^{2-}$, and $NH_3$ values agree well with the measurements, while the modelled $NH_4^+$ largely underestimate the measurements. Considering the unbalance between observed NH4+ and the sum of observed $SO_4^{2-}+NO_3^-+Cl^-$, we can confirm that other acids in aerosol particles are important in the conversion of $NH_3$ to $NH_4^+$. These other acids may be oxalic acid and other dicarboxylic acids. Although we did not measure organic acids in aerosol, the presence of oxalic acid and other low molecular weight dicarboxylic acids in aerosols is often reported (e.g., Hsieh et al., 2007; Kawamura et al., 2010, 2013; Sauerwein and Chan, 2017). There is no doubt about the presence of significant amount of dicarboxylic acids over the North China Plain particularly during summer (Kawamura et al., 2013). Therefore, it is highly possible that neutralizing dicarboxylic acids in aerosol particles contributed significantly to the conversion of ammonia to ammonium.

The simulated $HNO_3$ concentrations was $0.9 \pm 1.1$ μg m$^{-3}$, showing a maximum value of 7.41 μg m$^{-3}$ at 11:00 on 19 June 2013. The average diurnal variations of $HNO_3$ and $H_{air}^+$ are shown in Fig. S4. The fine particles were moderately acidic in summer, with an average pH values of 3.5. The pH values of aerosol water, estimated based on the simulated results using equation (4), are mostly in the range of 2.5-4.5, with an average of 3.5. On average, pH is over 3.5 during nighttime and below 3.5 during daytime (Fig. 6). Under the medium acidic conditions and high $NH_3$ concentrations, organic acid like diacids are able to reaction with ammonia to for ammonium. Because we used ISORROPIA-II for inorganic aerosol composition and no organic acids measurements are available, we cannot analyze in detail the role of organic acids though the model performed quite well (Fig. S5). "

[Figure]

Figure 5. Observed and modelled concentrations of NH$_3$, NH$_4^+$, SO$_4^{2-}$ and NO$_3^-$ in summer 2013.

[Figure]

Figure 6. Calculated diurnal variation of pH value of aerosol water.

[Figure]

Figure S5 Correlation of modelled $NH_4^+$ with modelled $SO_4^{2-}$, $SO_4^{2-}+NO_3^-$ and $SO_4^{2-}+NO_3^-+Cl^-$..

We have made another new section (3.6 A case study of a pollution period) to present and discuss the 7-11 August measurement with additional analysis.

"**3.6 A case study of a pollution period**

On several days during the study period, very high $NH_3$ and inorganic $PM_{2.5}$ concentrations were observed. Here make a case study of a pollution period during 7-11 August 2013. Data of gases, major aerosol ions and some key meteorological parameters are presented in Fig. 9. Some other measure and calculated parameters during this period are given in Fig. S6. As shown in Figs. 9 and S6, there was a sharp increase of NOx during the night and early morning of 10 August, followed by that of $NH_3$ (peak value 64 ppb at 03:00. In the meantime, a large peak of AWC occurred and gaseous HNO3 decreased to nearly zero (Fig. S6), suggesting rapid uptake of wet aerosol. This event caused the first largest peak of $[SO_4^{2-}]+[NO_3^-]+[NH_4^{+}]$. After this event NH3 rose again and reached a even higher peak (76.3) shortly before noon of 10 August. This peak of NH3 coincided with a valley of NOx, but the HNO3 level increased and pH value decreased was observed in parallel. A few hours later SO2 showed a large peak and the second largest peak of $[SO_4^{2-}]+[NO_3^-]+[NH_4^{+}]$ occurred. These data show that high $NH_3$ concentration was accompanied by the large increase in concentrations of $SO_4^{2-}$, $NO_3^-$ and $NH_4^+$, confirming that $NH_3$ play an important role in PM mass formation and that gas-particle

conversion occurred when $NH_3$ was available, though $SO_4^{2-}$ partitions to the aerosol phase regard less of $NH_3$ level (Gong et al., 2013). The secondary ions concentrations had similar temporal distributions with slow accumulation and relatively rapid clearing under favourable meteorological conditions. There were good correlation between $NH_3$ with $NH_4^+$, $SO_4^{2-}$ and $NO_3^-$ (R=0.33, 0.27 and 0.49, respectively, with P < 0.01). However, there was also situation when high NH3 did not associate with high $[SO_4^{2-}]+[NO_3^-]+[NH_4^{+-}]$, as indicated by the data around noon of 8 August (Fig. 9). During this case, AWC was extremely low and RH was around 40%. These conditions do not favor heterogeneous reactions.

During 7-11 August 2013, the relationships of the observed $NH_4^+$ versus those of $SO_4^{2-}$, the sum of $SO_4^{2-}$ and $NO_3^-$ and the sum of $SO_4^{2-}$, $NO_3^-$ and $Cl^-$ are presented in Fig. 10. It is known that $(NH_4)_2SO_4$ is preferentially formed and the least volatile, $NH_4NO_3$ is relatively volatile, while $NH_4Cl$ is the most volatile. $NH_4^+$ is thought to be first associated with $SO_4^{2-}$, afterwards, the excess of $NH_4^+$ is with nitrate and chloride (Meng et al., 2015). It is noted that the correlation of $NH_4^+$ with the sum of $SO_4^{2-}$ and $NO_3^-$ (R=0.91, slope=1.23, with P < 0.01) was better than that of $NH_4^+$ with $SO_4^{2-}$ (R=0.80, slope=1.65, with P < 0.01), suggesting that both $SO_4^{2-}$ and $NO_3^-$ were associated with $NH_4^+$. As shown in Fig.10, sulfate and nitrate were almost completely neutralized with most of the data above the 1:1 line. A few scattered data below the 1:1 line may be caused by uncertainties in measurements. Little different was found between the regression slopes of $NH_4^+$ with the sum of $SO_4^{2-}$ and $NO_3^-$ and the sum of $SO_4^{2-}$, $NO_3^-$ and $Cl^-$ due to the very low amount of $NH_4Cl$. In this study, the level of $NH_3$ was high enough to neutralize both $SO_4^{2-}$ and $NO_3^-$, and likely to be form $(NH_4)_2SO_4$ and $NH_4NO_3$. In addition to these substances, it is likely that NH3 also reacted with oxalic acid and other dicarboxylic acid to form ammonium oxalate and other organic ammonium aerosols, as discussed above."

[Figure]

Figure 9. Hourly concentrations of precursor gas and ionic species measured in the pollution episode (a) temporal variations and (b) correlations of $[NH_4^+]$ versus $[SO_4^{2-}]$, $[SO_4^{2-}]+[NO_3^-]$ and $[SO_4^{2-}]+[NO_3^-]+[Cl^-]$ during 7 −11 August 2013.

[Figure]

Figure 10. Correlations between $[NH_4^+]$ and $[SO_4^{2-}]$ (left), $[NH_4^+]$ and $[SO_4^{2-}]+[NO_3^-]$ (middle) and $[NH_4^+]$ and $[SO_4^{2-}]+[NO_3^-]+[Cl^-]$ (right) during 7-11 August 2013.

[Figure]

Figure S6. Time series of predicted fine particle pH, predicted particle water mass, predicted $H_{air}^+$ and measured $NH_3$ and measured inorganic ions during 7-11 August 2013

Hsieh, L.Y., Kuo, S.C., Chen, C.L., Tsai, Y.I.: Origin of low-molecular-weight dicarboxylic acids and their concentration and size distribution variation in suburban aerosol, Atmos. Environ., 41, 6648-6661, 2007.

Kawamura, K., Tachibana, E., Okuzawa, K.,Aggarwal,S.G., Kanaya, Y., and Wang, Z. F.: High abundances of water-soluble dicarboxylic acids, ketocarboxylic acids and α-dicarbonyls in the mountaintop aerosols over the North China Plain during wheat burning season, Atmos. Chem. Phys., 13, 8285–8302, 2013.

Kawamura, K., Barrie, L.A., and Desiree, T.-S.: Intercomparison of the measurements of oxalic acid in aerosols by gas chromatography and ion chromatography, Atmos. Environ., 44, 5316-5319, 2010.

Sauerwein, M. and Chan, C. K.: Heterogeneous uptake of ammonia and dimethylamine into sulfuric and oxalic acid particles, Atmos. Chem. Phys., 17, 6323–6339, 2017

The references are added in the revised manuscript.

Bian, Y. X., Zhao, C. S., Ma, N., Chen, J., and Xu, W. Y.: A study of aerosol liquid water content based on hygroscopicity measurements at high relative humidity in the North China Plain, Atmos. Chem. Phys., 14, 6417-6426, https://doi.org/10.5194/acp-14-6417-2014, 2014.

Bougiatioti, A., Nikolaou, P., Stavroulas, I., Kouvarakis, G., Weber, R., Nenes, A., Kanakidou, M., and Mihalopoulos, N.: Particle water and pH in the eastern Mediterranean: Source variability and implications for nutrient availability, Atmos. Chem. Phys., 16, 4579–4591, 2016.

Fountoukis, C. and Nenes, A.: ISORROPIA II: a computationally efficient thermodynamic equilibrium model for $K^+$-$Ca^{2+}$-$Mg^{2+}$-$NH_4^+$-$Na^+$-$SO_4^{2-}$-$NO_3^-$-$Cl^-$-$H_2O$ aerosols, Atmos. Chem. Phys., 7,

4639–4659, doi:10.5194/acp-7-4639-2007, 2007.

Fountoukis, C., Nenes, A., Sullivan, A., Weber, R., Van Reken, T., Fischer, M., Matias, E., Moya, M., Farmer, D., and Cohen, R. C.: Thermodynamic characterization of Mexico City aerosol during MILAGRO 2006, Atmos. Chem. Phys., 9, 2141–2156, doi:10.5194/acp-9-2141-2009, 2009.

Liu, M., Song, Y., Zhou, T., Xu, Z., Yan, C., Zheng, M., Wu, Z., Hu, M., Wu, Y., and Zhu, T.: Fine particle pH during severe haze episodes in northern China, Geophys. Res. Lett., 44, doi:10.1002/2017GL073210, 2017.

**In the places in the manuscript which use atmospheric chemistry to explain data (e.g. 3.4.2), there are no calculations to check if what is observed is what would be expected under the conditions. Is ozone being lost to the surface or is there a haze which allows aqueous processing in the atmosphere, what might be the role of organics...), what is the surface area of PM (given composition and RH) and hence can N2O5 hydrolysis explain the observations completely? There are lots of questions which are not touched upon, though they are key to understanding the role of NH3.**

**Answer:** It is true we did not calculate the $O_3$ loss, aerosol surface, $N_2O_5$ hydrolysis, etc. Having these results would be very helpful in understanding the role of NH3 in ammonium formation and other aspects of atmospheric chemistry at the site. However, our project was not designed for a "closure study" of gas-aerosol chemistry over the site. Many key parameters were not observed, such as deposition of O3, NH3, etc., concentrations of radical species, organic aerosols, aerosol surface and size distribution, changes in boundary layer, etc. In the absence of these parameters, many assumptions have to be made, which will lead to large uncertainties in the results. We think a comprehensive modeling study is needed to obtain quantitative assessments of all the chemical and physical processes. Such a modeling is beyond the scope of this observation-based study. Nevertheless, we have made some calculations and obtained some semi-quantitative results as answer to the reviewer's questions (Fig. R1). According to Verbeke et al. (2015), the annual mean $O_3$ dry deposition velocity in the NCP region is about 0.3 cm/s. Assuming a doubled deposition velocity (0.6 cm/s) in summer and a constant PBLH of 200 m (this may cause an overestimate during daytime), dry deposition of O3 is estimated to be in the range of 1.5-9 ppb/hr, which is much smaller than the estimated NO titration (37-117 ppb/hr). If reaction of $O_3$ with other gases and uptake by aerosol are neglected, a production rate in the range of 35-127 ppb/hr is required to balance the titration and deposition losses and cause the observed net change from -10 ppb/hr to 12 ppb/hr. Note that photochemical production during nighttime should be zero. Therefore the "production" during nighttime can be considered as source aloft the surface layer.

We have added "Nighttime formation, aerosol uptake and hydrolysis of $N_2O_5$ are highly uncertain as has been pointed out (e.g., Xue et al. 2014). The NOx concentration during nighttime was higher than during daytime, while the $NO_3^-$ level during nighttime was only slightly higher than that during daytime (Fig. R2). By assuming high aerosol surface to mass ratio (33.7 $m^2$/g, Okuda, 2013) and a high uptake coefficient (0.1, Seinfeld and Pandis, 2006), we estimate the nighttime $N_2O_5$ under the conditions over our site to be in the range of about 3-10 ppb, corresponding to a HNO3 production rate of about 1-3 ppb/hr (or 2.6-7.7 $\mu g/m^3$). This rate of $HNO_3$ production would cause an obvious night

production of $NH_4^+$. Indeed we can see increases in the $NH_4^+$ concentration and NHR during night (Fig. R3). However, a more or less accurate estimate of the relative contribution of the night $N_2O_5$ chemistry to $NH_3$ conversion needs to be made in the future."

[Figure]

Figure R1 Estimated budget of surface $O_3$ at Gucheng during summer 2013.

[Figure]

Figure R2 Diurnal variations of $NO_x$ and aerosol $NO_3^-$ at Gucheng during summer 2013.

[Figure]

Figure R3 Average diurnal cycles of NHR, SOR. NOR, gaseous precursors, major water soluble ions, and meteorological factors in summer 2013.

Okuda, T.: Measurement of the specific surface area and particle size distribution of atmospheric aerosol reference materials, Atmos. Environ., 75, 1-5, 2013.

Seinfeld, J. H. and Pandis, S. N.: Atmospheric chemistry and physics: from air pollution to climate change (2nd ed.), Wiley Interscience, New Jersey, 2006.

Verbeke, T., Lathière, J., Szopa, S., and de Noblet-Ducoudré, N.: Impact of future land-cover changes on HNO3 and O3 surface dry deposition, Atmos. Chem. Phys., 15, 13555-13568, 2015.

Xue, L. K., Wang, T., Gao, J., Ding, A. J., Zhou, X. H., Blake, D. R., Wang, X. F., Saunders, S. M., Fan, S. J., Zuo, H. C., Zhang, Q. Z., and Wang, W. X.: Ground-level ozone in four Chinese cities: precursors, regional transport and heterogeneous processes, Atmos. Chem. Phys., 14, 13175–13188, doi:10.5194/acp-14-13175-2014, 2014.

**Having read the paper I am still not sure what the authors want a reader to learn from the gas-particle ratios. I would suggest the authors revise to include pollution/wind rose diagrams to look at the pollution footprint (e.g. ones are available on Open air and other packages), use current thermodynamics and kinetics of the system to see if current models would accurately represent the observations, if not what may be missing?**

**Answer:** We have drawn the Figure of $NH_3$ rose and added the analysis of local source in Section 3.5 (now section 3.7) as follows:

"3.5 Local and long-rang transport source of ammonia and ammonium aerosol

Dependence of the concentrations of $NH_3$ on wind direction at Gucheng is studied to get insight into the distribution of local emission sources around the monitoring site. As shown in Fig. 11, during the sampling period, the prevailing surface winds at Gucheng were northeasterly and southwesterly. High $NH_3$ originated from the southwest sector of the measurement site, which may be due to a local unidentified agricultural or industrial source or transport from the Xushui township, which is approximately 15 km away from Gucheng. Lower $NH_3$ concentrations were observed under winds from other sectors. Since $NH_3$ is either readily converted to $NH_4^+$ or subjected to dry deposition, high concentrations are found only close to the surface and near the emission sources. Previous studies have reported an inverse relationship between ground-level concentrations of trace gases, such as ammonia, and wind speed (Robarge et al., 2002; Lin et al., 2011). Thus, $NH_3$ concentrations might be generally lower at higher wind speeds because of turbulent diffusion.

[Figure]

[Figure]

Figure 11. The average NH$_3$, NH$_4^+$ concentrations and meteorological data roses in different wind sectors during summer 2013."

**Oxidised nitrogen chemistry and the gas-aerosol partitioning dynamic are mentioned in passing but are key to understanding whether NH3 is driving the PM formation or it is a reservoir gas which grows PM when the presence of the other pollutants is there. Biomass burning is mentioned and K and CO as the indicators. With the dataset they could estimate the fraction of PM due to biomass burning, how much of the PM is explained by biomass burning, and does the biomass burning "seed" larger PM events. Finally the discussion and conclusion would be enhanced if some discussion about the impacts and potential solutions to the impacts. What is the evidence that limiting the NH3 emission would improve the air quality - it may well achieve this, but to make the case, evidence or hypothesis is needed to back the statements up.**

Answer: We have made simulations using the thermodynamic equilibrium model ISORROPIA II. The simulated results indicate that the strong acids (H$_2$SO$_4$, HNO$_3$ and HCl) are well balanced by NH3 (Fig. R4). However, the correlation between observed NH$_4^+$ and the sum of observed SO$_4^{2-}$, NO$_3^-$ and Cl$^-$ (Fig. R5) suggests that the neutralization of the strong acids explain 56% of the observed NH4+. In other words, nearly 44% of the observed NH$_4^+$ was due to the presence of other acids in aerosol particles. As mentioned above these acids may be oxalic acid and other dicarboxylic acids. The level of NH$_3$ at Gucheng in summer 2013 was very high. Under the NH$_3$-rich condition acid neutralization was easily achieved. As shown in Fig. R3, the average NHR values were around 30%, indicating NH$_3$ was not a factor limiting the PM formation. Some recent studies suggest that the formation of sulfate in aerosol can been largely enhanced by NO$_2$ oxidation under higher pH values (e.g., Xie et al., 2015; Cheng et al., 2016). Higher pH in aerosol water can be caused by high dust aerosol or NH3. The pH values in aerosol water at Gucheng during our observations was estimated to mostly range from 2.5 to 4.5, based on the ISORROPIA modeling results. Our data do not indicate any increase in sulfate content with the increased NO$_2$. Therefore, high NH$_3$ at our site was not driving more PM formation and it served as a reservoir gas to neutralize acids present in aerosol and gas phases.

Open biomass burning occurred occasionally in the NCP region during our campaign though open burning of agricultural wastes had been prohibited. We observed no significant open fires nearby Gucheng. However, fires in the surrounding areas may impact the measurements at the site. The fire maps show that open burning occurred mainly in June and July, which is usually the period of burning wheat straw. During a few days around middle and end June, we observed relatively higher

concentrations of CO and aerosol $K^+$, which may be resulted from biomass burning. The K+ concentration is fairly well correlated with the $PM_{2.5}$ concentration (Fig. R7), suggesting an impact of biomass burning on particle pollution. The concentration of $K^+$ is not correlated with those of Na and Ca. Therefore, the observed $K^+$ in aerosol should be mainly from biomass burning. Although the slope of the $K-PM_{2.5}$ regression line is small (0.026), the total contribution of biomass burning to $PM_{2.5}$ may be much larger than a few percentages considering carbonaceous aerosols and other species emitted by biomass burning (Cheng et al., 2013).

[Figure]

Figure R4 Correlation of simulated $NH_4^+$ with simulated $SO_4^{2-}$, $NO_3^-$ and $Cl^-$.

[Figure]

Figure R5 Correlation of observed $NH_4^+$ with observed $SO_4^{2-}$, $SO_4^{2-}+NO_3^-$ and $SO_4^{2-}+NO_3^-+Cl^-$..

[Figure]

Figure R6 Open fires in the NCP region during June-September, 2013. The red triangels indicate the position of the Gucheng site. Data source: https://firms.modaps.eosdis.nasa.gov/firemap/.

[Figure]

Figure R7 K+ in aerosol and CO concentrations observed at Gucheng during summer 2013.

[Figure]

Figure R8 Correlation between $K^+$ and $PM_{2.5}$ concentrations.

Cheng, Y., Engling, G., He, K.-B., Duan, F.-K., Ma, Y.-L., Du, Z.-Y., Liu, J.-M., Zheng, M., and Weber, R. J.: Biomass burning contribution to Beijing aerosol, Atmos. Chem. Phys., 13, 7765–7781, 2013

Cheng, Y., Zheng, G., Wei, C., Mu, Q., Zheng, B., Wang, Z., Gao, M., Zhang, Q., He, K., Carmichael, G., Pöschl, U., Su, H.: Reactive nitrogen chemistry in aerosol water as a source of sulfate during haze events in China. Sci. Adv. 2, e1601530, 2016.

Xie, Y., Ding, A., Nie, W., Mao, H., Qi, X., Huang, X., Xu, Z., Kerminen, V.-M., Petäjä, T., Chi, X., Virkula, A., Boy, M., Xue, L., Guo, J., Sun, J., Yang, X., Kulmala, M., Fu, C.: Enhanced sulfate formation by nitrogen dioxide: Implications from in-situ observations at the SORPES Station, J. Geophys. Res. 120, 12679-12694, 2015.

**Minor corrections:**

**P1 Line 22 The observation that NH3 drives NH4 content of PM is not new, so I do not think the word " suggesting" is appropriate**

**Answer:** We have changed the word "suggesting" to "reflecting".

**1 P1 Line 24: This is a percentage not a ratio.**

**Answer:** We have revised the percentage to the ratio according to the reviewer's comments.

**P1 Line 25: use previous NCP abbreviation**

**Answer:** We have added "in the NCP" in Line 25 according to the reviewer's comments.

**P2 line 15: Actually most atmospheric chemistry text books discuss this, would cite them rather than research papers.**

**Answer:** We have revised the sentence. "Some studies have suggested that reducing $NH_3$ concentrations could be an effective method for alleviating secondary inorganic $PM_{2.5}$ pollution."

**P5, line 19: asymmetric errors would be more appropriate given that one cannot have negative**

**concentrations.**

**Answer:** We have decided to use the range instead of standard deviation. The sentence reads now "During 15 May–25 September 2013, the average concentrations (ranges) of NH3, SO2 and NOx were 36.2 (0-862.9), 5.0 (0-86.8) and 15.4 (2.7-67.7) ppb, respectively.".

**P25 Figure 2: given that it rained during the 4 months, why does the RH never go above 90%?**

**Answer:** To accurately measure high RH is still difficult so that the absolute errors associated the high RH values may be large. The measured RH does include some values over 90%, with the maximum of 93%. There might be negative bias. On the other hand, raining does not necessarily mean the surface air is saturated with water.

**P27: for the PM composition it would be useful to have them as stacked so that one can see the variation of composition through time References: There are not many references from 2015 and 2016 despite many papers being published on this subject area. I would suggest the authors review the recent literature.**

**Answer:** The figure on this page has been redrawn. The PM compositions are shown as stacked. Global radiation and wind speed data are removed from the figure. Instead measurements of O3 and jNO2 are shown to provide information about photochemistry. We have reviewed the some recent papers (Guo et al., 2015; Han et al., 2016; Sudheer and Rengarajan, 2015; Tang et al., 2016; Wen et al., 2015; Xu et al., 2017; Zhao et al., 2016) and cited them in the revised manuscript.

[Figure]

Figure R9 Hourly concentrations of gaseous and ionic species and measurements of air temperature,

RH and jNO2 observed during the pollution episode 7-11 August 2013.

Guo, H., Xu, L., Bougiatioti, A., Cerully, K. M., Capps, S. L., Hite Jr., J. R., Carlton, A. G., Lee, S.-H., Bergin, M. H., Ng, N. L., Nenes, A., and Weber, R. J.: Fine-particle water and pH in the southeastern United States, Atmos. Chem. Phys., 15, 5211-5228, https://doi.org/10.5194/acp-15-5211-2015, 2015.

Han, B., Zhang, R., Yang, W., Bai, Z., Ma, Z., and Zhang, W.: Heavy haze episodes in Beijing during January 2013: inorganic ion chemistry and source analysis using highly time-resolved measurements from an urban site, Sci. Total Environ., 544, 319-329, 2016.

Sudheer, A. K., and Rengarajan, R.: Time-resolved inorganic chemical composition of fine aerosol and associated precursor gases over an urban environment in western India: Gas-aerosol equilibrium characteristics[J]. Atmospheric Environment, 109:217-227, 2015.

Tang, X., Zhang, X.S., Ci, Z.J., Guo, J and Wang, J. Q.: Speciation of the major inorganic salts in atmospheric aerosols of Beijing, China: Measurements and comparison with model. Atmos. Environ., 133:123-134, 2016.

Wen, L., Chen, J., Yang, L., Wang, X., Xu, C., & Sui, X., et al.: Enhanced formation of fine particulate nitrate at a rural site on the north china plain in summer: the important roles of ammonia and ozone. Atmos. Environ., 101, 294-302, 2015.

Xu, W., Song, W., Zhang, Y., Liu, X., Zhang, L., Zhao, Y., Liu, D., Tang, A., Yang, D., Wang, D., Wen, Z., Pan, Y., Fowler, D., Collett Jr., J. L., Erisman, J. W., Goulding, K., Li, Y., and Zhang, F.: Air quality improvement in a megacity: implications from 2015 Beijing Parade Blue pollution control actions, Atmos. Chem. Phys., 17, 31-46, https://doi.org/10.5194/acp-17-31-2017, 2017.

Zhao, M., Wang, S., Tan, J., Hua, Y., Wu, D., and Hao, J.: Variation of Urban Atmospheric Ammonia Pollution and its Relation with PM2.5 Chemical Property in Winter of Beijing, China. Sci. Total Environ., 16(6) , 1378-1389, 2016.

**Response to Reviewer #2's comments**

**Anonymous Reviewer #2**

**This manuscript presents a comprehensive ambient measurement dataset, including various trace gases and particulate species, for over four months at a rural site in the North China Plain (NCP). Ammonia (NH3) is the focus of this study for its role in the formation of secondary inorganic aerosols, which accounts for a major fraction of PM2.5 in NCP. The hourly resolution, higher than many of the previous ambient ammonia measurements, enables detailed studies on individual pollution events and the diurnal variations. However, I hope the authors can take better advantage of this dataset, and go deeper into Atmospheric Chemistry and Atmospheric Physics, as indicated by the journal name. For example, this work aims to understand the impact of ammonia on secondary ammonium aerosols (page 1, line 20), facilitate developing future ammonia emission control policies (page 1, line 32), and examine the sources of ammonia and ammonium and their chemical conversion mechanism (page 3, line 8). These are all important issues, but I am not convinced that this article has advanced our current knowledge and understanding about these issues after reading it.**

**Answer:** We thank your comments and suggestions. We have made additional data analysis and revised the manuscript according to the comments and suggestions by both reviewers. To gain more insight into the role of ammonia in the formation of secondary inorganic aerosol, simulations were made using the thermodynamic equilibrium model ISORROPIA II. The measurements were used as input of model to simulate the variations of the components in gas, liquid and solid phases, which are useful in the investigation of the gas-aerosol equilibrium characteristics.

**Major comments**

**1. It takes a significant part of this manuscript to explain the observed concentrations. However many of the explanations are qualitative and even speculative. Further quantitative evidences are needed. To name a few:**

**Page 8, lines 5-6, "the monthly concentration of SO2, NOx, and CO in July and August decreased because of raid photochemical reduction, additional removal by rainfall, and excellent vertical mixing." What are the evidences of photochemistry, wet scavenging, and vertical mixing?**

**Answer:** We have added the analysis of some related meteorological conditions and show figures with measurements of rainfall and the $NO_2$ photolysis frequency ($jNO_2$) during June-August 2013 as supplementary materials. Changes are made to text as follows:

"The monthly concentrations of $SO_2$, $NO_x$, and CO in July and August decreased compared to those in June. In addition to less influences from biomass burning, meteorological conditions were also in favor of lowering the concentrations of these gases. Figure S2 shows the monthly average diurnal variations of jNO2 and the time-series of hourly rainfall during June-August 2013. As can be seen, the average jNO2 increased from June to August, indicating better conditions for photochemical reduction in July and August. There was also a slight increase in rainfall from June to August, which may promote removal of the pollutants."

[Figure]

Figure S2. Monthly average diurnal variations of the NO$_2$ photolysis frequency ($j$NO$_2$) (a) and hourly rainfall (b) observed at Gucheng during June-August 2013.

**Page 8, lines 8-9, the ozone was highest in June because of "photochemical production, intense burning of biomass, and transport of regional pollution". What are the evidences of more photochemistry, biomass burning, and regional transport in June? Shouldn't July have larger photochemical rates?**

**Answer:** After careful consideration we think the statement should be revised. The sentence is changed to "For the secondary pollutant O$_3$, the highest concentration was observed in June. This is consistent with previous results from Gucheng (Lin et al., 2009) and should be related with the annual maximum of background O$_3$ in the NCP, which occurs in June (Lin et al., 2008; Ding et al., 2008). "

Ding, A. J., Wang, T., Thouret, V., Cammas, J.-P., and Nédélec, P.: Tropospheric ozone climatology over Beijing: analysis of aircraft data from the MOZAIC program, Atmos. Chem. Phys., 8, 1-13, 2008.

Lin, W., Xu, X., Zhang, X., and Tang, J.: Contributions of pollutants from North China Plain to surface ozone at the Shangdianzi GAW Station, Atmos. Chem. Phys., 8, 5889–5898, 2008.

Lin, W., Xu, X., and Zhang, X.: Characteristics of gaseous pollutants at Gucheng, a rural site southwest of Beijing, J. Geophys. Res., 114, 10339, 2009.

**Page 8, lines 26-27, the downward mixing of the residual layer containing higher ammonia concentration could lead to an increase of ammonia in the morning. This would require a very large pool of ammonia in the residual layer. Why did it not happen in other months?**

**Answer:** Indeed, the interpretation about the morning peak is mainly based on the opinions in cited references, which are mostly speculations. We think more investigations are necessary to be able to clearly and quantitatively explain the morning peak phenomenon. We have revised the 2nd and 3rd paragraphs in section 3.2.2 as follows:

"The morning peak of NH$_3$ was also observed elsewhere and could be resulted from emissions from fertilized soils and plant stomata, evaporation of dew, and human sources, as well as mixing down of

ammonia from the residual layer (Trebs et al., 2004; Norman et al., 2009; Bash et al., 2010; Ellis et al., 2011). Figure 3b reveals that the relative humidity (90%-89%) and temperature (21.5-22.1 ℃) remained relatively constant before 06:00, but increased later in the morning. The increasing temperature can heat the earth's surface and vegetation leaves and reduce the RH, potentially leading to evaporation of NH3 from soil and plants and volatilization of ammonium aerosol (Trebs et al., 2004; Norman et al., 2009; Ellis et al., 2011), which may increase $NH_3$ concentrations in the morning. When the emission was occurring into a shallow boundary layer, $NH_3$ increase would be more prominent. In addition, the morning rise might also be due to the breakup of the nocturnal boundary layer. During the sampling period, the majority of peaks of ammonia over 50 ppb occurred at night, which were attribute to local emissions, such as from agricultural activity, into a shallow nocturnal boundary layer. It was supposed by Ellis et al. (2011) that the downward mixing of air containing higher $NH_3$ from the residual layer could lead to an increase of surface $NH_3$ after the breakup of the nocturnal boundary layer. "

[Figure]

Figure 3. Diurnal variation (a) NH$_3$ and (b) meteorological parameters during the sampling period.

Bash, J. O., Walker, J. T., Katul, G. G., Jones, M. R., Nemitz, E., and Robarg, W. P.: Estimation of In-Canopy Ammonia Sources and Sinks in a Fertilized Zea mays Field, Environ. Sci. Tech., 44, 1683-1689, 2010.

Ellis, R. A., Murphy, J. G., Markovic, M. Z., VandenBoer, T. C., Makar, P. A., Brook, J., and Mihele, C.: The influence of gas-particle partitioning and surface-atmosphere exchange on ammonia during BAQS-Met, Atmos. Chem. Phys., 11, 133-145, 2011.

Norman, M., Spirig, C.,Wolff, V., Trebs, I., Flechard, C., Wisthaler, A., Schnitzhofer, R., Hansel, A.,

and Neftel, A.: Intercomparison of ammonia measurement techniques at an intensively managed grassland site (Oensingen, Switzerland), Atmos. Chem. Phys., 9, 2635–2645, 2009.

Schwab, J.J.: Ambient Gaseous Ammonia: Evaluation of Continuous Measurement Methods Suitable for Routine Deployment, Final Report Prepared for the New York State Energy Research and Development Authority (NYSERDA),Final Report 08-15, New York, October 2008.

Trebs, I., Meixner, F. X., Slanina J., Oties, R. P., and Andreae, M. O.: Real-time measurements of ammonia, acidic trace gases and water-soluble inorganic aerosol species at a rural site in the Amazon Basin, Atmos. Chem. Phys., 4, 967–987, 2004.

**Page 9, line 5, the author explains the earlier ammonia morning peak in July by increased emissions. Further evidence?**

**Answer:** We have no direct evidence of emissions. However, the Gucheng site is an experiment station for agrometeorological studies. Corn is the main crop in the station area and nearly all the agricultural areas in the surrounding. According the climate in the NCP, corn is planted around the middle of June and grows rapidly in July. Therefore, July is the key period for the application of nitrogen fertilizers like urea. For example, during last ten days of July 2013, 225-300 kg of urea were applied per hectare of station area (Meng et al., 2015), causing huge NH3 spikes during the end of July (Fig. 2). In addition, the highest nighttime temperature in July (Fig. 3b) could promote the soil emission of $NH_3$ and the relatively lower wind speed (Fig. 3b) and lowest PBLH (Fig. S3) in July was in favor of the accumulation of $NH_3$ in surface air. We have revised the 4th and 5th paragraphs in section 3.2.2 as follows:

"From Fig. 3a, it can be seen that in July the $NH_3$ level was the highest and peaked earliest. One reason for this might be the increased emissions of local agricultural $NH_3$ sources in July compared with those in June, August, and September. On the average, the level NH3 in July had a maximum nighttime increase (20.0 ppb from 20:00 to 06:00), which is much large than those in June (5.2 ppb), August (9.9 ppb) and September (1.8 ppb). The early morning increase of $NH_3$ in July started from a much higher level than in other months, resulting a earliest $NH_3$ peak in July.

There is no direct evidence of increased agricultural $NH_3$ emission in July. However, the Gucheng site is an experiment station for agrometeorological studies. Corn is the main crop in the station area and nearly all the agricultural areas in the surrounding. According the climate in the NCP, corn is planted around the middle of June and grows rapidly in July. Therefore, July is the key period for the application of nitrogen fertilizers like urea. As mentioned above, the urea application in the station on 20 July 2013 and a precipitation process afterwards caused huge $NH_3$ spikes during the end of July (Fig. 2b). In addition, the highest nighttime temperature in July (Fig. 3b) could promote the soil emission of $NH_3$, and the relatively lower wind speed (Fig. 3b) and lowest PBLH (Fig. S3) in July was in favor of the accumulation of $NH_3$ in surface air.

In summary, ambient $NH_3$ at Gucheng showed interesting diurnal cycles, which look significantly different in different summer months. We believe the interplay of some processes, such as emissions from agricultural sources, meteorological conditions (temperature, relativity humidity, wind speed, and BHL height, etc.) as well as chemical conversion are important in the determination of levels and

diurnal patterns of NH$_3$ at the site. Whether or not these processes are all important in the morning variation of NH$_3$? How important are they? And what makes the difference in the peaking time and concentration of NH$_3$ in different months? These are questions to be answered in the future."

[Figure]

Figure S3. The monthly planetary boundary layer heights at 14:00 during 2013 at Gucheng.

Guo, J., Miao, Y., Zhang, Y., Liu, H., Li, Z., Zhang, W., He, J., Lou, M., Yan, Y., Bian, L., and Zhai, P.: The climatology of planetary boundary layer height in China derived from radiosonde and reanalysis data, Atmos. Chem. Phys., 16, 13309-13319, 2016.

Miao, Y., Guo, J., Liu, S., Liu, H., Li, Z., Zhang, W., and Zhai, P.: Classification of summertime synoptic patterns in Beijing and their associations with boundary layer structure affecting aerosol pollution, Atmos. Chem. Phys., 17, 3097-3110, 2017.

**2. A high observed concentration can always be explained by more emission, less mixing, or less removal. I think a publication in ACP should go beyond reporting the concentrations of these short-lived species, as the concentrations are highly variable. This study used the ratio between ammonium and NHx to infer the gas-particle conversion of ammonia. However the ammonium and ammonia may be from different sources, where ammonium is formed in the city with NOx and SO2, and ammonia is emitted locally. In other words, what if NHx and NH3 are decoupled?**

**Answer:** We have made simulations using the ISORROPIA II model and analyzed the model results together measurements. We think we have gained more insight than before but also acknowledge that there are limitations in our observation, modeling and data analysis. Some open issues remain to be addressed in future studies.

The ratios NHR (NH$_4^+$/NHx), SOR (SO$_4^{2-}$/(SO$_4^{2-}$+SO$_2$) and NOR (NO$_3^-$/(NO$_3^-$+NOx)) were calculated as measures of chemical conversion of NH$_3$, SO$_2$ and NOx. You are right that NHx and NH3 may be decoupled. Sources of NH$_3$, SO$_2$ and NOx may be dislocated. The lifetimes of these gases are different and hence the dispersion areas. Fine aerosol particles may travel much longer than the precursor gases. In the real situation we always observe gases and aerosols originating both from cities and from rural areas, emitted by different sources, and chemically produced. Wherever we measure, we measure is a mixture impacted by different sources from locations and processes. In this sense, we should not attribute our results only to local impacts or local situation. When our observation at a site

covers a longer period, our results should be applicable to areas in varying size.

**At page 10, line 13, it is summarized that "This observation emphasizes the important role of NH3 in the formation of secondary SO4, NO3 and NH4 aerosols, which should be further explored…". The title of this manuscript is about the role of ammonia on secondary inorganic aerosols, but what exactly is this role? It is not satisfying to only know it is important and needs further exploration.**

**Answer:** We have deleted this sentence. New results from model simulation and data analysis are added and discussed in the revised manuscript. In particular, we have added two new sections:

"**3.4 Results from thermodynamic equilibrium simulation**

We have used the thermodynamic equilibrium model ISORROPIA II to investigate gas-aerosol partitioning characteristics. $NO_3^-$, $SO_4^{2-}$ and $NH_4^+$. The model outputs include equilibrium $NO_3^-$, $SO_4^{2-}$, $NH_4^+$. $H^+_{air}$, $HNO_3$, $NH_3$, AWC, etc. As shown in Fig. 5, the modelled $NO_3^-$, $SO_4^{2-}$, $NH_3$ show excellent correlations with the corresponding measurements, but modelled $NH_4^+$ is much worse correlated with the measured one. Modelled $NO_3^-$, $SO_4^{2-}$, and $NH_3$ values agree well with the measurements, while the modelled $NH_4^+$ largely underestimate the measurements. Considering the unbalance between observed NH4+ and the sum of observed $SO_4^{2-}+NO_3^-+Cl^-$, we can confirm that other acids in aerosol particles are important in the conversion of $NH_3$ to $NH_4^+$. These other acids may be oxalic acid and other dicarboxylic acids. Although we did not measure organic acids in aerosol, the presence of oxalic acid and other low molecular weight dicarboxylic acids in aerosols is often reported (e.g., Hsieh et al., 2007; Kawamura et al., 2010, 2013; Sauerwein and Chan, 2017). There is no doubt about the presence of significant amount of dicarboxylic acids over the North China Plain particularly during summer (Kawamura et al., 2013). Therefore, it is highly possible that neutralizing dicarboxylic acids in aerosol particles contributed significantly to the conversion of ammonia to ammonium.

[revised manuscript text omitted]

and other organic ammonium aerosols, as discussed above."

[Figure]

Figure 9. Hourly concentrations of precursor gas and ionic species measured in the pollution episode (a)

temporal variations and (b) correlations of $[NH_4^+]$ versus $[SO_4^{2-}]$, $[SO_4^{2-}]+[NO_3^-]$ and

$[SO_4^{2-}]+[NO_3^-]+[Cl^-]$ during 7 −11 August 2013.

[Figure]

Figure 10. Correlations between [NH₄⁺] and [SO₄²⁻] (left), [NH₄⁺] and [SO₄²⁻]+[NO₃⁻] (middle) and [NH₄⁺] and [SO₄²⁻]+[NO₃⁻]+[Cl⁻] (right) during 7-11 August 2013.

[Figure]

Figure S6. Time series of predicted fine particle pH, predicted particle water mass, predicted $H_{air}^+$ and measured NH₃ and measured inorganic ions during 7-11 August 2013

**Specific comments**

**Page 1, line 29: please define "transport of air mass from the North China Plain region", as the site itself is in the middle of NCP.**

**Answer:** Yes, the site is located in the middle of NCP. The concentrations of pollutant levels at Gucheng site are not only driven by local sources but also affected by long range transport. We have changed the title of Section 3.5 to "Long rang transport and local source of ammonia and ammonium" and revised the text as follows:

"Dependence of the concentrations of NH₃ on wind direction at Gucheng is studied to get insight into the distribution of local emission sources around the monitoring site. As shown in Fig. 11, during

the sampling period, the prevailing surface winds at Gucheng were northeasterly and southwesterly. High $NH_3$ originated from the southwest sector of the measurement site, which may be due to a local unidentified agricultural or industrial source or transport from the Xushui township, which is approximately 15 km away from Gucheng. Lower $NH_3$ concentrations were observed under winds from other sectors. Since $NH_3$ is either readily converted to $NH_4^+$ or subjected to dry deposition, high concentrations are found only close to the surface and near the emission sources. Previous studies have reported an inverse relationship between ground-level concentrations of trace gases, such as ammonia, and wind speed (Robarge et al., 2002; Lin et al., 2011). Thus, $NH_3$ concentrations might be generally lower at higher wind speeds because of turbulent diffusion.

[Figure]

Figure 11. The average $NH_3$, $NH_4^+$ concentrations and meteorological data roses in different wind sectors during summer 2013."

Lin, W., Xu, X., Ge, B., and Liu, X.: Gaseous pollutants in Beijing urban area during the heating period 2007-2008: variability, sources, meteorological and chemical impacts, Atmos. Chem. Phys., 11, 8157-8170, 2011.

Robarge, W. P., Walker, J. T., McCulloch, R. B., and Murray, G.: Atmospheric concentrations of ammonia and ammonium at an agricultural site in the southeast United States, Atmos. Environ., 36, 16611-1674, 2002.

**Page 3, line 22: how large is the "surrounding area" that impacts the measurements of this site?**
**Answer:** The site is situated in the middle of a large agricultural region with many villages. The information of Gucheng site in details can be found in Lin et al. (2009). According to the maps from Lin et al. (2009) shown below, the Gucheng site is surrounded by farms, dense villages/towns, and the transportation network in the NCP. The accurate size of the surrounding area that really impacts the measurements at Gucheng is not easy to define and varies with meteorological condition, particularly

wind speed. One can do footprint analysis by setting criteria, but this is out of the scope of this paper.

[Figure]

**Figure 1.** Location of Gucheng site and the topography of the surrounding region (a) from Google Earth imagery (copyright Google Inc., used with permission) and (b) from NASA satellite map (Community Landsat7 visible color; http://worldwind.arc.nasa.gov).

**Section 2.2: what's the response time of the Los Gatos instrument? What is the concentration and accuracy of the calibration gas?**

**Answer:** We have added information revised the related text as follows:

"Ambient $NH_3$ was measured using an ammonia analyzer (DLT-100, Los Gatos Research, USA), which utilize a unique laser absorption technology called Off-Axis Integrated Cavity Output Spectroscopy (OA-ICOS). The analyzer has a precision of 0.2 ppb at 100 sec average and a maximum drift of 0.2 ppb over 24 hrs. The response time of the analyzer is less than 2 s (with optional external N920 vacuum pump). During the campaign, $NH_3$ data were recorded as 100-s average. In principle, the NH3 analyzer does not need external calibration, because the measured fractional absorption of light at an ammonia resonant wavelength is an absolute measurement of the ammonia density in the cell (Manual of Economical Ammonia Analyzer - Benchtop Model 908-0016, Los Gatos Research). However, we confirmed the good performance of the NH3 analyzer using a reference gas mixture $NH_3/N_2$ (Scottgas, USA) traceable to US National Institute for Standards and Technology (NIST). The reference gas of NH3 (25.92 ppm with an accuracy of ±2%) was diluted to different concentrations using zero air and supplied to the analyzer and a sequence with 5 points of different $NH_3$ concentrations (including zero) were repeated for several times to check the performance of the analyzer. As shown in Fig. S1, the analyzer followed rapidly to changes of the $NH_3$ concentration,

produced stable response under stabilized NH$_3$ concentrations, and repeated accurately (within the uncertainty) the supplied NH$_3$ concentrations. The NH$_3$ analyzer contains an internal inlet aerosol filter, which was cleaned before our campaign. Nevertheless, some very fine particles can deposit on the mirrors of the ICOS cell, leading to gradual decline in reflectivity. However, slight mirror contamination does not cause errors in NH$_3$ measurements because the mirror reflectivity is continually monitored and the measurement is compensated using the mirror ringdown time. Interferences to NH$_3$ measurements can be from the sample inlets, for example, due to water condensation or adsorption/desorption effects (e.g., Schwab, 2008; Norman et al., 2009). Such interferences were not quantified but reduced as possibly as we could. PTFE tubing (4.8 mm ID), which is one of the well suited materials for NH$_3$ measurement (Norman et al., 2009), was used to induced ambient air. The length of the tubing was kept as short as possible (about 5 m) to limit the residue time to less than 3 s. The aerosol filter at the inlet was changed every two weeks. Water condensation was avoided. Nevertheless, we cannot exclude the influence from the adsorption and desorption, which can also occur on dry surfaces. However, this influence should be small at our site, where the NH$_3$ concentration is very high, and cause mainly a lag in the recorded NH$_3$ concentration. "

[Figure]

Figure S1 Confirmation of the performance of NH3 analyzer using diluted standard gas (mixture NH3/N2). (a) Instrument response to changed NH3 concentration and stability; (b) repeated multipoint calibrations.

Norman, M., Spirig, C.,Wolff, V., Trebs, I., Flechard, C., Wisthaler, A., Schnitzhofer, R., Hansel, A., and Neftel, A.: Intercomparison of ammonia measurement techniques at an intensively managed grassland site (Oensingen, Switzerland), Atmos. Chem. Phys., 9, 2635–2645, 2009.

Schwab, J.J.: Ambient Gaseous Ammonia: Evaluation of Continuous Measurement Methods Suitable for Routine Deployment, Final Report Prepared for the New York State Energy Research and

Development Authority (NYSERDA),Final Report 08-15, New York, October 2008.

**Page 6, line 3: please define "human activity" as it seems a very broad concept.**

**Answer:** We have changed "human activity" to "human excrement and waste disposal".

**Page 6, line 5: there is no "Zhng et al., 2010" in the reference list.**

**Answer:** This was one of the typos. We have corrected it to **"Zhang et al., 2010".**

**Page 6, line 8-9: please clarify how can these results "be used in improving NH3 emission inventory and making future emission control policies".**

**Answer:** We have revised our expression as follows:

"In recent year a few publications about China's national and regional emission inventories of $NH_3$ (e.g., Zhou et al., 2015; Xu et al., 2015, 2016; Kang et al., 2016). However, these inventories are based on bottom-up studies, subject to substantial uncertainties in spatial and temporal variations of $NH_3$ emissions. Ground based observations of $NH_3$ have been sparse. Our measurements, together with others, can be used for validating and constraining models that use bottom-up inventories, and hence help to reveal potential bias in $NH_3$ emission inventory."

Kang, K., Liu, M., Song, Y., Huang, X., Yao, H., Cai, X., Zhang, H., Kang, L., Liu, X., Yan, X., He, H., Zhang, Q., Shao, M., and Zhu, T.: High-resolution ammonia emissions inventories in China from 1980 to 2012, Atmos. Chem. Phys., 16, 2043–2058, 2016.

Xu, P., Zhang, Y., Gong, W., Hou, X., Kroeze, C., Gao, W.,and Luan, S: An inventory of the emission of ammonia from agricultural fertilizer application in China for 2010 and its high-resolution spatial distribution, Atmos. Environ., 115, 141-148, 2015.

Xu, P., Liao, Y. J., Lin, Y. H., Zhao, C. X., Yan, C. H., Cao, M. N., Wang, G. S., and Luan, S. J.: High-resolution inventory of ammonia emissions from agricultural fertilizer in China from 1978 to 2008, Atmos. Chem. Phys., 16, 1207–1218, 2016.

Zhou, Y., Cheng, S., Lang, J., Chen, D., Zhao, B., Liu, C., Xu, R., and Li, T.: A comprehensive ammonia emission inventory with high-resolution and its evaluation in the Beijing-Tianjin-Hebei (BTH) region, China, Atmos. Environ., 106, 305-317, 2015.

**Figure 2: I understand ammonia is shown in log scale because its concentration spanned three orders of magnitude. However I suggest add a sub plot with linear scale so that the variability is comparable at different concentration levels and the individual spikes from pollution episodes are clearer.**

**Answer:** Thank you for your suggestion. We have redrawn Figure 2 as shown below:

[Figure]

Figure 2. Time series of hourly data of NH$_3$, other trace gases and meteorological parameters measured during the sampling period (a) and a blow-up of the period with extremely high NH$_3$ values during 27-31 July 2013.

**Page 7, line 6: where the urea was applied and how large was the applied area?**
**Answer:** We have added the required information and revised the text as follows:
"The Gucheng station has a farmland of 8.67 hectares. The observation period was in the time of the wheat harvest and corn seeding and growing. Corn was sown and fertilized with about 600 kg of fertilizer per hectare in late June. On 20 July corn was additionally fertilized with 225 to 300 kg of urea per hectare. After this fertilization, there was a raining period. The NH$_3$ concentration increased rapidly on the seventh day after the urea application on 20 July, peaking during the 27–30 July period (Fig. 2b)."

**Page 7, line 29: what are these "trace gases"?**
**Answer:** We have added "such as NO$_x$ and CO"at the end of "trace gases" to in our revised version.

**Page 8, lines 14-15 and page 8, lines 30: these two sentences seem contradict each other.**

**Answer:** We have deleted the paragraph there.

**Page 10, line 1: higher NO3 level than what?**

**Answer:** We have deleted this sentence and changed the sentence before to " On the other hand, $NH_3$ was more efficient in summer to react with $SO_2$ to form $(NH_4)_2SO_4$. The average concentration of $NO_3^-$ in $PM_{2.5}$ was 11.3 ±9.1 μg m$^{-3}$. The highest value of 109.3 μg m$^{-3}$ was observed at 14:00 on 22 June 2013 at the highest RH (93%) and AWC (910 μg m$^{-3}$)".

**Page 10, line 20-25: I suggest add a figure showing the slope and correlations. The SO4 should be normalized with its number of charge. What is the evidence for the existence of NH4HSO4?**

**Answer:** We have added a figure to show the correlations as you suggested and revised the text accordingly.

" During 7-11 August 2013, the relationships of the observed $NH_4^+$ versus those of $SO_4^{2-}$, the sum of $SO_4^{2-}$ and $NO_3^-$ and the sum of $SO_4^{2-}$, $NO_3^-$ and $Cl^-$ are presented in Fig. 10. It is known that $(NH_4)_2SO_4$ is preferentially formed and the least volatile, $NH_4NO_3$ is relatively volatile, while $NH_4Cl$ is the most volatile. $NH_4^+$ is thought to be first associated with $SO_4^{2-}$, afterwards, the excess of $NH_4^+$ is with nitrate and chloride (Meng et al., 2015). It is noted that the correlation of $NH_4^+$ with the sum of $SO_4^{2-}$ and $NO_3^-$ (R=0.91, slope=1.23, with P < 0.01) was better than that of $NH_4^+$ with $SO_4^{2-}$ (R=0.80, slope=1.65, with P < 0.01), suggesting that both $SO_4^{2-}$ and $NO_3^-$ were associated with $NH_4^+$. As shown in Fig.10, sulfate and nitrate were almost completely neutralized with most of the data above the 1:1 line. A few scattered data below the 1:1 line may be caused by uncertainties in measurements. Little different was found between the regression slopes of $NH_4^+$ with the sum of $SO_4^{2-}$ and $NO_3^-$ and the sum of $SO_4^{2-}$, $NO_3^-$ and $Cl^-$ due to the very low amount of $NH_4Cl$. In this study, the level of $NH_3$ was high enough to neutralize both $SO_4^{2-}$ and $NO_3^-$, and likely to be form $(NH_4)_2SO_4$ and $NH_4NO_3$. In addition to these substances, it is likely that NH3 also reacted with oxalic acid and other dicarboxylic acid to form ammonium oxalate and other organic ammonium aerosols, as discussed above."

[Figure]

Figure 10. Correlations between $[NH_4^+]$ and $[SO_4^{2-}]$ (a), $[NH_4^+]$ and $[SO_4^{2-}]+[NO_3^-]$ (b) and $[NH_4^+]$ and $[SO_4^{2-}]+[NO_3^-]+[Cl^-]$ (c) during 7-11 August 2013.

Meng, Z. Y., Xie, Y. L., Jia, S. H., Zhang, R., Lin, W. L., Xu, X. B., and Yang W.: The characteristics of atmospheric ammonia at Gucheng, a Rural Site in the North China Plain in summer 2013, J.

applied. meteor. sci., 26, 141-150, 2015. (in Chinese).

**Page 11, line 17: again, it is better to have more evidences showing that NH3 dry deposition dominates NHx deposition.**

**Answer:** This is a speculation. No evidence is available. We have deleted this paragraph.

**Page 12, line 2: where is the reference Meng et al. 2017?**

**Answer:** We have added the reference.

[revised manuscript text omitted]

During May–September 2013, the intensive field measurements of $NH_3$ and other trace gases, water-soluble ions in $PM_{2.5}$, and meteorological parameters took place at a rural site in the NCP. In this article, we report the results on $NH_3$, trace gases and major water-soluble ions in $PM_{2.5}$. We discuss temporal variations and diurnal patterns of $NH_3$ and $NH_4^+$, and examine their sources and chemical conversion mechanism. In this study, wWe also show results from thermodynamic equilibrium simulations and compared them with observationspredicted fine particle acidity during pollution episodes in summer 2013 by using hourly measured particulate water-soluble ions and precursor gases. Our results would provide insight into secondary aerosol formation associated with aerosol acidity.

[revised manuscript text omitted]
 (Lin et al., 2008; Ding et al., 2008). ~~peak was observed in the rural site in the NCPhighest monthly value was observed in June 2013, which was subject to strong influences from photochemical production of O$_3$ as already discussed above. K$^+$ in PM$_{2.5}$ is the indicator for biomass burning emission, having the highest monthly concentration in June 2013. It has been shown that regional transport from neighbouring Tianjin, Shijiazhuang, and Shanxi can have a significant impact on background site air quality in the NCP (Meng et al., 2009; Lin et al., 2009). Similarly, Ding et al. (2008) analyzed satellite and meteorological data and reported that the O$_3$ maximum in June in Beijing located in the NCP was a result of strong photochemical production, transport of regional pollution, and more intense burnings of biomass in Central Eastern China., intense burning of biomass, and transport of regional pollution.~~

**3.2.2 Diurnal variations of NH$_3$**

The average diurnal variations of NH$_3$ during June to September 2013 are shown in Fig. 3. As indicated

in Fig. 3a, NH$_3$ concentration maxima and minima were observed during 08:00−13:00 and 19:00−23:00, respectively. As for July, NH$_3$ concentrations showed a considerably more pronounced diurnal pattern with a maximum of 59.5 ppb at 08:00. The concentration of NH$_3$ gradually increased during 00:00–03:00, remained relatively constant during 04:00–06:00, and then rapidly increased from 06:00 (beginning just after sunrise). After peaking at approximately 08:00, a decrease was observed until it reached the minimum of 29.8 ppb at 19:00.

The morning peak of NH$_3$ was also observed elsewhere and could be resulted from emissions from fertilised soils and plant stomata, evaporation of dew, and human sources, as well as mixing down of ammonia from the residual layer (Trebs et al., 2004; Norman et al., 2009; Bash et al., 2010; Ellis et al., 2011). Figure  3b reveals that the relative humidity (90%-89%) and temperature (21.5-22.1°C) remained relatively constant before 06:00, but increased later in the morning. The increasing temperature can heat the earth's surface and vegetation leaves  and reduce the RH, potentially leading to  evaporation of NH$_3$  from  soil and plant-s and volatilization of ammonium aerosol (Trebs et al., 2004; Norman et al., 2009; Ellis et al., 2011), which may increase NH$_3$ concentrations in the morning. When the emission was occurring into a shallow- boundary layer, NH$_3$ increase would be more prominent . In addition, the morning rise  might also be due to the break-up of the nocturnal boundary layer. It was supposed by Ellis et al. (2011) that  the downward mixing of air containing higher NH$_3$ from the residual layer could lead to an increase of surface NH$_3$  after the breakup of the nocturnal boundary layer.

~~NH$_3$ concentrations began to decrease in the afternoon as the daytime mixed layer further developed with the high wind speed. The decreasing of NH$_3$ level from midnight to early morning might be related to the relatively small amount of night time emissions under low temperature and the enhanced participation of NH$_3$ in particulate formation. Another simultaneous flux measurements and modeling studies are necessary for more robust explanation to the observed diurnal variations of NH$_3$.~~

From Fig. 3a, it can be seen that in July  the NH$_3$  level was the highest and  peaked earlier . One reason for this might be the increased emissions of local agricultural NH$_3$ sources in July compared with those in June, August, and September. On the

average, the level NH$_3$ in July had a maximum nighttime increase (20.0 ppb from 20:00 to 06:00), which is much large than those in June (5.2 ppb), August (9.9 ppb) and September (1.8 ppb~). The early morning increase of NH$_3$ in July started from a much higher level than in other months,  resulting a earliest NH$_3$ peak in July .

There is no direct evidence of increased agricultural NH$_3$ emission in July. However, the Gucheng site is an experiment station for agrometeorological studies. Corn is the main crop in the station area and nearly all the agricultural areas in the surrounding. According the climate in the NCP, corn is planted around the middle of June and grows rapidly in July. Therefore, July is the key period for the application of nitrogen fertilisers like urea. As mentioned above, the urea application in the station on 20 July 2013 and a precipitation process afterwards caused huge NH$_3$ spikes during the end of July (Fig. 2b). In addition, the highest nighttime temperature in July (Fig. 3b) could promote the soil emission of NH$_3$, and the relatively lower wind speed (Fig. 3b) and lowest PBLH (Fig. S3) in July was in favor of the accumulation of NH$_3$ in surface air.

In summary, ambient  NH$_3$ at Gucheng showed interesting diurnal cycles, which look significantly different in different summer months. We believe the interplay of some processes, such as  emissions from agricultural sources, meteorological condition s  (temperature, relativity humidity, wind speed, and BHL height, etc.) as well as chemical conversion, are important in the determination of levels and diurnal patterns of NH$_3$ at the site. Whether or not these processes are all important in the morning variation of NH$_3$? How important are they? And what makes the difference in the peaking time and concentration of NH$_3$ in different months? These are questions to be answered in the future.~~Another possible cause for the earlier peak in August than in June involves meteorological factors such as the higher temperature, RH, and the lower wind speed in August. For instance, the air temperature increased from 07:00 reaching maximum value (30.9°C) at 15:00 in August, which was about of 3.3°C higher than that in June, meanwhile the highest wind speed (2.73 m s$^{-1}$) at 13:00 in June was about of 1.0 m s$^{-1}$ higher than that in August.~~

**3.3 Ambient ammonium aerosol**

Secondary inorganic aerosols form from gas-phase precursors, which are mostly from anthropogenic activities such as industrial, agricultural, and motor vehicle emissions. Therefore, the major precursors ($NH_3$, $SO_2$ and $NO_x$) are responsible for the formation of particulate ammonium, sulphate, and nitrate.

The hourly $NH_4^+$ concentrations during 15 June–11 August 2013 ranged from 1.07 to 340.6 µg m$^{-3}$, with an average concentration of 19.8 ± 33.2 µg m$^{-3}$. The highest monthly level of $NH_4^+$ appeared in July and lowest level appeared in June 2013. Similar to $NH_3$, the concentration of $NH_4^+$ also increased sharply after urea fertilisation, with the highest value (360.6 µg m$^{-3}$) observed at 09:00 on 28 July 2013. The highest monthly level of $NH_4^+$ appeared in July and lowest level appeared in June 2013.

The highest hourly $SO_4^2$ concentration (116.9 µg m$^{-3}$) was observed at 10:00 on 9 July and the second highest value was 111.4 µg m$^{-3}$ at 18:00 on 6 August, 2013, with an average concentration of 20.5±13.6 µg m$^{-3}$. Despite the lower concentrations of $SO_2$, higher $SO_4^{2-}$ concentrations in summer were attributed to the higher temperature, $O_3$ concentration and solar radiation, which increase the photochemical activities, the atmospheric oxidation and markedly faster conversion of $SO_2$ to $SO_4^{2-}$. The average concentration of $NO_3^-$ in $PM_{2.5}$ was 11.3 ± 9.1 µg m$^{-3}$The highest value of 109.3 µg m$^{-3}$ was observed at 14:00 on 22 June 2013 at the highest RH  (93%) and AWC (910 µg m$^{-3}$).

Figure 4 shows correlations of observed NH4+ with the sum of observed with observed $SO_4^{2-}$, $SO_4^{2-}$+$NO_3^-$ and $SO_4^{2-}$+$NO_3^-$ +Cl$^-$. Although all the correlations are significant, the slopes of the regression lines are far from unit. This cannot be due bias in measerements. The major ion balance shows ratio of 1.05:1 for cation:anion. The slope is 0.56 when all three strong acid are considered, suggesting that the neutralization of the strong acids explain 56% of the observed NH4$^+$. In other words, nearly 44% of the observed $NH_4^+$ was due to the presence of other acids in aerosol particles.

**3.4 Results from thermodynamic equilibrium simulation**

We have used the thermodynamic equilibrium model ISORROPIA II to investigate gas-aerosol partitioning characteristics. $NO_3^-$, $SO_4^{2-}$ and $NH_4^+$. The model outputs include equilibrium $NO_3^-$, $SO_4^{2-}$, $NH_4^+$. $H^+_{air}$, $HNO_3$, $NH_3$, AWC, etc. As shown in Fig. 5, the modelled $NO_3^-$, $SO_4^{2-}$, $NH_3$ show excellent

correlations with the corresponding measurements, but modelled $NH_4^+$ is much worse correlated with the measured one. Modelled $NO_3^-$, $SO_4^{2-}$, and $NH_3$ values agree well with the measurements, while the modelled $NH_4^+$ largely underestimate the measurements. Considering the unbalance between observed NH4+ and the sum of observed $SO_4^{2-}+NO_3^-+Cl^-$, we can confirm that other acids in aerosol particles are important in the conversion of $NH_3$ to $NH_4^+$. These other acids may be oxalic acid and other dicarboxylic acids. Although we did not measure organic acids in aerosol, the presence of oxalic acid and other low molecular weight dicarboxylic acids in aerosols is often reported (e.g., Hsieh et al., 2007; Kawamura et al., 2010, 2013; Sauerwein and Chan, 2017). There is no doubt about the presence of significant amount of dicarboxylic acids over the North China Plain particularly during summer (Kawamura et al., 2013). Therefore, it is highly possible that neutralizing dicarboxylic acids in aerosol particles contributed significantly to the conversion of ammonia to ammonium.

The simulated $HNO_3$ concentrations was $0.9 \pm 1.1$ µg m$^{-3}$, showing a maximum value of 7.41 µg m$^{-3}$ at 11:00 on 19 June 2013. The average diurnal variations of $HNO_3$ and $H_{air}^+$ are shown in Fig. S4. The fine particles were moderately acidic in summer, with an average pH values of 3.5. The pH values of aerosol water, estimated based on the simulated results using equation (4), are mostly in the range of 2.5-4.5, with an average of 3.5. On average, pH is over 3.5 during nighttime and below 3.5 during daytime (Fig. 6). Under the medium acidic conditions and high $NH_3$ concentrations, organic acid like diacids are able to reaction with ammonia to for ammonium. Because we used ISORROPIA-II for inorganic aerosol composition and no organic acids measurements are available, we cannot analyze in detail the role of organic acids though the model performed quite well (Fig. S5).

~~On several days during the study period, very high $NH_3$ and inorganic PM$_{2.5}$ concentrations were observed. As shown in Fig. 6a, the peak NH$_3$ value of 76.3 ppb at 09:00 on 10 August was observed, with SO$_2$ and NO$_x$ concentrations of 3.33 ppb and 14.72 ppb, respectively. A few hours later, the concentration of SO$_4^{2-}$ increased by a factor of 2.0 from 18.3 to 35.9 µg m$^{-3}$ and NH$_4^+$ concentration rose by a factor of 1.6 from 15.9 to 24.8 µg m$^{-3}$, with NO$_3^-$ concentration from 14.3 to 15.7 µg m$^{-3}$ at 13:00 on 10 August. High NH$_3$ concentration was accompanied by the large increase in concentrations of SO$_4^{2-}$, NO$_3^-$ and NH$_4^+$ during this period.~~ Several pollution episodes of precursor gases, SO$_4^{2-}$, NO$_3^-$ and NH$_4^+$ in PM$_{2.5}$ were observed during the study period. As shown in Fig. 46a, the peak values of NH$_3$, SO$_2$ and NO$_x$ (76.3 ppb at 09:00, 14.9 ppb at 14:00 on and 42.2 ppb at 00:00 on 10 August, respectively) were observed, which were accompanied by the large increase in concentrations of SO$_4^{2-}$,

$NO_3^-$ and $NH_4^+$  ~~This suggests that NH₃ played an important role in PM mass formation and that gas-particle conversion occurred when $NH_3$ was available, though $SO_4^{2-}$ partitions to the aerosol phase regard less of $NH_3$ level (Gong et al., 2013). Due to the atmospheric conditions favoring the accumulation of pollutants, the concentrations of $NH_4^+$, $SO_4^{2-}$ and $NO_3^-$ (24.8, 35.9 and 15.7 µg m⁻³, respectively) at 13:00 on 10 August were detected and higher compared with the average concentrations during 7-11 August 2013 by around 127%, 81% and 83%, respectively. These secondary ions concentrations had similar temporal distributions with slow accumulation and relatively rapid clearing under favourable meteorological conditions. There were good correlation between $NH_3$ with $NH_4^+$, $SO_4^{2-}$ and $NO_3^-$ (R=0.33, 0.27 and 0.49, respectively, with P < 0.01).Our study and other research reveal that reducing $NH_3$ concentrations could be an effective method for alleviating secondary inorganic $PM_{2.5}$ pollution in the NCP, besides the reduction of primary $SO_2$, NOx and aerosol emissions.This observation emphasizes the important role of NH₃ in the formation of secondary $SO_4^{2-}$, $NO_3^-$ and $NH_4^+$ aerosols, which should be further explored to solve the current air pollution problems in NCP and other regions of the world.~~

~~also analyzed. It is known that $(NH_4)_2SO_4$ is preferentially formed and the least volatile, $NH_4NO_3$ is relatively volatile, while NH₄Cl is the most volatile. $NH_4^+$ is thought to be first associated with $SO_4^{2-}$, afterwards, the excess of $NH_4^+$ is with nitrate and chloride (Meng et al., 2015). It is noted that the correlation of $NH_4^+$ with the sum of $SO_4^{2-}$ and $NO_3^-$ (R=0.91, slope=1.23, with P < 0.01) was better than that of $NH_4^+$ with $SO_4^{2-}$ (R=0.80, slope=1.65, with P < 0.01), suggesting that both $SO_4^{2-}$ and $NO_3^-$ were associated with $NH_4^+$.As shown in Fig.4b, sulfate and nitrate were almost completely neutralized with most of the data above the 1:1 line. A few scattered data below the 1:1 line .Little different was found between the regression slopes of $NH_4^+$ with the sum of $SO_4^{2-}$ and $NO_3^-$ and the sum of $SO_4^{2-}$, $NO_3^-$ and $Cl^-$ due to the very low amount of NH₄Cl. In this study, $NH_4^+$ was enough to neutralize both $SO_4^{2-}$ and $NO_3^-$, and likely to be in the form of $(NH_4)_2SO_4$, $NH_4HSO_4$, and $NH_4NO_3$.~~

~~The analysis of water-soluble ions in $PM_{2.5}$ at NCP rural site also indicated that biomass burning was a crucial source of aerosol species. This attribution is supported by the concentrations of $K^+$ in $PM_{2.5}$, an excellent indicator for biomass burning emission. The monthly concentrations of $K^+$ in $PM_{2.5}$ were 2.17, 1.62 and 0.98 µg m⁻³ in June, July and August, respectively, with the highest concentration in~~

June 2013. During the period of 16–19 June, 2013, elevated concentrations of K$^+$(33.28 µg m$^{-3}$ at 08:00 on 16 June), NH$_4^+$(33.28 µg m$^{-3}$), SO$_4^{2-}$ (24.53 µg m$^{-3}$), and NO$_3^-$ (29.79 µg m$^{-3}$) were observed, which is consistent with the increase of CO, SO$_2$, NO$_x$ and NH$_3$ at the same time.

The comparison of the model and measurement results of the partitioning between NH$_3$ and inorganic ions in PM$_{2.5}$ was shown in Fig. 5. The model predictions and measurements were in reasonable agreement, and the agreement on NH$_3$ was better than the NO$_3^-$, SO$_4^{2-}$ and NH$_4^+$. According to Fig. 5, the simulated NH$_3$ was in good agreement with observation with an R$^2$=0.94 and normalized mean bias (NMB) of 26.9% for all the data, implying a good performance of ISORROPIA II. Howheve, in most cases the model tended to under-predict aerosols species, especially for NH$_4^+$ in summer.

During sampling period, aerosols water content (AWC), ranged between 2.3 and 910 µg m$^{-3}$, with an average concentration of 78.7 ± 74.3 µg m$^{-3}$, which was primarily due to both high RH and denser fine particles in the atmospheric boundary layer (Bian et al., 2014).

**3. 5 Relationship between ammonia and ammonium aerosol**

The gas-to-particle conversion between NH$_3$ and NH$_4^+$ has been reported to be strongly affected by temperature, RH, radiation conditions, the concentration of primary acid gas, and other factors. In this study, NH$_4^+$ concentrations correlated significantly and positively with NH$_3$  with a correlation coefficient of 0.78 and a slope of 1.48 (Fig.7a, n=915, P < 0.01), suggesting that NH$_3$ played a important precursor role in NH$_4^+$ in PM$_{2.5}$ formation.

The ratio of NH$_3$ to NH$_x$ (NH$_3$+NH$_4^+$) has been used to identify the source of NH$_x$ and the relative contribution of NH$_3$ to NH$_x$ deposition (Lefer et al., 1999; Walker et al., 2004). A value higher than 0.5 signifies that NH$_x$ is mainly from local NH$_3$ sources and that the dry deposition of NH$_3$ dominates the NH$_x$ deposition. Robarge et al.(2002) reported that more than 70% of NH$_x$ was in the form of NH$_3$ at an agricultural site in the South-eastern United States, and concluded that given a larger deposition velocity of ammonia compared with that of ammonium, a considerable fraction of NH$_x$ could be deposited locally rather than be transported out of the region. According to hourly average concentrations, the ratio of NH$_3$/NH$_x$ varied from 0.22 to 0.97, with a mean ratio of 0.69±0.14, suggesting that NH$_3$ remained predominantly in the gas phase rather than the aerosol phase in summer 2013 at Gucheng.

The diurnal changes in gaseous precursors and aerosol species are controlled by emission and

deposition processes, horizontal and vertical transport and gas-particle partitioning. To investigate gas-particle conversion further, diurnal variation of $NH_x$ was showed in Fig. 7b. Between 08:00–18:00, a decrease in $NH_3$ would result in an increase in $NH_4^+$, which coincided with higher sulfate concentrations (discussed in Sect. 3.4). The decrease in gas phase ammonia is likely the result of uptake onto aerosols to from $(NH_4)_2SO_4$. The diurnal variability of $NH_x$ may be controlled by transport and vertical exchange. Between the hours of 08:00-18:00, $NH_3$ decreased by 43% while $NH_x$ decreased by 49%, suggested that $NH_3$ remained predominantly in the gas phase. Between 19:00 and 07:00, $NH_3$ increased by 42% and $NH_x$ increased by 51%, indicating that gas-particle partitioning contributes significantly to the decrease in gas phase ammonia during this time.

**3.5.1 Gas- to-particle conversion ratio of $NH_3$**

Sulphate and nitrate oxidation ratios (SOR and NOR) are defined in literature to investigate $SO_4^{2-}$ and $NO_3^-$ formation and gas-particle transformation (Zhang et al., 2011). The average values of SOR and NOR were estimated to be 0.64 and 0.24 during the observation period at Gucheng, with SOR and NOR being higher than previous measurements (Zhou et al., 2009; Du et al., 2011; Zhang et al., 2011). Yao et al.(2002) pointed out an SOR lower than 0.10 under conditions of primary source emissions and higher than 0.10 when sulphate was mainly produced through the secondary transformation of $SO_2$ oxidation. The value of SOR reached to 0.70 in August 2013, which may due to the enhanced atmospheric oxidant levels, sufficient ammonia for neutralization, and higher RH in summer at Gucheng (Tang et al., 2016).

To gain further insights into the transformation of $NH_3$ to $NH_4^+$, the conversion ratio of ammonium (NHR) was investigated. NHR is a measure of the extent of transformation from $NH_3$ to $NH_4^+$ in areas with significant local $NH_3$ sources, although it encompasses both transport and local equilibrium, the latter dominating further downwind from the source. In this study, the average hourly values of NHR ranged from 0.03 to 0.77, with an average of 0.30 during summer 2013. The average NHR level in this study was higher than that observed at an urban site in Beijing (Meng et al., 2017), indicating that high $NH_3$ concentrations resulting from agricultural

activities had a marked influence on the formation of ammonium. After applying fertilizer such as urea, $NH_3$ volatilization achieved the peak values, especially after the rainfall, which was in favour of the formation of $NH_4^+$. To specifically explore the behaviour and chemical transformation, we analyzed in detail the episode during which the highest hourly $NH_3$ and $NH_4^+$ were observed.

Figure 5 98 illustrates the time series of the concentrations of air pollutants related to $NH_4^+$ formation and meteorological data on 27 to 31 July, 2013. $NH_3$ concentrations increased gradually from the afternoon of 27 July, and the obvious formation of $NH_4^+$ was also observed. In addition, during the night time (20:00 on 27-07:00 on 28 July), the average RH was 92% and the wind speed was 0.4 m s$^{-1}$, respectively, which may enhance the conversion of ammonia to ammonium via aqueous phase processes. As the rapid increase of solar radiation, the highest hourly value of $NH_4^+$ was observed at 09:00 on 28 July, and the corresponding NHR peaked (0.63%) at the same time.

$NH_3$ exhibited a maximum average concentration at 04:00 on 29 July, while $NH_4^+$ concentrations increased rapidly and remained high levels in the morning, with the NHR value rising to 0.53% at 09:00 on 29 July. Elevated $NH_3$ levels coincided with elevated $NH_4^+$, indicating that $NH_3$ was the main factor promoting $NH_4^+$ formation.

Furthermore, on 30 July, the higher $NH_4^+$ concentration and NHR occurred around noon after $NH_3$ decreased, which is consistent with increasing solar radiation and $O_3$ concentration. This suggests that strong solar radiation intensity accelerated the photochemical formation of sulphuric acid or nitric acid, which subsequently reacted with $NH_3$ to form $NH_4^+$. The increases in NHR coincided with those of $NH_3$ and $O_3$ concentrations and strong solar radiation intensity.

[revised manuscript text omitted]

Figure S5 Correlation of modelled $NH_4^+$ with modelled $SO_4^{2-}$, $SO_4^{2-}+NO_3^-$ and $SO_4^{2-}+NO_3^-+Cl^-$..

Figure S6. Time series of predicted fine particle pH, predicted particle water mass, predicted $H_{air}^+$ and measured $NH_3$ and measured inorganic ions during 7-11 August 2013.

Figure 1. Sampling location in the North China Plain with emission distributions of $NH_3$ for the year 2012 from the multi-resolution emission inventory of China (http://meicmodel.org/index.html).

Figure 2. Time series of hourly data of $NH_3$, other trace gases and meteorological parameters measured during the sampling period.

Figure 3. The hourly amount of precipitation in summer 2013 (a) and monthly boundary layer heights at 14:00 during the sampling period (b).

Figure 34. Diurnal variation (a) $NH_3$ and (b) meteorological parameters during the sampling period.

Figure 45. Hourly concentrations of precursor gas and ionic species measured in the pollution episode (a) temporal variations and (b) correlations of $[NH_4^+]$ versus $[SO_4^{2-}]$, $[SO_4^{2-}]+[NO_2^-]$ and $[SO_4^{2-}]+[NO_2^-]+[Cl^-]$ during 7-11 August 2013.

Figure 6. Observed and modelled concentrations of $NH_3$, $NH_4^+$, $SO_4^{2-}$ and $NO_2^-$ in summer 2013.

Figure 7. Time series of predicted fine particle pH, particle water mass, $H_{air}^+$ and $NH_3$ during 7-11 August 2013.

Figure 8. Relationship between $NH_3$ and $NH_4^+$ (a) and diurnal variation of $NH_x$ (b) in summer 2013.

Figure 59. Time series of the concentrations of air pollutants related to $NH_4^+$ formation and meteorological data on 27 to 31 July, 2013.

[revised manuscript text omitted]

Figure S5 Correlation of modelled $NH_4^+$ with modelled $SO_4^{2-}$, $SO_4^{2-}+NO_3^-$ and $SO_4^{2-}+NO_3^-+Cl^-$..

[Figure]

Figure S6. Time series of predicted fine particle pH, predicted particle water mass, predicted $H_{air}^+$ and measured $NH_3$ and measured inorganic ions during 7-11 August 2013.

---

## Author Response (AR1)

**Response to comments by editor and anonymous reviewers**

We appreciate the editor and anonymous reviewers' comments and helpful suggestions. We have revised the manuscript according to their comments and suggestions. We hope the revised manuscript can meet the quality requirements of *Atmospheric Chemistry and Physics*.

**Response to Reviewer #1's comments**

**Anonymous Reviewer #1**

**Overview**

**This paper as the other reviewer has pointed out potentially has an amazing dataset which is really needed for greater understanding of air pollution and its impacts in agricultural regions. The paper could be hugely improved by moving away from the gas-particle ratio analysis to more detailed atmospheric chemistry and physics which would allow insight into the processes occurring and whether current understanding of emission, transformation and deposition can explain the observations. There is an overuse of "the data suggests..." and "this indicates..." without backup of information.**

**Answer:** Thank you for your comments and suggestions, which are valuable in improving the quality of our manuscript. We have made additional data analysis and revised the manuscript according to the comments and suggestions by both reviewers. To gain more insight into the role of ammonia in the formation of secondary inorganic aerosol, simulations were made using the thermodynamic equilibrium model ISORROPIA II. The measurements were used as input of model to simulate the variations of the components in gas, liquid and solid phases, which are useful in the investigation of the gas-aerosol equilibrium characteristics.

**Major comments:**

**Introduction:**

**I think the ambition of the paper (as described in the last paragraph) needs to be more detailed and then the critical analysis done in the paper.**

**P3 More details of instrumentation is needed, in particular the calibration and response time of the NH3 instrument is required. Did the authors see an influence on the response time from PM**

**deposition on the inlet and instrument filters (see Bobrutski et al 2009 and other papers for details of this issue). Some raw data and calibrations would be useful – ACP is not figure limited. Though rainfall is mentioned as a key meteorological driver, the method of measurement and the data are not shown at all.**

**Answer:** We have added some more details about the instrumentation and issues regarding quality control. We paid attention to the influences on $NH_3$ measurements from the inlet and PM deposition on it and tried to reduce such influences. Although we do not exclude some unperceivable influences from adsorption and desorption, these influences should be much smaller than the high $NH_3$ values we observed and cause mainly slightly slower response or a lag in the recorded $NH_3$ concentration, which may not impact our analysis based on hourly average data. Some figures are added in the revised manuscript and supplementary materials to provide information about calibration, meteorological condition, etc. Section 2.2 "Sampling and analysis" has been rewritten as follows:

Ambient $NH_3$ was measured using an ammonia analyser (DLT-100, Los Gatos Research, USA), which utilize a unique laser absorption technology called Off-Axis Integrated Cavity Output Spectroscopy (OA-ICOS). The analyzer has a precision of 0.2 ppb at 100 sec average and a maximum drift of 0.2 ppb over 24 hrs. The response time of the analyzer is less than 2 s (with optional external N920 vacuum pump). During the campaign, $NH_3$ data were recorded as 100-s average. In principle, the $NH_3$ analyzer does not need external calibration, because the measured fractional absorption of light at an ammonia resonant wavelength is an absolute measurement of the ammonia density in the cell (Manual of Economical Ammonia Analyzer - Benchtop Model 908-0016, Los Gatos Research). However, we confirmed the good performance of the $NH_3$ analyzer using a reference gas mixture $NH_3/N_2$ (Scottgas, USA) traceable to US National Institute for Standards and Technology (NIST). The reference gas of $NH_3$ (25.92 ppm with an accuracy of $\pm2\%$) was diluted to different concentrations using zero air and supplied to the analyzer and a sequence with 5 points of different $NH_3$ concentrations (including zero) were repeated for several times to check the performance of the analyzer. As shown in Fig. S1, the analyzer followed rapidly to changes of the $NH_3$ concentration, produced stable response under stabilized $NH_3$ concentrations, and repeated accurately (within the uncertainty) the supplied $NH_3$ concentrations. The $NH_3$ analyzer contains an internal inlet aerosol filter, which was cleaned before our campaign. Nevertheless, some very fine particles can deposit on the mirrors of the ICOS cell, leading to gradual decline in reflectivity. However, slight mirror

[revised manuscript text omitted]

Figure S1 Confirmation of the performance of NH$_3$ analyzer using diluted standard gas (mixture NH$_3$/N$_2$). Instrument response to changed NH$_3$ concentration and stability (a) and repeated multipoint calibrations (b).

[Figure]

(a)

(b)

Figure S2. Monthly average diurnal variations of the $NO_2$ photolysis frequency ($jNO_2$) (a) and hourly rainfall (b) observed at Gucheng during June-August 2013.

The references are added in the revised manuscript.

Han, B., Zhang, R., Yang, W., Bai, Z., Ma, Z., and Zhang, W.: Heavy haze episodes in Beijing during January 2013: inorganic ion chemistry and source analysis using highly time-resolved measurements from an urban site, Sci. Total Environ., 544, 319-329, 2016.

Lin, W., Xu, X., and Zhang, X.: Characteristics of gaseous pollutants at Gucheng, a rural site southwest of Beijing, J. Geophys. Res., 114, 10339, 2009.

Norman, M., Spirig, C.,Wolff, V., Trebs, I., Flechard, C., Wisthaler, A., Schnitzhofer, R., Hansel, A., and Neftel, A.: Intercomparison of ammonia measurement techniques at an intensively managed grassland site (Oensingen, Switzerland), Atmos. Chem. Phys., 9, 2635-2645, 2009.

Schwab, J.J.: Ambient Gaseous Ammonia: Evaluation of Continuous Measurement Methods Suitable for Routine Deployment, Final Report Prepared for the New York State Energy Research and Development Authority (NYSERDA),Final Report 08-15, New York, October 2008.

Wu, W. S. and Wang, T.: On the performance of a semi-continuous $PM_{2.5}$ sulphate and nitrate instrument under high loadings of particulate and sulphur dioxide. Atmos. Environ., 41, 5442-5451, 2007.

**P 11 section 3.4: Relationship between ammonium and ammonia: This discussion is very brief and limited. In particular after noting previously (and probably correctly) that local ammonia emissions dominate the chemical speciation observed, the authors then infer "NH3 dominates NHx deposition". With the dataset they have they could have performed calculations of**

**deposition vs emission over the 4 month period would have given much more insight, i.e. the process is bidirectional therefore it is uncertain whether any net deposition would occur under the ambient conditions. This is a missed opportunity to explore the atmospheric chemistry and physics of the system.**

**Answer:** Thank you for your valuable comments. Indeed it would be great if we could use our measurements to systematically explore the atmospheric chemistry and physics over the our site and to gain more insight, including the emission and deposition of $NH_3$ and $NH_4^+$. However, we did not observe the emission and deposition of $NH_3$ and $NH_4^+$. The air/surface exchange of $NH_3$ is very complex, bidirectional and highly variable, and is influenced by many factors (e.g., Schrader et al., 2016). Given the parameters we observed, it is hardly possible to obtain robust and quantitative results about the emission vs deposition purely based on available measurements. To well understand the processes of emission, conversion and deposition of $NH_x$ species in the polluted NCP region, it is highly needed to design observational and 3D modeling studies. Our in-situ measurements present in this paper can be used as supporting data in future modeling studies.

Schrader, F., Brümmer, C., Flechard, C.R., Wichink Kruit, R.J., van Zanten, M.C., Zöll, U., Hensen, A., and Erisman, J.W.: Non-stomatal exchange in ammonia dry deposition models: comparison of two state-of-the-art approaches, Atmos. Chem. Phys., 16, 13417–13430, 2016.

**Given that there are several thermodynamic models freely available, it would have been useful to explore the dataset against what is predicted by models. What is the ion balance – are dicarboxylic acids needed to explain aerosol neutralisation? (is it neutral?). Is the atmospheric chemistry at the site dominated by thermodynamic equilibrium or is there kinetic limitations on the processes? The authors have missed an opportunity with the dataset to fully understand the chemistry and rely in the results and discussions to discuss the ratios between gas and aerosol partitioning to explain scientific concepts which are known and therefore not surprising.**

**Answer:** Thank you for your suggestions. We have used the thermodynamic equilibrium model (ISORROPIA II) to investigate gas-aerosol partitioning characteristics and compared the modeling results with our measurements. The pH values of aerosol, estimated based on the simulated results, are mostly in the range of 2.5-4.5, with an average of 3.5. While simulated sulfate and nitrate are well

comparable with the measurements, simulated ammonium substantially underestimates the observed one, indicating the importance of organic acids in the formation of ammonium. Although we did not measure organic acids in aerosol, the presence of oxalic acid and other low molecular weight dicarboxylic acids in aerosols is often reported (e.g., Hsieh et al., 2007; Kawamura et al., 2010, 2013; Sauerwein and Chan, 2017). There is no doubt about the presence of significant amount of dicarboxylic acids over the North China Plain particularly during summer (Kawamura et al., 2013). Therefore, it is highly possible that neutralizing dicarboxylic acids in aerosol particles contributed significantly to the conversion of ammonia to ammonium. Our results also suggest that the gas-aerosol partitioning at the Gucheng site is dominated by thermodynamic equilibrium. We have added a section about the ISORROPIA II model in the revised manuscript and more text about the simulation results and related discussions as follows:

**2.3.2 ISORROPIA II model**

Thermodynamic gas-aerosol equilibrium characteristics during summer 2013 were examined using ISORROPIA II model (Fountoukis and Nenes, 2007). ISORROPIA II is a thermodynamic equilibrium model for inorganic gases and aerosols in the atmosphere (available at http://isorropia. eas.gatech.edu/index.php?title=Main_Page). The concentrations of the measured $NH_3$ and water-soluble ions in $PM_{2.5}$ were input into the model as total (gas + aerosol) concentrations, along with simultaneously measured relative humidity and temperature data. ISORROPIA II was run in the forward mode with metastable aerosol state salts precipitate once the aqueous phase becomes saturated with respect to salts, which often showed better performance than the stable state solution (solid + liquid) and was commonly applied in previous pH predictions (Guo et al., 2015; Bougiatioti et al., 2016; Liu et al., 2017). In this study, the aerosol properties as acidity and the water content of the aerosol are needed to investigate the aerosol acidity characteristics and role of heterogeneous chemistry in nitrate formation. The pH of aerosol water was calculated using the following equation:

$$pH = -\log_{10} \frac{1000 H^+_{air}}{AWC} \qquad (4)$$

where $H^+_{air}$ ($\mu g\ m^{-3}$) is the ion concentration of equilibrium particle hydronium and AWC ($\mu g\ m^{-3}$) is the aerosol water content from the ISORROPIA-II simulation. The evaluation of AWC prediction showed a good performance compared with observed particle water (Bian et al., 2014; Guo et al., 2015).

We have added a new section (Section 3.4 Results from thermodynamic equilibrium simulation) to present our main modeling results and discussed them also other section.

**3.4 Results from thermodynamic equilibrium simulation**

We have used the thermodynamic equilibrium model ISORROPIA II to investigate gas-aerosol partitioning characteristics. The model outputs include equilibrium $NO_3^-$, $SO_4^{2-}$, $NH_4^+$, $H^+_{air}$, $HNO_3$, $NH_3$, AWC, etc. As shown in Fig. 5, the modelled $NO_3^-$, $SO_4^{2-}$, $NH_3$ show excellent correlations with the corresponding measurements, but modelled $NH_4^+$ is much worse correlated with the measured one. Modelled $NO_3^-$, $SO_4^{2-}$, and $NH_3$ values agree well with the measurements, while the modelled $NH_4^+$ largely underestimate the measurements. Considering the unbalance between observed $NH_4^+$ and the sum of observed $SO_4^{2-}+NO_3^- +Cl^-$, we can confirm that other acids in aerosol particles are important in the conversion of $NH_3$ to $NH_4^+$. These other acids may be oxalic acid and other dicarboxylic acids. Although we did not measure organic acids in aerosol, the presence of oxalic acid and other low molecular weight dicarboxylic acids in aerosols is often reported (e.g., Hsieh et al., 2007; Kawamura et al., 2010, 2013; Sauerwein and Chan, 2017). There is no doubt about the presence of significant amount of dicarboxylic acids over the North China Plain particularly during summer (Kawamura et al., 2013). Therefore, it is highly possible that neutralizing dicarboxylic acids in aerosol particles contributed significantly to the conversion of ammonia to ammonium.

[revised manuscript text omitted]

**In the places in the manuscript which use atmospheric chemistry to explain data (e.g. 3.4.2), there are no calculations to check if what is observed is what would be expected under the conditions. Is ozone being lost to the surface or is there a haze which allows aqueous processing in the atmosphere, what might be the role of organics...), what is the surface area of PM (given composition and RH) and hence can N2O5 hydrolysis explain the observations completely? There are lots of questions which are not touched upon, though they are key to understanding the role of NH3.**

**Answer:** It is true we did not calculate the $O_3$ loss, aerosol surface, $N_2O_5$ hydrolysis, etc. Having these results would be very helpful in understanding the role of $NH_3$ in ammonium formation and other aspects of atmospheric chemistry at the site. However, our project was not designed for a "closure study" of gas-aerosol chemistry over the site. Many key parameters were not observed, such as deposition of $O_3$, $NH_3$, etc., concentrations of radical species, organic aerosols, aerosol surface and size distribution, changes in boundary layer, etc. In the absence of these parameters, many assumptions have to be made, which will lead to large uncertainties in the results. We think a comprehensive modeling study is needed to obtain quantitative assessments of all the chemical and physical processes. Such a modeling is beyond the scope of this observation-based study. Nevertheless, we have made some calculations and obtained some semi-quantitative results as answer to the reviewer's questions (Fig. R1). According to Verbeke et al. (2015), the annual mean $O_3$ dry deposition velocity in the NCP region is about 0.3 cm s$^{-1}$. Assuming a doubled deposition velocity (0.6 cm s$^{-1}$) in summer and a

constant PBLH of 200 m (this may cause an overestimate during daytime), dry deposition of $O_3$ is estimated to be in the range of 1.5-9 ppb hr$^{-1}$, which is much smaller than the estimated NO titration (37-117 ppb hr$^{-1}$). If reaction of $O_3$ with other gases and uptake by aerosol are neglected, a production rate in the range of 35-127 ppb hr$^{-1}$ is required to balance the titration and deposition losses and cause the observed net change from -10 ppb hr$^{-1}$ to 12 ppb hr$^{-1}$. Note that photochemical production during nighttime should be zero. Therefore the "production" during nighttime can be considered as source aloft the surface layer.

[Figure]

Figure R1. Estimated budget of surface $O_3$ at Gucheng during summer 2013.

We have added "Nighttime formation, aerosol uptake and hydrolysis of $N_2O_5$ are highly uncertain as has been pointed out (e.g., Xue et al. 2014). The $NO_x$ concentration during nighttime was higher than during daytime, while the $NO_3^-$ level during nighttime was only slightly higher than that during daytime (Fig. R2).

[Figure]

Figure R2. Diurnal variations of $NO_x$ and aerosol $NO_3^-$ at Gucheng during summer 2013.

By assuming high aerosol surface to mass ratio (33.7 $m^2$ $g^{-1}$, Okuda, 2013) and a high uptake coefficient (0.1, Seinfeld and Pandis, 2006), we estimate the nighttime $N_2O_5$ under the conditions over our site to be in the range of about 3-10 ppb, corresponding to a $HNO_3$ production rate of about 1-3 ppb $hr^{-1}$ (or 2.6-7.7 $\mu g$ $m^{-3}$). This rate of $HNO_3$ production would cause an obvious night production of $NH_4^+$. Indeed we can see increases in the $NH_4^+$ concentration and NHR during night (Fig. R3). However, a more or less accurate estimate of the relative contribution of the night $N_2O_5$ chemistry to $NH_3$ conversion needs to be made in the future.

[Figure]

Figure R3. Average diurnal cycles of NHR, SOR. NOR, gaseous precursors, major water soluble ions, and meteorological factors in summer 2013.

The references are added in the revised manuscript.

Okuda, T.: Measurement of the specific surface area and particle size distribution of atmospheric aerosol reference materials, Atmos. Environ., 75, 1-5, 2013.

Seinfeld, J. H. and Pandis, S. N.: Atmospheric chemistry and physics: from air pollution to climate change (2nd ed.), Wiley Interscience, New Jersey, 2006.

Verbeke, T., Lathière, J., Szopa, S., and de Noblet-Ducoudré, N.: Impact of future land-cover changes on $HNO_3$ and $O_3$ surface dry deposition, Atmos. Chem. Phys., 15, 13555-13568, 2015.

Xue, L. K., Wang, T., Gao, J., Ding, A. J., Zhou, X. H., Blake, D. R., Wang, X. F., Saunders, S. M., Fan, S. J., Zuo, H. C., Zhang, Q. Z., and Wang, W. X.: Ground-level ozone in four Chinese cities: precursors, regional transport and heterogeneous processes, Atmos. Chem. Phys., 14, 13175-13188, doi:10.5194/acp-14-13175-2014, 2014.

**Having read the paper I am still not sure what the authors want a reader to learn from the gas-particle ratios. I would suggest the authors revise to include pollution/wind rose diagrams to look at the pollution footprint (e.g. ones are available on Open air and other packages), use current thermodynamics and kinetics of the system to see if current models would accurately represent the observations, if not what may be missing?**

**Answer:** We have drawn the Figure of $NH_3$ rose and added the analysis of local source in Section 3.5 (now section 3.7) as follows:

**3.5 Local and long-rang transport source of ammonia and ammonium aerosol**

The Gucheng site is located in the densely populated rural area in the NCP, it is influenced by local sources in the surrounding areas and by long range transport of pollutants from the residential and industrial centers around it. Dependence of the concentrations of $NH_3$ on wind direction at Gucheng is studied to get insight into the distribution of local emission sources around the monitoring site.

[Figure]

Figure 11. The average $NH_3$, $NH_4^+$ concentrations and meteorological data roses in different wind sectors during summer 2013.

As shown in Fig. 11, during the sampling period, the prevailing surface winds at Gucheng were northeasterly and southwesterly. High $NH_3$ originated from the southwest sector of the measurement site, which may be due to a local unidentified agricultural or industrial source or transport from the Xushui township, which is approximately 15 km away from Gucheng. Lower $NH_3$ concentrations were observed under winds from other sectors. Since $NH_3$ is either readily converted to $NH_4^+$ or subjected to dry deposition, high concentrations are found only close to the surface and near the emission sources. Previous studies have reported an inverse relationship between ground-level concentrations of trace gases, such as ammonia, and wind speed (Robarge et al., 2002; Lin et al., 2011). Thus, $NH_3$ concentrations might be generally lower at higher wind speeds because of turbulent diffusion.

**Oxidised nitrogen chemistry and the gas-aerosol partitioning dynamic are mentioned in passing but are key to understanding whether NH3 is driving the PM formation or it is a reservoir gas which grows PM when the presence of the other pollutants is there. Biomass burning is mentioned and K and CO as the indicators. With the dataset they could estimate the fraction of PM due to biomass burning, how much of the PM is explained by biomass burning, and does the biomass burning "seed" larger PM events. Finally the discussion and conclusion would be enhanced if some discussion about the impacts and potential solutions to the impacts. What is the evidence that limiting the NH3 emission would improve the air quality - it may well achieve this, but to make the case, evidence or hypothesis is needed to back the statements up.**

**Answer:** We have made simulations using the thermodynamic equilibrium model ISORROPIA II. The simulated results indicate that the strong acids ($H_2SO_4$, $HNO_3$ and HCl) are well balanced by $NH_3$ (Fig. R4).

However, the correlation between observed $NH_4^+$ and the sum of observed $SO_4^{2-}$, $NO_3^-$ and $Cl^-$ (Fig. R5) suggests that the neutralization of the strong acids explain 56% of the observed $NH_4^+$. In other words, nearly 0.44 of the observed $NH_4^+$ was due to the presence of other acids in aerosol particles. As mentioned above these acids may be oxalic acid and other dicarboxylic acids. The level of $NH_3$ at Gucheng in summer 2013 was very high. Under the $NH_3$-rich condition acid neutralization was easily achieved.

[Figure]

Figure R4. Correlation of simulated $NH_4^+$ with simulated $SO_4^{2-}$, $NO_3^-$ and $Cl^-$.

As shown in Fig. R3, the average NHR values were around 0.30, indicating $NH_3$ was not a factor limiting the PM formation. Some recent studies suggest that the formation of sulfate in aerosol can been largely enhanced by $NO_2$ oxidation under higher pH values (e.g., Xie et al., 2015; Cheng et al., 2016). Higher pH in aerosol water can be caused by high dust aerosol or $NH_3$. The pH values in aerosol water at Gucheng during our observations was estimated to mostly range from 2.5 to 4.5, based on the ISORROPIA modeling results. Our data do not indicate any increase in sulfate content with the increased $NO_2$. Therefore, high $NH_3$ at our site was not driving more PM formation and it served as a reservoir gas to neutralize acids present in aerosol and gas phases.

[Figure]

Figure R5. Correlation of observed $NH_4^+$ with observed $SO_4^{2-}$, $SO_4^{2-}+NO_3^-$ and $SO_4^{2-}+NO_3^-+Cl^-$.

Open biomass burning occurred occasionally in the NCP region during our campaign though open burning of agricultural wastes had been prohibited. We observed no significant open fires nearby Gucheng. However, fires in the surrounding areas may impact the measurements at the site. The fire maps (Fig. R6) show that open burning occurred mainly in June and July, which is usually the period of burning wheat straw.

[Figure]

Figure R6. Open fires in the NCP region during June-September, 2013. The red triangels indicate the position of the Gucheng site. Data source: https://firms.modaps.eosdis.nasa.gov/firemap/.

During a few days around middle and end June, we observed relatively higher concentrations of CO and aerosol $K^+$ (Fig. R7), which may be resulted from biomass burning.

[Figure]

Figure R7. $K^+$ in aerosol and CO concentrations observed at Gucheng during summer 2013.

The $K^+$ concentration is fairly well correlated with the $PM_{2.5}$ concentration (Fig. R8), suggesting an

impact of biomass burning on particle pollution. The concentration of K$^+$ is not correlated with those of Na and Ca. Therefore, the observed K$^+$ in aerosol should be mainly from biomass burning. Although the slope of the K-PM$_{2.5}$ regression line is small (0.026), the total contribution of biomass burning to PM$_{2.5}$ may be much larger than a few percentages considering carbonaceous aerosols and other species emitted by biomass burning (Cheng et al., 2013).

[Figure]

Figure R8. Correlation between K$^+$ and PM$_{2.5}$ concentrations.

Cheng, Y., Engling, G., He, K.-B., Duan, F.-K., Ma, Y.-L., Du, Z.-Y., Liu, J.-M., Zheng, M., and Weber, R. J.: Biomass burning contribution to Beijing aerosol, Atmos. Chem. Phys., 13, 7765-7781, 2013.

Cheng, Y., Zheng, G., Wei, C., Mu, Q., Zheng, B., Wang, Z., Gao, M., Zhang, Q., He, K., Carmichael, G., Pöschl, U., Su, H.: Reactive nitrogen chemistry in aerosol water as a source of sulfate during haze events in China. Sci. Adv. 2, e1601530, 2016.

Xie, Y., Ding, A., Nie, W., Mao, H., Qi, X., Huang, X., Xu, Z., Kerminen, V.-M., Petäjä, T., Chi, X., Virkula, A., Boy, M., Xue, L., Guo, J., Sun, J., Yang, X., Kulmala, M., Fu, C.: Enhanced sulfate formation by nitrogen dioxide: Implications from in-situ observations at the SORPES Station, J. Geophys. Res. 120, 12679-12694, 2015.

**Minor corrections:**

**P1 Line 22 The observation that NH3 drives NH4 content of PM is not new, so I do not think the word " suggesting" is appropriate**

Answer: We have changed the word "suggesting" to "reflecting".

**1 P1 Line 24: This is a percentage not a ratio.**

**Answer:** We have revised the percentage to the ratio according to the reviewer's comments.

**P1 Line 25: use previous NCP abbreviation**

**Answer:** We have added "in the NCP" in Line 25 according to the reviewer's comments.

**P2 line 15: Actually most atmospheric chemistry text books discuss this, would cite them rather than research papers.**

**Answer:** We have revised the sentence. "Some studies have suggested that reducing $NH_3$ concentrations could be an effective method for alleviating secondary inorganic $PM_{2.5}$ pollution.

**P5, line 19: asymmetric errors would be more appropriate given that one cannot have negative concentrations.**

**Answer:** We have decided to use the range instead of standard deviation. The sentence reads now "During 15 May-25 September 2013, the average concentrations (ranges) of $NH_3$, $SO_2$ and $NO_x$ were 36.2 (0.1-862.9), 5.0 (0-86.8) and 15.4 (2.7-67.7) ppb, respectively.

**P25 Figure 2: given that it rained during the 4 months, why does the RH never go above 90%?**

**Answer:** To accurately measure high RH is still difficult so that the absolute errors associated the high RH values may be large. The measured RH does include some values over 90%, with the maximum of 93%. There might be negative bias. On the other hand, raining does not necessarily mean the surface air is saturated with water.

**P27: for the PM composition it would be useful to have them as stacked so that one can see the variation of composition through time References: There are not many references from 2015 and 2016 despite many papers being published on this subject area. I would suggest the authors review the recent literature.**

**Answer:** The figure on this page has been redrawn. The PM compositions are shown as stacked. Global radiation and wind speed data are removed from the figure. Instead measurements of $O_3$ and

$j$NO$_2$ are shown to provide information about photochemistry. We have reviewed the some recent papers (Guo et al., 2015; Sudheer and Rengarajan, 2015; Wen et al., 2015; Han et al., 2016; Tang et al., 2016; Zhao et al., 2016; Xu et al., 2017) and cited them in the revised manuscript.

[Figure]

Figure R9. Hourly concentrations of gaseous, ionic species and $j$NO$_2$ measured in the pollution episode during 7-11 August 2013.

The references are added in the revised manuscript.

Guo, H., Xu, L., Bougiatioti, A., Cerully, K. M., Capps, S. L., Hite Jr., J. R., Carlton, A. G., Lee, S.-H., Bergin, M. H., Ng, N. L., Nenes, A., and Weber, R. J.: Fine-particle water and pH in the southeastern United States, Atmos. Chem. Phys., 15, 5211-5228, https://doi.org/10.5194/acp-15-5211-2015, 2015.

Han, B., Zhang, R., Yang, W., Bai, Z., Ma, Z., and Zhang, W.: Heavy haze episodes in Beijing during January 2013: inorganic ion chemistry and source analysis using highly time-resolved measurements from an urban site, Sci. Total Environ., 544, 319-329, 2016.

Tang, X., Zhang, X.S., Ci, Z.J., Guo, J and Wang, J. Q.: Speciation of the major inorganic salts in atmospheric aerosols of Beijing, China: Measurements and comparison with model. Atmos. Environ., 133, 123-134, 2016.

Xu, W., Song, W., Zhang, Y., Liu, X., Zhang, L., Zhao, Y., Liu, D., Tang, A., Yang, D., Wang, D., Wen, Z., Pan, Y., Fowler, D., Collett Jr., J. L., Erisman, J. W., Goulding, K., Li, Y., and Zhang, F.: Air quality improvement in a megacity: implications from 2015 Beijing Parade Blue pollution control actions, Atmos. Chem. Phys., 17, 31-46, https://doi.org/10.5194/acp-17-31-2017, 2017.

Zhao, M., Wang, S., Tan, J., Hua, Y., Wu, D., and Hao, J.: Variation of Urban Atmospheric Ammonia Pollution and its Relation with $PM_{2.5}$ Chemical Property in Winter of Beijing, China. Sci. Total Environ., 16 (6) , 1378-1389, 2016.

**Response to Reviewer #2's comments**

**Anonymous Reviewer #2**

**This manuscript presents a comprehensive ambient measurement dataset, including various trace gases and particulate species, for over four months at a rural site in the North China Plain (NCP). Ammonia (NH3) is the focus of this study for its role in the formation of secondary inorganic aerosols, which accounts for a major fraction of PM2.5 in NCP. The hourly resolution, higher than many of the previous ambient ammonia measurements, enables detailed studies on individual pollution events and the diurnal variations. However, I hope the authors can take better advantage of this dataset, and go deeper into Atmospheric Chemistry and Atmospheric Physics, as indicated by the journal name. For example, this work aims to understand the impact of ammonia on secondary ammonium aerosols (page 1, line 20), facilitate developing future ammonia emission control policies (page 1, line 32), and examine the sources of ammonia and ammonium and their chemical conversion mechanism (page 3, line 8). These are all important issues, but I am not convinced that this article has advanced our current knowledge and understanding about these issues after reading it.**

**Answer:** Thank you for your comments and suggestions, which are valuable in improving the quality of our manuscript. We have made additional data analysis and revised the manuscript according to the comments and suggestions by both reviewers. To gain more insight into the role of ammonia in the formation of secondary inorganic aerosol, simulations were made using the thermodynamic equilibrium model ISORROPIA II. The measurements were used as input of model to simulate the variations of the components in gas, liquid and solid phases, which are useful in the investigation of the gas-aerosol equilibrium characteristics.

**Major comments**

**1. It takes a significant part of this manuscript to explain the observed concentrations. However many of the explanations are qualitative and even speculative. Further quantitative evidences are needed. To name a few:**

**Page 8, lines 5-6, "the monthly concentration of SO2, NOx, and CO in July and August decreased because of raid photochemical reduction, additional removal by rainfall, and excellent vertical**

**mixing." What are the evidences of photochemistry, wet scavenging, and vertical mixing?**

**Answer:** We have added the analysis of some related meteorological conditions and show figures with measurements of rainfall and the $NO_2$ photolysis frequency ($jNO_2$) during June-August 2013 as supplementary materials. Changes are made to text as follows:

The monthly concentrations of $SO_2$, $NO_x$, and CO in July and August decreased compared to those in June. In addition to less influences from biomass burning, meteorological conditions were also in favor of lowering the concentrations of these gases. Figure S2 shows the monthly average diurnal variations of $jNO_2$ and the time-series of hourly rainfall during June-August 2013. As can be seen, the average $jNO_2$ increased from June to August, indicating better conditions for photochemical reduction in July and August. There was also a slight increase in rainfall from June to August, which may promote removal of the pollutants.

[Figure]

Figure S2. Monthly average diurnal variations of the $NO_2$ photolysis frequency ($jNO_2$) (a) and hourly rainfall (b) observed at Gucheng during June-August 2013.

**Page 8, lines 8-9, the ozone was highest in June because of "photochemical production, intense burning of biomass, and transport of regional pollution". What are the evidences of more photochemistry, biomass burning, and regional transport in June? Shouldn't July have larger photochemical rates?**

**Answer:** After careful consideration we think the statement should be revised. The sentence is changed to "For the secondary pollutant $O_3$, the highest concentration was observed in June. This is consistent with previous results from Gucheng (Lin et al., 2009) and should be related with the annual maximum of background $O_3$ in the NCP, which occurs in June (Lin et al., 2008; Ding et al., 2008).

The references are added in the revised manuscript.

Ding, A. J., Wang, T., Thouret, V., Cammas, J.-P., and Nédélec, P.: Tropospheric ozone climatology over Beijing: analysis of aircraft data from the MOZAIC program, Atmos. Chem. Phys., 8, 1-13, 2008.

Lin, W., Xu, X., Zhang, X., and Tang, J.: Contributions of pollutants from North China Plain to surface ozone at the Shangdianzi GAW Station, Atmos. Chem. Phys., 8, 5889–5898, 2008.

Lin, W., Xu, X., and Zhang, X.: Characteristics of gaseous pollutants at Gucheng, a rural site southwest of Beijing, J. Geophys. Res., 114, 10339, 2009.

**Page 8, lines 26-27, the downward mixing of the residual layer containing higher ammonia concentration could lead to an increase of ammonia in the morning. This would require a very large pool of ammonia in the residual layer. Why did it not happen in other months?**

**Answer:** Indeed, the interpretation about the morning peak is mainly based on the opinions in cited references, which are mostly speculations. We think more investigations are necessary to be able to clearly and quantitatively explain the morning peak phenomenon. We have revised the 2nd and 3rd paragraphs in section 3.2.2 as follows:

The morning peak of $NH_3$ was also observed elsewhere and could be resulted from emissions from fertilized soils and plant stomata, evaporation of dew, and human sources, as well as mixing down of ammonia from the residual layer (Trebs et al., 2004; Norman et al., 2009; Bash et al., 2010; Ellis et al., 2011). Figure 3b reveals that the relative humidity (90%-89%) and temperature (21.5-22.1 °C) remained relatively constant before 06:00, but increased later in the morning. The increasing temperature can heat the earth's surface and vegetation leaves and reduce the RH, potentially leading to evaporation of $NH_3$ from soil and plants and volatilization of ammonium aerosol (Trebs et al., 2004; Norman et al., 2009; Ellis et al., 2011), which may increase $NH_3$ concentrations in the morning. When the emission was occurring into a shallow boundary layer, $NH_3$ increase would be more prominent. In addition, the morning rise might also be due to the breakup of the nocturnal boundary layer. During the sampling period, the majority of peaks of ammonia over 50 ppb occurred at night, which were attribute to local emissions, such as from agricultural activity, into a shallow nocturnal boundary layer. It was supposed by Ellis et al. (2011) that the downward mixing of air containing higher $NH_3$ from the residual layer could lead to an increase of surface $NH_3$ after the breakup of the nocturnal boundary layer.

[Figure]

Figure 3. Diurnal variation NH$_3$ (a) and meteorological parameters (b) during the sampling period.

Bash, J. O., Walker, J. T., Katul, G. G., Jones, M. R., Nemitz, E., and Robarg, W. P.: Estimation of In-Canopy Ammonia Sources and Sinks in a Fertilized Zea mays Field, Environ. Sci. Tech., 44, 1683-1689, 2010.

Ellis, R. A., Murphy, J. G., Markovic, M. Z., VandenBoer, T. C., Makar, P. A., Brook, J., and Mihele, C.: The influence of gas-particle partitioning and surface-atmosphere exchange on ammonia during BAQS-Met, Atmos. Chem. Phys., 11, 133-145, 2011.

Norman, M., Spirig, C.,Wolff, V., Trebs, I., Flechard, C., Wisthaler, A., Schnitzhofer, R., Hansel, A., and Neftel, A.: Intercomparison of ammonia measurement techniques at an intensively managed grassland site (Oensingen, Switzerland), Atmos. Chem. Phys., 9, 2635-2645, 2009.

Schwab, J.J.: Ambient Gaseous Ammonia: Evaluation of Continuous Measurement Methods Suitable for Routine Deployment, Final Report Prepared for the New York State Energy Research and Development Authority (NYSERDA),Final Report 08-15, New York, October 2008.

Trebs, I., Meixner, F. X., Slanina J., Oties, R. P., and Andreae, M. O.: Real-time measurements of ammonia, acidic trace gases and water-soluble inorganic aerosol species at a rural site in the Amazon Basin, Atmos. Chem. Phys., 4, 967-987, 2004.

**Page 9, line 5, the author explains the earlier ammonia morning peak in July by increased emissions. Further evidence?**

**Answer:** We have no direct evidence of emissions. However, the Gucheng site is an experiment station for agrometeorological studies. Corn is the main crop in the station area and nearly all the agricultural areas in the surrounding. According the climate in the NCP, corn is planted around the middle of June and grows rapidly in July. Therefore, July is the key period for the application of nitrogen fertilizers like urea. For example, during last ten days of July 2013, 225-300 kg of urea were applied per hectare of station area (Meng et al., 2015), causing huge $NH_3$ spikes during the end of July (Fig. 2). In addition, the highest nighttime temperature in July (Fig. 3b) could promote the soil emission of $NH_3$ and the relatively lower wind speed (Fig. 3b) and lower PBLH (Fig. S3) in July was in favor of the accumulation of $NH_3$ in surface air. We have revised the 4th and 5th paragraphs in section 3.2.2 as follows:

From Fig. 3a, it can be seen that in July the $NH_3$ level was the highest and peaked earliest. One reason for this might be the increased emissions of local agricultural $NH_3$ sources in July compared with those in June, August, and September. On the average, the level NH3 in July had a maximum nighttime increase (20.0 ppb from 20:00 to 06:00), which is much large than those in June (5.2 ppb), August (9.9 ppb) and September (1.8 ppb). The early morning increase of $NH_3$ in July started from a much higher level than in other months, resulting a earliest $NH_3$ peak in July.

Gucheng site is an experiment station for agrometeorological studies. Corn is the main crop in the station area and nearly all the agricultural areas in the surrounding. According the climate in the NCP, corn is planted around the middle of June and grows rapidly in July. Therefore, July is the key period for the application of nitrogen fertilizers like urea. As mentioned above, the urea application in the station on 20 July 2013 and a precipitation process afterwards caused huge $NH_3$ spikes during the end of July (Fig. 2b). In addition, the highest nighttime temperature in July (Fig. 3b) could promote the soil emission of $NH_3$, and the relatively lower wind speed (Fig. 3b) and lower PBLH (Fig. S3) in July was in favor of the accumulation of $NH_3$ in surface air.

In summary, ambient $NH_3$ at Gucheng showed interesting diurnal cycles, which look significantly different in different summer months. We believe the interplay of some processes, such as emissions from agricultural sources, meteorological conditions (temperature, relativity humidity, wind speed, and PBLH, etc.) as well as chemical conversion are important in the determination of levels and diurnal

patterns of NH$_3$ at the site. Whether or not these processes are all important in the morning variation of NH$_3$? How important are they? And what makes the difference in the peaking time and concentration of NH$_3$ in different months? These are questions to be answered in the future.

[Figure]

Figure S3. The monthly planetary boundary layer heights at 14:00 during 2013 at Gucheng.

The references are added in the revised manuscript.

Guo, J., Miao, Y., Zhang, Y., Liu, H., Li, Z., Zhang, W., He, J., Lou, M., Yan, Y., Bian, L., and Zhai, P.: The climatology of planetary boundary layer height in China derived from radiosonde and reanalysis data, Atmos. Chem. Phys., 16, 13309-13319, 2016.

Miao, Y., Guo, J., Liu, S., Liu, H., Li, Z., Zhang, W., and Zhai, P.: Classification of summertime synoptic patterns in Beijing and their associations with boundary layer structure affecting aerosol pollution, Atmos. Chem. Phys., 17, 3097-3110, 2017.

**2. A high observed concentration can always be explained by more emission, less mixing, or less removal. I think a publication in ACP should go beyond reporting the concentrations of these short-lived species, as the concentrations are highly variable. This study used the ratio between ammonium and NHx to infer the gas-particle conversion of ammonia. However the ammonium and ammonia may be from different sources, where ammonium is formed in the city with NOx and SO2, and ammonia is emitted locally. In other words, what if NHx and NH3 are decoupled?**

**Answer:** We have made simulations using the ISORROPIA II model and analyzed the model results together measurements. We think we have gained more insight than before but also acknowledge that there are limitations in our observation, modeling and data analysis. Some open issues remain to be

addressed in future studies.

The ratios NHR ($NH_4^+/NH_x$), SOR ($SO_4^{2-}/(SO_4^{2-}+SO_2)$) and NOR ($NO_3^-/(NO_3^-+NO_x)$) were calculated as measures of chemical conversion of $NH_3$, $SO_2$ and $NO_x$. You are right that $NH_x$ and $NH_3$ may be decoupled. Sources of $NH_3$, $SO_2$ and $NO_x$ may be dislocated. The lifetimes of these gases are different and hence the dispersion areas. Fine aerosol particles may travel much longer than the precursor gases. In the real situation we always observe gases and aerosols originating both from cities and from rural areas, emitted by different sources, and chemically produced. Wherever we measure, we measure is a mixture impacted by different sources from locations and processes. In this sense, we should not attribute our results only to local impacts or local situation. When our observation at a site covers a longer period, our results should be applicable to areas in varying size.

**At page 10, line 13, it is summarized that "This observation emphasizes the important role of NH3 in the formation of secondary SO4, NO3 and NH4 aerosols, which should be further explored...". The title of this manuscript is about the role of ammonia on secondary inorganic aerosols, but what exactly is this role? It is not satisfying to only know it is important and needs further exploration.**

**Answer:** We have deleted this sentence. New results from model simulation and data analysis are added and discussed in the revised manuscript. In particular, we have added two new sections:

[revised manuscript text omitted]

**Specific comments**

**Page 1, line 29: please define "transport of air mass from the North China Plain region", as the site itself is in the middle of NCP.**

**Answer:** Yes, the site is located in the middle of NCP. The concentrations of pollutant levels at Gucheng site are not only driven by local sources but also affected by long range transport. We have changed the title of Section 3.7 to "Long rang transport and local source of ammonia and ammonium" and revised the text as follows:

The Gucheng site is located in the densely populated rural area in the NCP, it is influenced by local sources in the surrounding areas and by long range transport of pollutants from the residential and industrial centers around it. Dependence of the concentrations of NH₃ on wind direction at Gucheng is studied to get insight into the distribution of local emission sources around the monitoring site. As shown in Fig. 11, during the sampling period, the prevailing surface winds at Gucheng were northeasterly and southwesterly. High NH₃ originated from the southwest sector of the measurement site, which may be due to a local unidentified agricultural or industrial source or transport from the Xushui township, which is approximately 15 km away from Gucheng. Lower NH₃ concentrations were observed under winds from other sectors. Since NH₃ is either readily converted to NH₄⁺ or subjected to

dry deposition, high concentrations are found only close to the surface and near the emission sources. Previous studies have reported an inverse relationship between ground-level concentrations of trace gases, such as ammonia, and wind speed (Robarge et al., 2002; Lin et al., 2011). Thus, $NH_3$ concentrations might be generally lower at higher wind speeds because of turbulent diffusion.

[Figure]

Figure 11. The average $NH_3$, $NH_4^+$ concentrations and meteorological data roses in different wind sectors during summer 2013.

Lin, W., Xu, X., Ge, B., and Liu, X.: Gaseous pollutants in Beijing urban area during the heating period 2007-2008: variability, sources, meteorological and chemical impacts, Atmos. Chem. Phys., 11, 8157-8170, 2011.

Robarge, W. P., Walker, J. T., McCulloch, R. B., and Murray, G.: Atmospheric concentrations of ammonia and ammonium at an agricultural site in the southeast United States, Atmos. Environ., 36, 16611-1674, 2002.

**Page 3, line 22: how large is the "surrounding area" that impacts the measurements of this site?**

**Answer:** The site is situated in the middle of a large agricultural region with many villages. The information of Gucheng site in details can be found in Lin et al. (2009). According to the maps from Lin et al. (2009) shown below, the Gucheng site is surrounded by farms, dense villages/towns, and the transportation network in the NCP. The accurate size of the surrounding area that really impacts the

measurements at Gucheng is not easy to define and varies with meteorological condition, particularly wind speed. One can do footprint analysis by setting criteria, but this is out of the scope of this paper.

[Figure]

**Figure 1.** Location of Gucheng site and the topography of the surrounding region (a) from Google Earth imagery (copyright Google Inc., used with permission) and (b) from NASA satellite map (Community Landsat7 visible color; http://worldwind.arc.nasa.gov).

**Section 2.2: what's the response time of the Los Gatos instrument? What is the concentration and accuracy of the calibration gas?**

**Answer:** We have added information revised the related text as follows:

Ambient $NH_3$ was measured using an ammonia analyzer (DLT-100, Los Gatos Research, USA), which utilize a unique laser absorption technology called Off-Axis Integrated Cavity Output Spectroscopy (OA-ICOS). The analyzer has a precision of 0.2 ppb at 100 sec average and a maximum drift of 0.2 ppb over 24 hrs. The response time of the analyzer is less than 2 s (with optional external N920 vacuum pump). During the campaign, $NH_3$ data were recorded as 100-s average.

In principle, the $NH_3$ analyzer does not need external calibration, because the measured fractional absorption of light at an ammonia resonant wavelength is an absolute measurement of the ammonia density in the cell (Manual of Economical Ammonia Analyzer - Benchtop Model 908-0016, Los Gatos Research). However, we confirmed the good performance of the $NH_3$ analyzer using a reference gas

mixture $NH_3/N_2$ (Scottgas, USA) traceable to US National Institute for Standards and Technology (NIST). The reference gas of $NH_3$ (25.92 ppm with an accuracy of ±2%) was diluted to different concentrations using zero air and supplied to the analyzer and a sequence with 5 points of different $NH_3$ concentrations (including zero) were repeated for several times to check the performance of the analyzer.

[Figure]

Figure S1 Confirmation of the performance of $NH_3$ analyzer using diluted standard gas (mixture $NH_3/N_2$). Instrument response to changed $NH_3$ concentration and stability (a) and repeated multipoint calibrations (b).

As shown in Fig. S1, the analyzer followed rapidly to changes of the $NH_3$ concentration, produced stable response under stabilized $NH_3$ concentrations, and repeated accurately (within the uncertainty) the supplied $NH_3$ concentrations. The $NH_3$ analyzer contains an internal inlet aerosol filter, which was cleaned before our campaign. Nevertheless, some very fine particles can deposit on the mirrors of the ICOS cell, leading to gradual decline in reflectivity. However, slight mirror contamination does not cause errors in $NH_3$ measurements because the mirror reflectivity is continually monitored and the measurement is compensated using the mirror ringdown time. Interferences to $NH_3$ measurements can be from the sample inlets, for example, due to water condensation or adsorption/desorption effects (e.g., Schwab, 2008; Norman et al., 2009). Such interferences were not quantified but reduced as possibly as we could. PTFE tubing (4.8 mm ID), which is one of the well suited materials for $NH_3$ measurement (Norman et al., 2009), was used to induced ambient air. The length of the tubing was kept as short as possible (about 5 m) to limit the residue time to less than 3 s. The aerosol filter at the inlet was changed every two weeks. Water condensation was avoided. Nevertheless, we cannot exclude the influence

from the adsorption and desorption, which can also occur on dry surfaces. However, this influence should be small at our site, where the $NH_3$ concentration is very high, and cause mainly a lag in the recorded $NH_3$ concentration.

The references are added in the revised manuscript.

Norman, M., Spirig, C.,Wolff, V., Trebs, I., Flechard, C., Wisthaler, A., Schnitzhofer, R., Hansel, A., and Neftel, A.: Intercomparison of ammonia measurement techniques at an intensively managed grassland site (Oensingen, Switzerland), Atmos. Chem. Phys., 9, 2635-2645, 2009.

Schwab, J.J.: Ambient Gaseous Ammonia: Evaluation of Continuous Measurement Methods Suitable for Routine Deployment, Final Report Prepared for the New York State Energy Research and Development Authority (NYSERDA),Final Report 08-15, New York, October 2008.

**Page 6, line 3: please define "human activity" as it seems a very broad concept.**

**Answer:** We have changed "human activity" to "human excrement and waste disposal".

**Page 6, line 5: there is no "Zhng et al., 2010" in the reference list.**

**Answer:** This was one of the typos. We have corrected it to **"Zhang et al., 2010".**

**Page 6, line 8-9: please clarify how can these results "be used in improving NH3 emission inventory and making future emission control policies".**

**Answer:** We have revised our expression as follows:

In recent year a few publications about China's national and regional emission inventories of $NH_3$ (e.g., Zhou et al., 2015; Xu et al., 2015, 2016; Kang et al., 2016). However, these inventories are based on bottom-up studies, subject to substantial uncertainties in spatial and temporal variations of $NH_3$ emissions. Ground based observations of $NH_3$ have been sparse. Our measurements, together with others, can be used for validating and constraining models that use bottom-up inventories, and hence help to reveal potential bias in $NH_3$ emission inventory.

Kang, K., Liu, M., Song, Y., Huang, X., Yao, H., Cai, X., Zhang, H., Kang, L., Liu, X., Yan, X., He, H., Zhang, Q.,  Shao, M., and Zhu, T.: High-resolution ammonia emissions inventories in China from 1980 to 2012, Atmos. Chem. Phys., 16, 2043-2058, 2016.

Xu, P., Zhang, Y., Gong, W., Hou, X., Kroeze, C., Gao, W.,and Luan, S: An inventory of the emission of ammonia from agricultural fertilizer application in China for 2010 and its high-resolution spatial distribution, Atmos. Environ., 115, 141-148, 2015.

Xu, P., Liao, Y. J., Lin, Y. H., Zhao, C. X., Yan, C. H., Cao, M. N., Wang, G. S., and Luan, S. J.: High-resolution inventory of ammonia emissions from agricultural fertilizer in China from 1978 to 2008, Atmos. Chem. Phys., 16, 1207–1218, 2016.

Zhou, Y., Cheng, S., Lang, J., Chen, D., Zhao, B., Liu, C., Xu, R., and Li, T.: A comprehensive ammonia emission inventory with high-resolution and its evaluation in the Beijing-Tianjin-Hebei (BTH) region, China, Atmos. Environ., 106, 305-317, 2015.

**Figure 2: I understand ammonia is shown in log scale because its concentration spanned three orders of magnitude. However I suggest add a sub plot with linear scale so that the variability is comparable at different concentration levels and the individual spikes from pollution episodes are clearer.**

**Answer:** Thank you for your suggestion. We have redrawn Figure 2 as shown below:

[Figure]

Figure 2. Time series of hourly data of NH$_3$, other trace gases and meteorological parameters measured during the sampling period (a) and a blow-up of the period with extremely high NH$_3$ values during 27-31 July 2013 (b).

**Page 7, line 6: where the urea was applied and how large was the applied area?**

**Answer:** We have added the required information and revised the text as follows:

The Gucheng station has a farmland of 8.67 hectares. The observation period was in the time of the wheat harvest and corn seeding and growing. Corn was sown and fertilized with about 600 kg of fertilizer per hectare in late June. On 20 July corn was additionally fertilized with 225 to 300 kg of urea per hectare. After this fertilization, there was a raining period. The $NH_3$ concentration increased rapidly on the seventh day after the urea application on 20 July, peaking during the 27-30 July period (Fig. 2b).

**Page 7, line 29: what are these "trace gases"?**

**Answer:** We have added "such as $NO_x$ and CO"at the end of "trace gases" to in our revised version.

**Page 8, lines 14-15 and page 8, lines 30: these two sentences seem contradict each other.**

**Answer:** We have deleted the paragraph there.

**Page 10, line 1: higher NO3 level than what?**

**Answer:** We have deleted this sentence and changed the sentence before to " On the other hand, $NH_3$ was more efficient in summer to react with $SO_2$ to form $(NH_4)_2SO_4$. The average concentration of $NO_3^-$ in $PM_{2.5}$ was 11.3 ±9.1 µg m$^{-3}$. The highest value of 109.3 µg m$^{-3}$ was observed at 14:00 on 22 June 2013 at the highest RH (93%) and AWC (910 µg m$^{-3}$). The high RH conditions in summer might dissolve a significant fraction of $HNO_3$ and $NH_3$ in humid particles, therefore increasing the concentrations of $NO_3^-$ and $NH_4^+$ in the atmosphere (Krupa et al., 2003; Trebs et al., 2004; Ianniello et al., 2010). The high $NO_3^-$ concentrations were also mostly associated with large aerosol water content, which indicates the importance of heterogeneous hydrolysis in the production of nitrate (Pathak et al., 2009).

The reference is added in the revised manuscript.

Pathak, R. K., Wu, W. S., and Wang, T.: Summertime $PM_{2.5}$ ionic species in four major cities of China: nitrate formation in an ammonia-deficient atmosphere, Atmos. Chem. Phys., 9, 1711-1722, https://doi.org/10.5194/acp-9-1711-2009, 2009.

**Page 10, line 20-25: I suggest add a figure showing the slope and correlations. The SO4 should be normalized with its number of charge. What is the evidence for the existence of NH4HSO4?**

**Answer:** We have added a figure to show the correlations as you suggested and revised the text accordingly.

During 7-11 August 2013, the relationships of the observed $NH_4^+$ versus those of $SO_4^{2-}$, the sum of $SO_4^{2-}$ and $NO_3^-$ and the sum of $SO_4^{2-}$, $NO_3^-$ and $Cl^-$ are presented in Fig. 10. It is known that $(NH_4)_2SO_4$ is preferentially formed and the least volatile, $NH_4NO_3$ is relatively volatile, while $NH_4Cl$ is the most volatile. $NH_4^+$ is thought to be first associated with $SO_4^{2-}$, afterwards, the excess of $NH_4^+$ is with nitrate and chloride (Meng et al., 2015). It is noted that the correlation of $NH_4^+$ with the sum of $SO_4^{2-}$ and $NO_3^-$ (R=0.91, slope=1.23, with P < 0.01) was better than that of $NH_4^+$ with $SO_4^{2-}$ (R=0.80, slope=1.65, with P < 0.01), suggesting that both $SO_4^{2-}$ and $NO_3^-$ were associated with $NH_4^+$. As shown in Fig.10, sulfate and nitrate were almost completely neutralized with most of the data above the 1:1 line. A few scattered data below the 1:1 line may be caused by uncertainties in measurements. Little different was found between the regression slopes of $NH_4^+$ with the sum of $SO_4^{2-}$ and $NO_3^-$ and the sum of $SO_4^{2-}$, $NO_3^-$ and $Cl^-$ due to the very low amount of $NH_4Cl$. In this study, the level of $NH_3$ was high enough to neutralize both $SO_4^{2-}$ and $NO_3^-$, and likely to be form $(NH_4)_2SO_4$ and $NH_4NO_3$. In addition to these substances, it is likely that $NH_3$ also reacted with oxalic acid and other dicarboxylic acid to form ammonium oxalate and other organic ammonium aerosols, as discussed above.

[Figure]

Figure 10. Correlations between $[NH_4^+]$ and $[SO_4^{2-}]$ (left), $[NH_4^+]$ and $[SO_4^{2-}]+[NO_3^-]$ (middle) and $[NH_4^+]$ and $[SO_4^{2-}]+[NO_3^-]+[Cl^-]$ (right) during 7-11 August 2013.

Meng, Z. Y., Xie, Y. L., Jia, S. H., Zhang, R., Lin, W. L., Xu, X. B., and Yang W.: The characteristics of atmospheric ammonia at Gucheng, a Rural Site in the North China Plain in summer 2013, J. applied. meteor. sci., 26, 141-150, 2015. (in Chinese).

**Page 11, line 17: again, it is better to have more evidences showing that NH3 dry deposition dominates NHx deposition.**

**Answer:** This is a speculation. No evidence is available. We have deleted this paragraph.

**Page 12, line 2: where is the reference Meng et al. 2017?**

**Answer:** We have added the reference.

[revised manuscript text omitted]

[revised manuscript text omitted]

oxidation. The value of SOR reached to 0.70 in August 2013, which may due to the enhanced atmospheric oxidant levels, sufficient ammonia for neutralization, and higher RH in summer at Gucheng (Tang et al., 2016).

To gain further insights into the transformation of $NH_3$ to $NH_4^+$, the conversion ratio of ammonium (NHR) was investigated. NHR is a measure of the extent of transformation from $NH_3$ to $NH_4^+$ in areas with significant local $NH_3$ sources, although it encompasses both transport and local equilibrium, the latter dominating further downwind from the source. In this study, the average hourly values of NHR ranged from 0.03 to 0.77, with an average of 0.30 during summer 2013. The average NHR level in this study was higher than that observed at an urban site in Beijing (Meng et al., 2017), indicating that high $NH_3$ concentrations resulting from agricultural activities had a marked influence on the formation of ammonium.

**3.5.2 Diurnal patterns of NHR, SOR and NOR**

Fig. 8 presents the diurnal patterns of NHR, SOR, NOR, gaseous precursors, and major water soluble ions, and meteorological factors. As a key species contributing to the oxidisation capacity of the atmosphere, $O_3$ can promote $HNO_3$ formation, affecting the conversion ratio of $NH_3$. $O_3$ exhibited low levels in the morning and enhanced levels in the late afternoon. The lower morning concentrations may be due to the depositional loss of $O_3$ under stable atmospheric conditions in early morning hours, and the higher levels in the afternoon could be due to the photochemical production of $O_3$. The $NH_4^+$ concentration started to increase from morning, reaching the maximum value (16.1 μg m$^{-3}$) at 18:00 , with a diurnal difference of 3.7 μg m$^{-3}$. This diurnal pattern may be due to a combination of high $NH_3$ concentrations, the intense solar radiation at noon, and the high oxidisation capacity of the atmosphere in the afternoon. A clear diurnal cycle of NHR existed, with an amplitude of 0.10 and a peak of 0.35 at 18:00, which is consistent with the higher SOR and RH.

The $SO_2$ concentration showed a maximum at 09:00, with a secondary peak at 22:00. The concentration of $SO_4^{2-}$ showed small peak at 11:00, 14:00 and 18:00, respectively, but no strong diurnal variation. SOR displayed a diurnal cycle with the highest value of 0.74 observed at 05:00. It is noted that the SOR was lower during daytime when photochemical reaction is intense. Higher SOR during nighttime suggests importance of dark reactions. $SO_2$ is highly soluble and can easily absorbed by wet aerosol particles. The much RH during night may promoted this process.

As for the diurnal cycle of $NO_x$, a peak was observed at 06:00 when the mixing layer was stable,

[revised manuscript text omitted]

but probably with oxalic acid and other diacids, which may present under the medium aerosol acidity (pH around 3.5).

The back trajectory analysis indicates that the transport from the North China Plain region contributed for 56% of air mass with high $NH_3$ levels, meanwhile the long-distance transport from southeast accounted for 32% of air mass with high $NH_4^+$, $SO_4^{2-}$ and $NO_3^-$ at the rural site in the NCP.

$NH_3$ is currently not included in China's emission control policies of air pollution precursors though people have been discussing the necessity for years. Our findings highlight the important role of $NH_3$ in the participation of secondary inorganic and organic aerosol formation. As the emission and concentration of $NH_3$ in the NCP are much higher than needed for aerosol acids neutralization, we speculate that a substantial amount of reduction in $NH_3$ emission is required to see its effect on the alleviation of $PM_{2.5}$ pollution in the NCP. Therefore, further strong reduction of the emissions of primary aerosol, $NH_3$, $SO_2$, $NO_x$, and VOCs is suggested to address the serious occurrence of $PM_{2.5}$ pollution on the North China Plain.

[revised manuscript text omitted]

---

## Author Response (AR2)

**Response to comments by editor**

We appreciate the editor and referee's comments and helpful suggestions. We have revised the manuscript according to the comments and suggestions. We hope the revised manuscript can meet the quality requirements of *Atmospheric Chemistry and Physics*.

**Response to editor's comments**

**Comments to the Author:**

**Dear Authors: I agree with the Referee that the manuscript has substantially improved and is closer to being ready for publication. In addition to addressing the suggestion by the Referee, please consider my comments below. Thanks.**

**Answer:** Thank you very much. We have revised the manuscript according to your comments and suggestions.

**Page 4 line 10: Please list the range of non-zero concentrations used for the calibration in the text).**

**Answer:** We have listed the range of non-zero concentrations used for the calibration in revised the manuscript, as follows:

The reference gas of $NH_3$ (25.92 ppm with an accuracy of ±2%) was diluted to different concentrations using zero air and supplied to the analyzer and a sequence with 5 points of different $NH_3$ concentrations (including zero and the concentrations ranging from 45-180 ppb) were repeated for several times to check the performance of the analyzer.

**Page 4 line 13: Please provide a response time (not just "rapidly") for the instrument.**

**Answer:** The response time of $NH_3$ instrument has been shown in Page 4 line 3:

The response time of the analyzer is less than 2 s (with optional external N920 vacuum pump). During the calibration, it took about 20 min for the instrument to show 90% of the changes in supplied $NH_3$ concentrations. However, this time delay contained also the balance time needed for the calibration system. The lag caused by the tubing and analyzer should be much smaller.

**Page 4: Was the calibration through the 5 m inlet line and the aerosol filter at the inlet? In other**

**words, was the calibration gas introduced at the beginning of the inlet or closer to the instrument? This would obviously affect response time and have implications for the measurements. Please comment.**

**Answer:** The calibration gas was introduced into the entrance of instrument through an aerosol filter and a PTFE tubing (about 3 m). This setup was not identical but close to that for the observation. As stated above, it took about 20 min for the instrument to show 90% of the changes in supplied $NH_3$ concentrations, which contained also the balance time needed for the calibration system. Even if the lag time during the observation was a little longer than the residue time (3 s), we do not expected a substantial influence from it on our hourly average data that are used in this paper.

In response to this and the above comments, we have revised the text by inserting "During the calibration, it took about 20 min for the instrument to show 90% of the changes in the $NH_3$ concentrations supplied through an aerosol filter and a PTFE tubing (4.8 mm ID, about 3 m). However, these time delays contained also the balance time needed for the calibration system. The lag caused by the tubing and analyzer should be much smaller." in line 13 on page 4 and adding "As hourly averages are used in the work, this lag may not exert a significant influence on our results." at the end of line 26 on page 4.

**Page 4 line 24: Was there any observation of different NH3 losses on a clean versus dirty inlet filter?**

**Answer:** There was no observation of different $NH_3$ losses on a clean versus dirty inlet filter in this study, which should be done carefully in future study. As mentioned on page 4, we changed the filter every two weeks to reduce potential impacts of dirty filters. And the impacts if any might be small as the concentration of $NH_3$ at our site was often very high and the acids in aerosol were well neutralized.

**Page 5 line 16-17: Please include information on chloride detection limits.**

**Answer:** Thank you for your suggestion. We have added the required information as follows:

The detection limit of $NH_4^+$, $SO_4^{2-}$, $NO_3^-$ and $Cl^-$ were 0.05 μg m$^{-3}$, 0.04 μg m$^{-3}$, 0.05 μg m$^{-3}$ and 0.01 μg m$^{-3}$, respectively.

**Page 12 line 25: Are the correlations statistically significant? If not, please use a different word.**

**Answer:** Thank you for your suggestion. We have changed the word "significant" to "relatively high". All correlations shown in Fig. 5 are statistically significant ($\alpha=0.01$) but the correlation between modelled and observed $NH_4^+$ is not as good as the others. We say in this sentence that the modelled $SO_4^{2-}$, $NO_3^-$ and $NH_3$ show excellent correlations with the corresponding measurements ($R^2>0.81$). We have changed our expression to "As shown in Fig. 5, the modelled $SO_4^{2-}$, $NO_3^-$ and $NH_3$ are highly correlated with the corresponding measurements ".

**Page 13 line 3: Please list all the model outputs. Don't use "etc."**

**Answer:** Thank you for your suggestion. We have listed all the model outputs in the text, and deleted the word "etc".

**Page 13 line 3-6: Please also comment on the slope of the regression, not just the correlation. There seems to be a high bias for NH3 and a low bias for sulfate and nitrate (although this may be due to outliers – see comment regarding Fig. 5). Please discuss this and the implications for the results of this manuscript. Please consider including normalized mean error and bias of the measurement-model comparison.**

**Answer:** Thank you for your suggestion. We have added the calculation equations of normalized mean error and bias of the measurement-model comparison in Section 2.3.2, as follows:

The statistical analysis of the comparison between observations and model predictions was performed by calculating normalized mean bias (NMB) and normalized mean error (NME):

$$NMB = \frac{\sum_{i=1}^{N} M_i - O_i}{\sum_{i=1}^{N} O_i} \times 100\% \quad (5)$$

$$NME = \frac{\sum_{i=1}^{N} |M_i - O_i|}{\sum_{i=1}^{N} O_i} \times 100\% \quad (6)$$

where, $M_i$ and $O_i$ are modelled and observed data, respectively. N is the number of data points.

You are right. The low biases for sulfate and nitrate were due to the outliers. After removing the

outliers the slope becomes 0.994 for the correlation between modeled and observed sulfate and 1.01 for that between modeled and observed nitrate, indicating no significant biases for sulfate and nitrate. We have revised the first paragraph in Section 3.4 as follows:

We have used the thermodynamic equilibrium model ISORROPIA II to investigate gas-aerosol partitioning characteristics. The model outputs include equilibrium $SO_4^{2-}$, $NO_3^-$, $NH_4^+$, $H_{air}^+$, $HNO_3$, $NH_3$, AWC. As shown in Fig. 5, the modelled $SO_4^{2-}$, $NO_3^-$ and $NH_3$ are highly correlated with the corresponding measurements, with the slops of regression lines being 0.99, 1.01 and 1.13 for $SO_4^{2-}$, $NO_3^-$ and $NH_3$, respectively. The correlation between modelled and observed $NH_4^+$ is comparably poor but still significant ($R^2$=0.46, P<0.01), with a slope of 0.45. The modelled $SO_4^{2-}$ and $NO_3^-$ agree nearly perfectly with the measurements, while the modelled $NH_3$ and $NH_4^+$ show a slight overestimate and a large underestimate of the respective measurements. Considering the unbalance between observed $NH_4^+$ and the sum of observed $SO_4^{2-}$+$NO_3^-$+$Cl^-$ (see Fig. 4), we believe that other acids in aerosol particles are important in the conversion of $NH_3$ to $NH_4^+$. These other acids may be oxalic acid and other dicarboxylic acids. Although we did not measure organic acids in aerosol, the presence of oxalic acid and other low molecular weight dicarboxylic acids in aerosols is often reported (e.g., Hsieh et al., 2007; Kawamura et al., 2010, 2013; Sauerwein and Chan, 2017). There is no doubt about the presence of significant amount of dicarboxylic acids over the North China Plain particularly during summer (Kawamura et al., 2013). Therefore, it is highly possible that neutralizing dicarboxylic acids in aerosol particles contributed significantly to the conversion of ammonia to ammonium. The presence of significant amount of dicarboxylic acids can well explain the substantial underestimate of $NH_4^+$ and slight overestimate of $NH_3$ by the model as the model simulations did not include organics, which were not observed. The low slope (0.45) of the regression line for modelled and observed $NH_4^+$ implies that organic diacids contributed at least a half to the conversion of ammonia to ammonium. Since more $NH_4^+$ existed in aerosol than required for neutralizing inorganic acids, the ISORROPIA model simulated higher equilibrium $NH_3$ than observed, leading to an overestimate of $NH_3$.

Comparisons between the measurements and the modelled values give a positive NMB for $NH_3$ (26.9%) and negative NMB for $SO_4^{2-}$ (-0.5%), $NO_3^-$ (-12.5%) and $NH_4^+$ (-32.6%). NME gives an estimation of overall discrepancy between measurement and model (Sudheer and Rengarajan, 2015). The NME values for the model-observation comparison of $NH_3$, $SO_4^{2-}$, $NO_3^-$ and $NH_4^+$ are 31.6%, 0.9%, 14.8% and 45.8%, respectively. These results indicate that the model simulated $SO_4^{2-}$ and $NO_3^-$ well but

over-predicted $NH_3$ and under-predicted $NH_4^+$ by around 30% mainly because of the neglect of organics. Therefore, organics should be observed and included in the model simulations in future studies to better investigate aerosol components and gas-to-aerosol conversion.

[Figure]

Figure 5. Observed and modelled concentrations of $NH_3$, $NH_4^+$, $SO_4^{2-}$ and $NO_3^-$ in summer 2013.

The reference is added in the revised manuscript.

Sudheer, A. K. and Rengarajan, R.: Time-resolved inorganic chemical composition of fine aerosol and associated precursor gases over an urban environment in western India: gas-aerosol equilibrium characteristics, Atmos. Environ., 109, 217-227, 2015.

**Page 17 line 1: Please consider changing "confirming" to something along the lines of "is consistent with." Given the qualitative nature of the discussion here, it seems strong to say "confirms."**

**Answer:** Thank you for your suggestion. We have changed the word "confirming" to "which is consistent with the view".

**Page 17 lines 10-23 & Figure 10: There almost appears to be two distinct regions of data that exhibit different trends– one at low sulfate and one at high sulfate. Can you comment on these two different regimes?**

**Answer:** Thank you for pointing out this. Yes, it looks like there are two groups of data, with the one (Group 1) covering low to middle levels but exhibiting a larger slope (or trend) and the other (Group 2) covering middle to high levels but exhibiting a smaller slope. Data in Group 2 were mainly from 10

August, 2013 when high sulfate, nitrate and ammonium were observed (see Fig. 9). The difference in slope might be caused by different fractions of organic diacids that are considered to convert more ammonia to ammonium. Although we have no measurements of diacids, fractional changes of different aerosol components during pollution episodes could be large as shown in Wang et al. (2016). We have added interpretations as follows:

Data in Fig. 10 seem to be distributed in two groups, with the one (Group 1) covering low to middle levels but exhibiting a larger slope and the other (Group 2) covering middle to high levels but exhibiting a smaller slope. Data in Group 2 were mainly from 10 August, 2013 when high $NH_4^+$, $SO_4^{2-}$ and $NO_3^-$ were observed (see Fig. 9). The difference in slope between the two groups might be caused by different fractions of organic diacids that are considered to convert more ammonia to ammonium. Although no measurements of organic diacids are available to prove this, the significant changes in fractions of organic and inorganic aerosol components are often observed during pollution episodes in China (e.g., Wang et al. 2016). Reduced impact from organic diacids may bring the correlation of $NH_4^+$ with the sum of $SO_4^{2-}$, $NO_3^-$ and $Cl^-$ closer to the 1:1 line.

The reference is added in the revised manuscript.

Wang, G., Zhang, R., Gomez, M. E., Yang, L., Zamora, M. L., Hu, M., Lin, Y., Peng, J., Guo, S., Meng, J., Li, J., Cheng, C., Hua, T., Ren, Y., Wang, Y., Gao, J., Cao, J., An, Z., Zhou, W., Li, G., Wang, J., Tian, P., Marrero-Ortiz, W., Secrest, J., Du, Z., Zheng, J., Shang, D., Zeng, L., Shao, M., Wang, W., Huang, Y., Wang, Y., Zhu, Y., Li, Y., Hu, J., Pan, B., Cai, L., Cheng, Y., Ji, Y., Zhang, F., Rosenfeld, D., Liss, P. S., Duce, R. B., Kolb, C. E, and Molina, M .J.: Persistent sulfate formation from London Fog to Chinese haze, PNAS, 113,13630-13635, 2016.

**Sect 3.7: Would "transport effects on local ammonia and ammonium" be a better title for this section?**

**Answer:** Thank you for your suggestion. We have changed the title to "Transport effects on local ammonia and ammonium " according to your suggestion.

**Figures: Please make sure subpanels are labeled in all figures. The resolution on several figures needs to be improved in order to make the axis labels legible.**

**Figure 4: Please keep the y-axis range the same for all subpanels to make it easier to compare between the subpanels.**

**Answer:** Thank you for your suggestion. We have redrawn Fig.4 as shown below:

[Figure]

Figure 4. Correlation of observed $NH_4^+$ with observed $SO_4^{2-}$, $SO_4^{2-}+NO_3^-$ and $SO_4^{2-}+NO_3^-+Cl^-$.

**Figure 5: In the modeled vs observed sulfate and nitrate there appear to be a few outliers (low modeled, high observed) that are perhaps pulling the fit down. What is the source of these outliers? Are the outlier points for sulfate and nitrate at the same time?**

**Answer:** Thank you for your suggestion. Just like what the editor means, some outliers (low modeled, high observed) of sulfate and nitrate pull the fit down. We have redrawn Fig.5 as shown above after removing the outliers (three for sulfate and four for nitrate). The slopes for sulfate and nitrate become 0.99 and 1.01, respectively, very close unit. The outlier points for sulfate and nitrate were found at different times. The outliers points of sulfate occurred at 10:00 and 11:00 on 9 July and 09:00 on 27 July, while those of nitrate occurred at 14:00 on 22 June, 23:00 on 28 June, and 04:00 and 20:00 on 1 July. The outliers may be attributed to different causes. After close inspection of our measurements, we believe that the outliers for sulfate were mainly due to large positive biases in sulfate analysis and those for nitrate were mainly resulted from obvious short-time influences of biomass burning (enhanced $K^+$) and dust (enhanced $Ca^{2+}$ and $Mg^{2+}$). These problems occurred only during a few short periods hence do not influence the overall analysis.

**Figure 10: Please include units.**

**Answer:** Thank you for your suggestion. We have redrawn Fig.10 as shown below:

[Figure]

Figure 10. Correlations between $NH_4^+$ and $SO_4^{2-}$, $SO_4^{2-}+NO_3^-$, $SO_4^{2-}+NO_3^-+Cl^-$ during 7-11 August 2013.

**Figure S1: Please improve the resolution of the figure. How long is the signal smoothed for in panel a? Why was the signal smoothed?**

**Answer:** Thank you for your suggestion. The plot with smoothed data (Figure S1a) was a screenshot for the $NH_3$ analyzer. Smoothing is a default setting for displaying measured $NH_3$, which is not necessary for our purpose. We have redrawn Fig.S1 as shown below:

[Figure]

Figure S1. Confirmation of the performance of $NH_3$ analyzer using diluted standard gas (mixture $NH_3/N_2$). Instrument response to changed $NH_3$ concentration and stability (a) and repeated multipoint calibrations (b).

[revised manuscript text omitted]
  $SO_4^{2-}$, $NO_3^-$, $NH_3$, $NH_4^+$, H$^+$$_{air}$, $HNO_3$,  AWC. As shown in Fig. 5, the modelled $SO_4^{2-}$, $NO_3^-$,  $NH_3$ are highly correlated with the corresponding measurements, with the slops of regression lines being 0.99, 1.01 and 1.13 for $SO_4^{2-}$  for $NO_3^-$  and  $NH_3$ , respectively.  The correlation between modelled and observed $NH_4^+$ is comparably poor but still significant (R$^2$=0.46, P<0.01), with a slope of 0.45. The Modelled  $SO_4^{2-}$, and $NO_3^-$ agree nearly perfectly with the measurements, while the modelled $NH_3$ and $NH_4^+$ show a slight overestimate and a large underestimate of the respective measurements. Considering the unbalance between observed $NH_4^+$ and the sum of observed $SO_4^{2-}+NO_3^-+Cl^-$ (see Fig. 4), we believe that other acids in aerosol particles are important in the conversion of $NH_3$ to $NH_4^+$. These other acids may be oxalic acid and other dicarboxylic acids. Although we did not measure organic acids in aerosol, the presence of oxalic acid and other low molecular weight dicarboxylic acids in aerosols is often reported (e.g., Hsieh et al.,

2007; Kawamura et al., 2010, 2013; Sauerwein and Chan, 2017). There is no doubt about the presence of significant amount of dicarboxylic acids over the North China Plain particularly during summer (Kawamura et al., 2013). Therefore, it is highly possible that neutralizing dicarboxylic acids in aerosol particles contributed significantly to the conversion of ammonia to ammonium. The presence of significant amount of dicarboxylic acids can well explain the substantial underestimate of $NH_4^+$ and slight overestimate of $NH_3$ by the model as the model simulations did not include organics, which were not observed. The low slope (0.45) of the regression line for modelled and observed $NH_4^+$ implies that organic diacids contributed at least a half to the conversion of ammonia to ammonium. Since more $NH_4^+$ existed in aerosol than required for neutralizing inorganic acids, the ISORROPIA model simulated higher equilibrium $NH_3$ than observed, leading to an overestimate of $NH_3$.

Comparisons between the measurements and the modelled values give a positive NMB for $NH_3$ (26.9%) and negative NMB for $SO_4^{2-}$ (-0.5%), $NO_3^-$ (-12.5%) and $NH_4^+$ (-32.6%). NME gives an estimation of overall discrepancy between measurement and model (Sudheer and Rengarajan, 2015). The NME values for the model-observation comparison of $NH_3$, $SO_4^{2-}$, $NO_3^-$ and $NH_4^+$ are 31.6%, 0.9%, 14.8% and 45.8%, respectively. These results indicate that the model simulated $SO_4^{2-}$ and $NO_3^-$ well but over-predicted $NH_3$ and under-predicted $NH_4^+$ by around 30% mainly because of the neglect of organics. Therefore, organics should be observed and included in the model simulations in future studies to better investigate aerosol components and gas-to-aerosol conversion.

[revised manuscript text omitted]
. Data in Fig. 10 seem to be distributed in two groups, with the one (Group 1) covering low to middle levels but exhibiting a larger slope and the other (Group 2) covering middle to high levels but exhibiting a smaller slope. Data in Group 2 were mainly from 10 August, 2013 when high $NH_4^+$, $SO_4^{2-}$ and $NO_3^-$ were observed (see Fig.

9). The difference in slope between the two groups might be caused by different fractions of organic diacids that are considered to convert more ammonia to ammonium. Although no measurements of organic diacids are available to prove this, the significant changes in fractions of organic and inorganic aerosol components are often observed during pollution episodes in China (e.g., Wang et al. 2016). Reduced impact from organic diacids may bring 
[revised manuscript text omitted]

20   Sudheer, A. K. and Rengarajan, R.: Time-resolved inorganic chemical composition of fine aerosol and associated precursor gases over an urban environment in western India: gas-aerosol equilibrium characteristics, Atmos. Environ., 109, 217-227, 2015.

Tang, X., Zhang, X.S., Ci, Z.J., Guo, J. and Wang, J. Q.: Speciation of the major inorganic salts in atmospheric aerosols of Beijing, China: Measurements and comparison with model. Atmos.

25   Environ., 133, 123-134, 2016.

Trebs, I., Meixner, F. X., Slanina J., Oties, R. P., and Andreae, M. O.: Real-time measurements of ammonia, acidic trace gases and water-soluble inorganic aerosol species at a rural site in the Amazon Basin, Atmos. Chem. Phys., 4, 967-987, 2004.

Walker, J. T., Robarge, W. P., Shendrikar, A., and Kimball, H.: Inorganic $PM_{2.5}$ at a U.S. agricultural

30   site, Environ. Pollut., 139, 258-271, 2006.

Walker, J. T., Whitall, D. R., Robarge, W., and Paerl, H. W.: Ambient ammonia and ammonium aerosol across a region of variable ammonia emission density, Atmos. Environ., 38, 1235-1246, 2004.

Wang, G., Zhang, R., Gomez, M. E., Yang, L., Zamora, M. L., Hu, M., Lin, Y., Peng, J., Guo, S., Meng, J., Li, J., Cheng, C., Hua, T., Ren, Y., Wang, Y., Gao, J., Cao, J., An, Z., Zhou, W., Li, G., Wang, J., Tian, P., Marrero-Ortiz, W., Secrest, J., Du, Z., Zheng, J., Shang, D., Zeng, L., Shao, M., Wang, W., Huang, Y., Wang, Y., Zhu, Y., Li, Y., Hu, J., Pan, B., Cai, L., Cheng, Y., Ji, Y., Zhang, F., Rosenfeld, D., Liss, P. S., Duce, R. B., Kolb, C. E, and Molina, M .J.: Persistent sulfate formation from London Fog to Chinese haze, PNAS, 113,13630-13635, 2016.

[revised manuscript text omitted]